# Stromule extension along microtubules coordinated with actin-mediated anchoring guides perinuclear chloroplast movement during innate immunity

Amutha Sampath Kumar[1†], Eunsook Park[2,3†‡], Alexander Nedo[1,4], Ali Alqarni[1,4,5], Li Ren[5], Kyle Hoban[1,4], Shannon Modla[1], John H McDonald[4], Chandra Kambhamettu[5,6], Savithramma P Dinesh-Kumar[2,3]*, Jeffrey Lewis Caplan[1,4,5]*

[1]Delaware Biotechnology Institute, University of Delaware, Newark, United States; [2]Department of Plant Biology, College of Biological Sciences, University of California, Davis, Davis, United States; [3]The Genome Center, College of Biological Sciences, University of California, Davis, Davis, United States; [4]Department of Biological Sciences, College of Arts and Sciences, University of Delaware, Newark, United States; [5]Department of Plant and Soil Sciences, College of Agriculture and Natural Resources, University of Delaware, Newark, United States; [6]Department of Computer and Information Sciences, College of Engineering, University of Delaware, Newark, United States

*For correspondence:
spdineshkumar@ucdavis.edu
(SPD-K);
jcaplan@udel.edu (JLC)

[†]These authors contributed equally to this work

Present address: [‡]Department of Plant Sciences, College of Agriculture and Life Sciences, Seoul National University, Seoul, Korea

Competing interests: The authors declare that no competing interests exist.

**Abstract** Dynamic tubular extensions from chloroplasts called stromules have recently been shown to connect with nuclei and function during innate immunity. We demonstrate that stromules extend along microtubules (MTs) and MT organization directly affects stromule dynamics since stabilization of MTs chemically or genetically increases stromule numbers and length. Although actin filaments (AFs) are not required for stromule extension, they provide anchor points for stromules. Interestingly, there is a strong correlation between the direction of stromules from chloroplasts and the direction of chloroplast movement. Stromule-directed chloroplast movement was observed in steady-state conditions without immune induction, suggesting it is a general function of stromules in epidermal cells. Our results show that MTs and AFs may facilitate perinuclear clustering of chloroplasts during an innate immune response. We propose a model in which stromules extend along MTs and connect to AF anchor points surrounding nuclei, facilitating stromule-directed movement of chloroplasts to nuclei during innate immunity.
DOI: https://doi.org/10.7554/eLife.23625.001

## Introduction

Stroma-filled tubular structures called stromules emanate from chloroplasts and have been observed in several genera in the plant kingdom, although they are most common in non-green plastids (*Gray et al., 2001*; *Hanson and Sattarzadeh, 2008*; *Köhler and Hanson, 2000*; *Kumar et al., 2014*; *Natesan et al., 2005*). Stromules are developmentally regulated and induced in response to biotic and abiotic stress, symbiotic association, and changes in plastid number and size (*Brunkard et al., 2015*; *Caplan et al., 2015*; *Caplan et al., 2008*; *Erickson et al., 2014*; *Gray et al., 2012*; *Kumar et al., 2014*; *Schattat and Klösgen, 2011*; *Waters et al., 2004*). The dynamic extension of stromules increases the surface area of chloroplasts, presumably facilitating transport of

**eLife digest** Within a plant's cells, compartments called chloroplasts harvest energy from sunlight. This process, termed photosynthesis, keeps the plant alive and growing. Yet this is not all that chloroplasts do. For example, if a harmful microbe infects the plant, its chloroplasts rapidly change shape and move toward the cell's nucleus – the compartment of the cell that contains the bulk of the plant's genetic material. The chloroplasts then send warning signals to the nucleus that boost the plant's defenses.

In 2015, researchers showed that, during an infection, tiny tubes called stromules extend out from chloroplasts and make contact with the nucleus. Stromules are flexible structures that can extend and retract. However, it is not yet understood what exactly the stromules do, or how they establish connections with the nucleus.

Some scientists had suggested the internal skeleton of the cell – a complex network of protein filaments called actin and microtubules – might regulate the movements of the stromules. Now, Kumar, Park et al. – including several researchers from the 2015 study – have monitored the interaction between stromules and this internal "cytoskeleton" in leaf cells from a plant called *Nicotiana benthamiana*. Stromules, microtubules and actin were marked with fluorescent tags, which allowed them to be tracked under a microscope. This showed that the stromules actively extend along microtubules and anchor to a network of actin filaments.

Further work showed that chloroplasts move in the direction of the stromules. This suggests that chloroplast movement may actually be directed by these structures. This movement was seen in healthy plants growing under normal conditions, suggesting that it may be a more general occurrence in plants. Next, Kumar, Park et al. used a viral protein to provoke an immune response in the plants, and saw that a larger number of chloroplasts moved and clustered around the nucleus. This chloroplast clustering was also guided by stromules anchored to the actin filaments that surround the nucleus.

These findings shed new light on how chloroplasts communicate with the nucleus. Future work is needed to determine if stromules just guide chloroplast movement or if they provide a physical force that drives this process.

DOI: https://doi.org/10.7554/eLife.23625.002

signals or macromolecules to the nucleus, cytosol, plasma membrane or other organelles (*Gunning, 2005*, *2004a*; *Kwok and Hanson, 2004c*). We have recently shown that stromules are induced and function during innate immunity (*Caplan et al., 2015*). The induced stromules make connections with the nuclei to facilitate transport of chloroplast-localized defense protein NRIP1 (N receptor interacting protein 1) and the pro-defense molecule, hydrogen peroxide ($H_2O_2$), from chloroplasts into nuclei during an immune response (*Caplan et al., 2015*). Stromules may also facilitate certain number of chloroplasts to maintain contact with the moving nuclei (*Erickson et al., 2017a*). However, the mechanism(s) that facilitates chloroplast stromules connections to nuclei and eventual peri-nuclear clustering of chloroplasts is unknown.

Stromule length is variable as they extend, retract and branch, changing their shape and position (*Gray et al., 2001*; *Gunning, 2005*; *Kwok and Hanson, 2004c*; *Waters et al., 2004*). However, mechanisms that regulate the dynamic nature of stromule morphology and motility are poorly understood. Studies using inhibitors in non-green tissue have implicated cytoskeleton elements such as actin microfilaments (AFs) and microtubules (MTs) in regulating stromule frequency, length and motility (*Gunning, 2005*; *Kwok and Hanson, 2003*; *Kwok and Hanson, 2004a*). Treatment with AF inhibitors, Cytochalasin D (CTD) and Latrunculin B, resulted in the reduction of stromule frequency in tobacco hypocotyls (*Kwok and Hanson, 2003*). Stromules have been observed to extend parallel to AFs and the tips of stromules make contact with AFs in Arabidopsis hypocotyl epidermal cells (*Kwok and Hanson, 2004a*). Treatment with myosin ATPase inhibitor 2,3 butanedione 2-monoxime (BDM) affects stromule movement and length; furthermore, Myosin XI family motor proteins have been implicated in stromule movement and anchoring to the cytoskeleton in *Nicotiana* (*Natesan et al., 2009*; *Sattarzadeh et al., 2009*). These findings suggest that stromules move along AFs using myosin motors; however, direct evidence for movement along AFs is lacking. Treatment

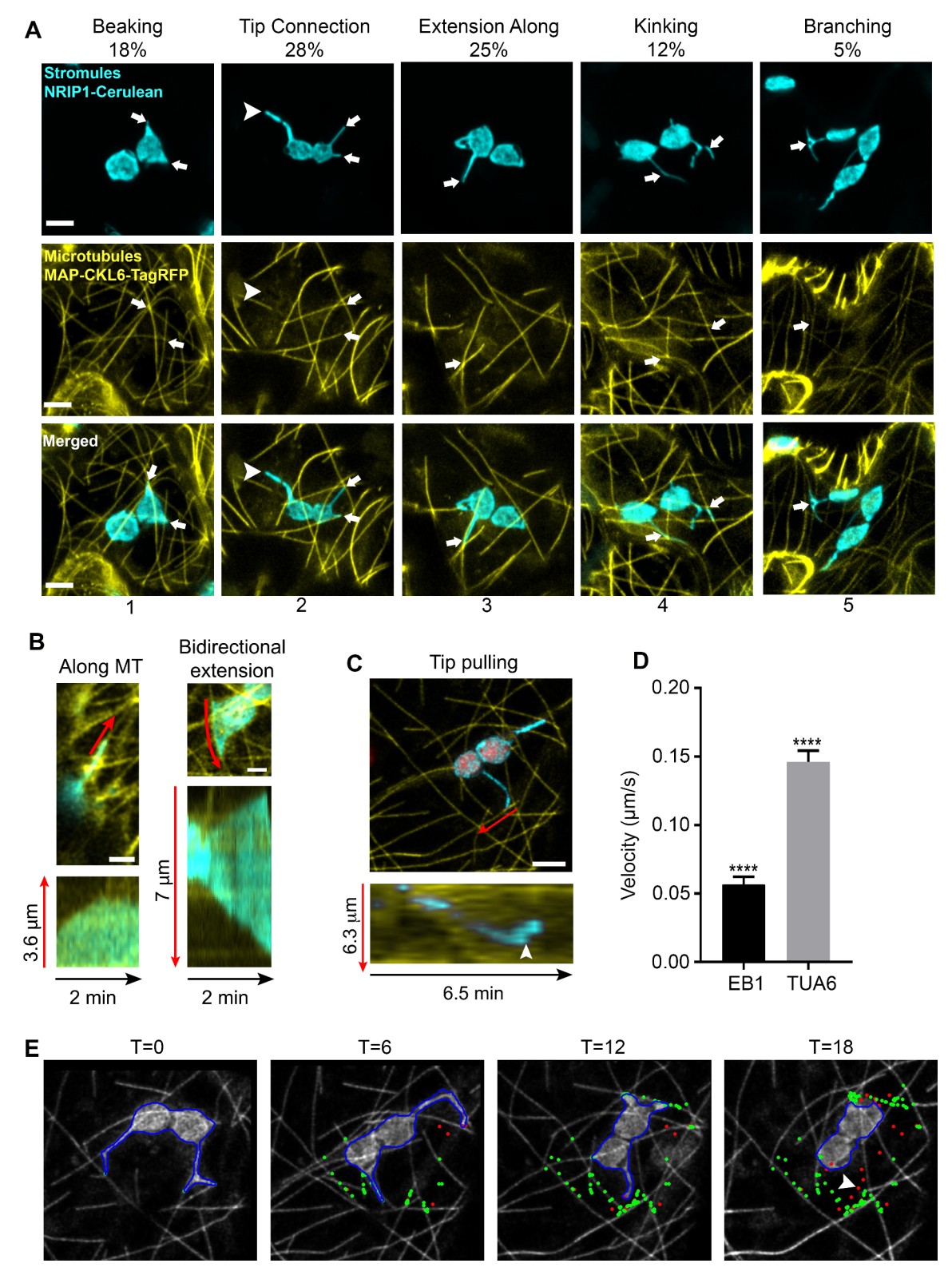

**Figure 1.** Chloroplast stromules extend along microtubules. (**A**) MTs were marked by transiently expressing TagRFP-MAP-CKL6 (yellow) in *N. benthamiana* transgenic plants expressing NRIP1-Cerulean that marks stromules (cyan). Confocal micrographs of stromule-to-MT interactions in lower epidermal pavement cells are shown. Arrows indicate stromule beaking (1), tip contact (2), extension along (3), kinking (4) and branching (5) associations with MTs. Approximately 11% of stromules were not attached to MTs (2, arrowhead). Images are maximum intensity projection of confocal z-stacks.
*Figure 1 continued on next page*

*Figure 1 continued*

Scale bars equal 2 µm. Total of 103 stromules were observed in 11 biological replicates to generate this data. (**B**) Stromules were marked by expressing NRIP1(cTP)-TagRFP in transgenic GFP-TUA6 *N. benthamiana* plants. Time-lapse images were acquired and kymographs over 2 min were generated. Stromules were observed extending along MTs (left) and in both directions along MTs (right). Kymographs (bottom) were generated adjacent to the red lines in top images. (**C**) MTs were marked by expressing EB1-Citrine (yellow) in *N. benthamiana* transgenic lines expressing NRIP1-Cerulean that marks stromules (cyan). Stromules were observed extending with only the tip being pulled along MTs. Kymographs (bottom) were generated adjacent to the red lines in top image and show that a stromule tip translated along a MT at a constant rate and then rapidly changed direction (arrowhead). (**D**) The average velocity of stromules along MTs was calculated from manually tracked stromule tips moving along MTs marked with EB1-Citrine or GFP-TUA6. Data represented as the mean standard error of the mean (SEM), ****$p<0.0001$ by a Student's t-test with Welch's correction. (**E**) A stromule tip was tracked using a combination of fuzzy c-means and active contour algorithm, with shape analysis to calculate the length of the stromule, the tip velocity and the association with microtubules (*Lu et al., 2017*). Tip associations (green dots) with MTs (gray scale) were mapped over a time series. Tips not associating with MTs are depicted as red dots. Moving stromule tips were associated with MTs except when stromules were retracting (arrowhead).

DOI: https://doi.org/10.7554/eLife.23625.003

The following source data and figure supplement are available for figure 1:

**Source data 1.** Statistical analysis of the stromule velocity along MT.

DOI: https://doi.org/10.7554/eLife.23625.005

**Figure supplement 1.** Transmission electron microscopy of stromule-to-microtubule interactions.

DOI: https://doi.org/10.7554/eLife.23625.004

with MT inhibitor amiprophosmethyl (APM) reduced stromules, and co-treatment with AF and MT inhibitors decreased stromule frequency and length (*Kwok and Hanson, 2003*). In contrast, 'chloroplast protrusions' from mesophyll chloroplasts of the arctic plant *Oxyria digyna* remained unaffected by the MT inhibitor Oryzalin or the AF inhibitor LatB (*Holzinger et al., 2007b*). Therefore, the precise role of AFs and MTs during stromule dynamics in green tissue chloroplasts is not well understood.

Here, we analyzed the mechanism of stromule extension and movement in chloroplasts of green leaf tissue and perinuclear chloroplast clustering during innate immunity. Our results show that MTs are required for stromule extension and movement. MT depolymerization led to stromule retraction, and MT stabilization increased stromule frequency. Silencing the gene for γ-tubulin complex protein 4 (GCP4) caused enhanced bundling and disrupted dynamics of MTs, which resulted in longer stromules, but slower extension and retraction. Although stromule extension does not require AFs, they function as anchor points that stabilize stromules and anchor the body of chloroplasts. AFs play an important role in type of chloroplast movement that appears to be directed by stromules. This new type of stromule-directed movement is completely disrupted by AF inhibitors. However, stromule-directed chloroplast movement was still observed when AFs were partially disrupted, suggesting that chloroplast anchoring might restrict stromule directed movement. We hypothesize that a biological function of stromules is to direct the movement of chloroplasts. During an innate immune response, we propose a model where stromules extend along MTs towards nuclei and attach to the nuclei at actin anchor points; and, these perinuclear stromule attachments guide chloroplasts to the nucleus.

## Results

### Stromules interact and extend along microtubules

To examine the interactions of stromules with MTs, we expressed TagRFP fused to the N-terminal microtubule-associated protein domain of CKL6 (*Ben-Nissan et al., 2008*) (TagRFP-MAP-CKL6) in transgenic *Nicotiana benthamiana* plants expressing NRIP1 fused to Cerulean (NRIP1-Cerulean) that mark stromules (*Caplan et al., 2015*; *Caplan et al., 2008*). Marking both stromules and MTs revealed that these two structures overlapped in confocal microscopy images. These sites of overlap were designated as potential stromule-to-MT interactions. These observations were made in maximum intensity projections of z-stacks generated by confocal microscopy, and all observations in this study, were made in epidermal pavement cells of *N. benthamiana* plants. The varied morphology of stromules appeared to be correlated with MT interactions (*Figure 1*). Stromules often initiate as beak-like structures. The tips of beaks were seen interacting with MTs (*Figure 1A*; column 1). Beaks

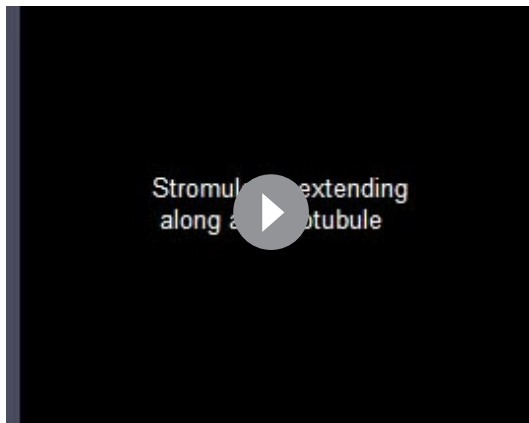

**Video 1.** Stromules extend along microtubules. Stromules were marked by expressing NRIP1(cTP)-TagRFP (Cyan) in GFP-TUA6 (yellow) transgenic *N. benthamiana* plants. Time-lapse confocal microscopy was used to acquire images every 0.92 s that are displayed at 30 frames/s. The first half of the video (148 frames) shows a single stromule extending along microtubules and the second half (164 frames) of a stromule moving bidirectionally along microtubules. This video was used to generate the kymographs in *Figure 1B*.
DOI: https://doi.org/10.7554/eLife.23625.006

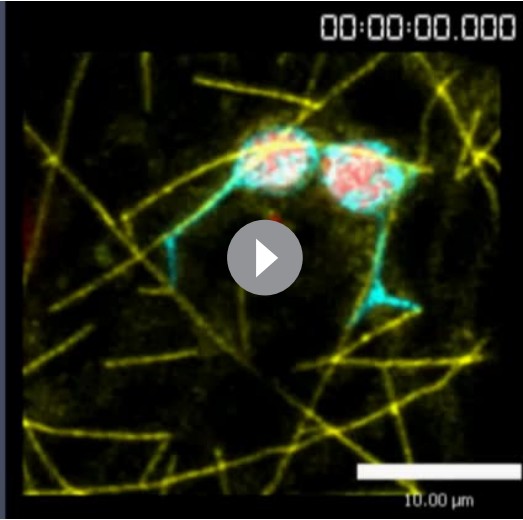

**Video 2.** Tracking of stromule tip interactions along MTs. The first half of the video shows maximum intensity projections of time-lapse confocal microscopy of 120 images taken every 8.1 s and displayed at 8 frames/second. MTs were marked by expressing EB1-Citrine (yellow) in transgenic *N. benthamiana* plants expressing NRIP1-Cerulean that marks stromules (cyan). Chlorophyll autofluorescence is pseudo-colored in red. Stromules were observed extending with only the tip being pulled along MTs. The second half of the video shows the tracking of stromule tips using the method described in *Figure 1—figure supplement 1*. The contours of stromules were defined in blue. The tip associations (green dots) with microtubules (grayscale) and non-associating tips (red dots) were mapped over a time series. The video was used to generate *Figure 1C–E*.
DOI: https://doi.org/10.7554/eLife.23625.007

extend into longer stromules. Longer stromules were seen as just the tips of stromules interacting with MTs or the tip and the full length of the stromule aligned with MTs (*Figure 1A*; columns 2 and 3). More complex stromule structures, such as kinked or branched stromules, were found at the junction of two MTs (*Figure 1A*; columns 4 and 5). However, approximately 11% of stromules did not interact with MTs (*Figure 1A*, arrowhead), suggesting there is a MT-independent mechanism of stromule formation.

A stromule-to-MT interaction was designated if these two structures were overlapping or not resolvable by confocal microscopy. However, since the resolution of confocal microscopy is relatively low, we verified the close interaction between stromules and MTs using transmission electron microscopy (TEM). Microtubules were originally detected and described in plants using TEM and can readily be observed as hollow, tubule-like structures that are 24 nm in diameter (*Ledbetter and Porter, 1963*). We were able to observe MTs by TEM and the close interactions of MTs with stromule tips and kink points (*Figure 1—figure supplement 1*). MTs were seen directly associated with the chloroplast outer envelope membrane at a kink point. Serial sections near the tip of a stromule that graze the chloroplast outer envelope membrane show MTs in line and in close proximity to the stromule.

Since our initial observations were from static images of stromules interacting with MTs, to look at the dynamics of stromules along MTs, we used an established transgenic *N. benthamiana* MT marker line expressing green fluorescent protein fused with the tubulin alpha 6 (GFP-TUA6) (*Gillespie et al., 2002*). In this transgenic line, we expressed NRIP1's chloroplast transit peptide fused to TagRFP [NRIP1(cTP)-TagRFP] to mark stromules (*Caplan et al., 2015*). Time-lapse imaging of GFP-TUA6-labeled MTs revealed that stromules dynamically extended along MTs (*Figure 1B*, *Video 1*). Kymographs of the motion showed stromules extending and retracting in line with MTs in a single direction (*Figure 1B*, left kymograph) or moving bi-directionally in opposite directions

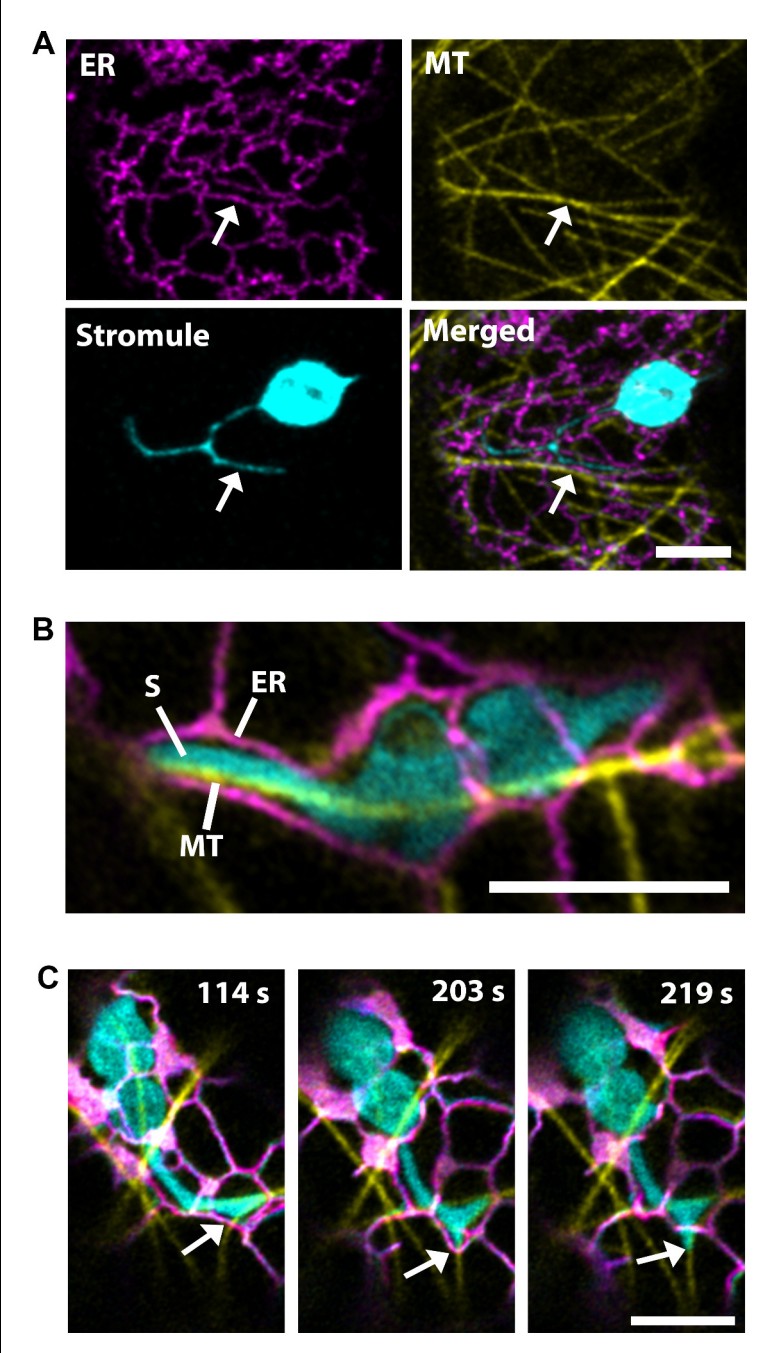

**Figure 2.** Microtubules direct stromule movement along the ER. (**A**) The endoplasmic reticulum (ER) were marked by transiently expressing SP-Citrine-HDEL (magenta) and MTs were marked by TagRFP-MAP-CKL6 (yellow) in *N. benthamiana* transgenic lines expressing NRIP1-Cerulean that marks stromules (cyan). Confocal microscopy time-lapse images of lower epidermis of leaves indicating changes in stromule extension along MTs and push through the ER network (bottom right, merged). Arrows indicate stromule extension within an ER channel (ER) along MT. (**B**) High resolution airyscan confocal micrograph showing the interaction of the stromule (**S**) with the MT within the ER channel. (**C**) High-resolution time-lapse images showing the extension of stromules (cyan) along the MTs (yellow) within and away from the ER channel (magenta). Arrows indicate active stromule extension, while the ER follows the course of the extending stromule (203 s) followed by the ER remodeling (219 s) independent of the stromule extension. Scale bars equal 5 μm.

DOI: https://doi.org/10.7554/eLife.23625.008

(*Figure 1B*, bottom right kymograph). We also verified the stromule movement using another MT marker, the end binding one protein (*Chan et al., 2003*) fused to Citrine (EB1-Citrine). EB1-Citrine was initially chosen to examine the direction of movement because EB1 marks the positive end of MTs; however, the *Agrobacterium*-mediated expression often led to even staining of the MTs (*Figure 1C*). Kymographs of the motion of stromules showed clear movement of stromule tips along MTs (*Figure 1C*). The time lapse video show the dynamic interactions of stromules with MT, including branching, tip contact, transfer between microtubules, stromule initiation and bidirectional extension (*Video 2*). Our results from using three different MT markers via transgenic and transient expression indicate that stromules extend along MTs.

To quantify the motion, we manually tracked the velocity of stromules extending along MTs. All stromule extension was correlated with movement along MTs. The velocity of stromule extension was significantly lower when MTs were marked with EB1-Citrine (0.0565 µm/s) compared to GFP-TUA6 (0.146 µm/s) (*Figure 1D*). An automated algorithm for detecting stromule tips in maximum intensity projections was developed (*Lu et al., 2017*). The MTs were segmented and skeletonized (data not shown). Using the skeletonized images, the points of interaction (*Figure 1E*, green) and the points of no interaction (*Figure 1E*, red) were mapped over a time series (*Video 2*). Linear arrays of interaction points along MTs were clearly seen in time points T = 6, 12, and 18 min (*Figure 1E*). A retraction event had limited interaction with MTs (*Figure 1E*, arrowhead). The algorithm only accurately detected the slower moving motion when EB1-Citrine was used as a MT marker, and therefore, was not used in other experiments. The length, velocities, extension and retraction frequencies, and types of motion were quantified manually in all other experiments.

## Stromule extensions through the ER is directed by microtubules

Stromules and the endoplasmic reticulum (ER) have correlated dynamics and three-dimensional arrangement; therefore, it was hypothesized that contact points along the ER direct their extension (*Schattat et al., 2011*). Since, our data suggested that the extension of stromules is directed by MTs, we examined stromules, ER and MTs simultaneously by co-expression of labels for the ER (SP-Citrine-HDEL) and MTs (TagRFP-MAP-CKL6) in NRIP1-Cerulean transgenic *N. benthamiana* plants that mark stromules. Similar to the previous report (*Schattat et al., 2011*), stromules were surrounded by ER, but here we show that MTs direct the movement through ER (*Figure 2A*; *Video 3*). Imaging using a high-resolution airyscan confocal microscope revealed that the ER forms channels around the stromules, and MTs were found at the stromule-to-ER interface (*Figure 2B*). Time lapse studies under similar imaging conditions showed that stromule extension occurred actively along the MTs, while the ER changed its direction and formed a channel around the extended stromule tip (*Figure 2C*, middle panel; *Video 4*). The stromule continued to extend along the MTs past the ER and no longer formed a channel around the extending stromule tip (*Figure 2C*, right panel). These time-lapse studies indicate that stromule extension is active along MTs and ER reorganization follows stromule extension along MTs.

## Microtubules are required for stromule formation and extension

To further demonstrate that stromules extend along MTs, we expressed the MT marker TagRFP-MAP-CKL6 in NRIP1-Cerulean transgenic *N. benthamiana* plants and then disrupted MTs using 20 µM APM or 300 µM Oryzalin. Compared to the DMSO vehicle control (*Figure 3A*, top panel), depolymerization of MTs was noticeable 5 to 15 min after APM and Oryzalin treatment, leaving behind remnants of partially depolymerized MTs and an increase of the MAP-CKL6 MT marker in the cytosol (*Figure 3A*, middle and bottom panels). Although mock control with DMSO resulted in an increase in stromules compared to the infiltration media control, the APM or Oryzalin disruption of MTs for 15 min significantly inhibited this increase in stromule number (*Figure 3B*). MT depolymerizers APM and Oryzalin not only decreased stromule number but also restricted stromules to MT fragments causing changes in stromule movement (*Video 5*). Beak-like protrusions from chloroplasts that did not result in stromules were also observed in APM and Oryzalin treatment (*Figure 3A*, asterisk; *Video 5*), however, we did not determine if these increased with the treatments compared to the DMSO vehicle control. In the time-lapsed data set shown in *Figure 3A*, stromule length was gradually reduced during 15-min treatment with APM that was caused by stromule retraction and correlated with simultaneous depolymerization of the MTs and, eventually, complete retraction of

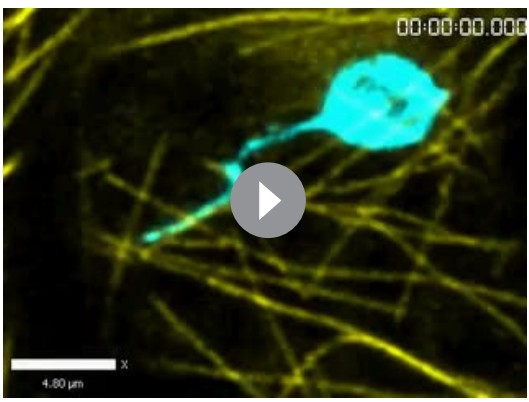

**Video 3.** Stromule move in channels of ER along MTs. Video show stromules (cyan) first moving along MTs (yellow) to show dynamic association, then merged with the ER (magenta), and with stromules and ER only to show movement through ER channels. The video was used to generate *Figure 2A*.

DOI: https://doi.org/10.7554/eLife.23625.009

stromules (*Figure 3A*, middle and bottom panels, arrowhead). Similarly, with Oryzalin treatment at 0 min, we observed a region of the stromule overlapped with a segment of the MT (*Figure 3A*, bottom panel, T = 0). As the time course progressed, the segment of MT became shorter and there was a corresponding reduction in the length of the stromule (*Figure 3A*, bottom panels). In that time-lapsed data set, at 15-min treatment with Oryzalin we observed that stromules completely retracted from and changed course from the MT (*Figure 3A*, bottom panel, T = 15; *Video 5*). These results indicate that the disappearance of stromules may be a combination of disruption of extended stromules and the prevention of induction of new stromules.

## Changes in microtubule organization increases stromule number, length and stability

Since our results from MT inhibitor studies indicated that stromule formation and extension require MTs, we tested the effect of stabilizing MTs using Taxol (*Schiff and Horwitz, 1980*). Infiltration of Paclitaxel-BODIPY conjugate into leaves of transgenic NRIP1-Cerulean *N. benthamiana* plants showed extensive MT stabilization after 30 min of treatment compared to the mock control (*Figure 3C*) and significantly induced stromules compared to mock control (*Figure 3C–D*). Interestingly, after Paclitaxel treatment, we observed long stromules and multiple stromules emanating from individual chloroplasts (*Figure 3C*, bottom panels). These results suggest that MT stabilization is sufficient to induce stromules.

To more specifically alter MT organization and dynamics, we knocked-down the expression of *GCP4* in *N. benthamiana* plants using virus-induced gene silencing (VIGS) approach (*Dinesh-Kumar et al., 2003*). GCP4 is a subunit of the γ-tubulin complex and artificial miRNA (amiR)-mediated knockdown of *Arabidopsis GCP4* resulted in hyper-parallel and bundled cortical MT in leaf epidermal cells (*Kong et al., 2010*). We silenced *NbGCP4* in NRIP1-Cerulean and GFP-TUA6 transgenic *N. benthamiana* plants to visualize the effect on stromules and MTs, respectively. Since *amiR-AtGCP4* in *Arabidopsis* plants resulted in a significant growth phenotype, we first determined how many days of *NbGCP4* VIGS resulted in a MT alteration without a severe growth phenotype. Four days after silencing, *NbGCP4*-silenced plants phenotypically looked similar to that of VIGS vector control plants (*Figure 4—figure supplement 1A*). However, six days post-silencing, leaves of the *NbGCP4*-silenced plants developed a crinkled leaf phenotype (*Figure 4—figure supplement 1B*, right panel). In addition, at this time point, the NRIP1-Cerulean stromule marker begin to leak out of chloroplasts compared to the VIGS control plants (*Figure 4—figure supplement 1C*, right panel). Fourteen days post-silencing, *NbGCP4*-silenced plants showed severe growth arrest and morphological distortion (*Figure 4—figure supplement 1D*). Thus, we observed stromules in leaf epidermal cells of the plants after 4 days of *NbGCP4* VIGS, to minimize potential physiological changes that might occur due to the alterations of MT organization

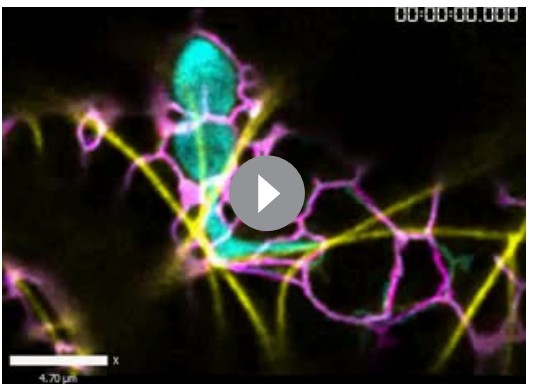

**Video 4.** Stromules remodeling ER. Stromules (cyan) move along MTs (yellow) remodeling the ER (magenta). Video was created from three consecutive time series and was used to generate *Figure 2B–C*.

DOI: https://doi.org/10.7554/eLife.23625.010

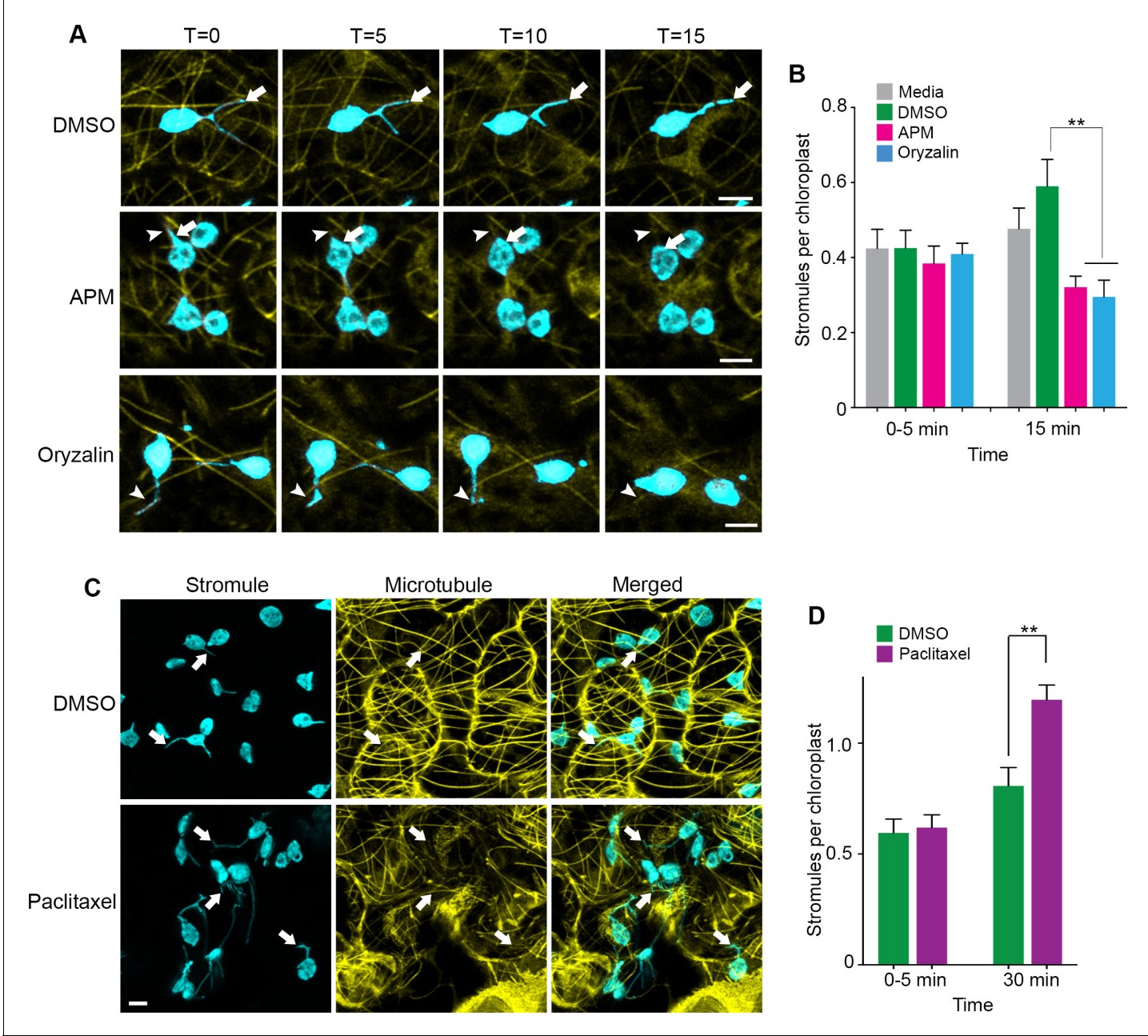

**Figure 3.** Microtubule disruption lead to stromule retraction and microtubule stabilization increased stromules. (**A**) Dynamics of MT depolymerization and stromule retraction in the lower epidermis of NRIP1-Cerulean transgenic *N. benthamiana* plant leaves after infiltration of mock control (0.2% DMSO) or MT depolymerizing agents, APM (20 μM) or Oryzalin (300 μM). Images are maximum intensity projections represented at time points with a 5-min interval after infiltration. Arrows in mock (top row) indicate extended and branching stromules. MT depolymerization due to APM and Oryzalin causes simultaneous stromule retraction within 15 min (arrowhead; middle and bottom panels). A beaking was initiated but failed to progress to stromule (asterisk; middle and bottom panels). Scale bar equals 2 μm. (**B**) Stromules were increased by DMSO vehicle control treatment from 0 to 15 min compared to no significant increase from the infiltration media control. Compared to the DMSO vehicle control, stromules significantly decreased after treatment with APM (20 μM) and Oryzalin (300 μM) at 15 min and no other comparisons were significant. The experiment was repeated four times with three to five replicates per experiment. Error bars represent mean ± standard error of the mean (SEM) **p<0.05. (**C**) Treatment with microtubule stabilizing agent Paclitaxel (0.8 nM) produced multiple stromules from single chloroplast after 30 min (arrows, bottom left panel). The extended stromules overlapped with the MTs (arrows, merged panel). Images are maximum intensity projections. Scale bar equals 5 μm. (**D**) Paclitaxel treatment increased stromules per chloroplasts after 30-min treatment compared to mock treated leaves. The experiments were repeated four times with two replicates per experiment. Error bars indicate mean ± standard error of the mean (SEM) **p<0.05 by a Student's t-test with Welch's correction.

DOI: https://doi.org/10.7554/eLife.23625.011

*Figure 3 continued on next page*

*Figure 3 continued*

The following source data is available for figure 3:

**Source data 1.** Statistics of the stromule frequency under MT disrupting drug treatments.
DOI: https://doi.org/10.7554/eLife.23625.012

and dynamics. Although at 4 days-post silencing, *NbGCP4* mRNA levels are reduced by only 50% in the silenced plants compared to the VIGS control plants (*Figure 4B*), cortical MT organization was significantly altered in the leaves of *NbGCP4*-silenced plants compared to the control (*Figure 4A*) in a similar way to *amiR-AtGCP4* in *Arabidopsis* (*Kong et al., 2010*). To quantify these changes, we used SOAX software that uses Stretching Open Active Contours (SOACs) to quantify filamentous networks (*Xu et al., 2015*). SOAX analysis showed that MTs were more parallel or aligned in *NbGCP4*-silenced plants compared to the vector control (*Figure 4C*), which was visible by displaying the MT direction by color-coding the azimuthal angles (*Figure 4D*). Quantitative SOAX analysis shows that silencing *NbGCP4* decreases the curvature (*Figure 4E*) and increases the snake length fitted to MTs (*Figure 4F*). The snake length is not a direct measurement of MT length, since this approach cannot accurately distinguish between two MTs that are bundled together. Nonetheless, this measurement further suggests that silencing *NbGCP4* alters MTs.

The alteration in MT organization at four days-post silencing of *NbGCP4* (*Figure 4*), resulted in more than twice the number of stromules in *NbGCP4*-silenced plants compared to the VIGS vector control (*Figure 5A*, top panels; 5B, compare bars in mock treatment). Stromules were on average significantly longer in *NbGCP4*-silenced plants compared to VIGS vector control (*Figure 5A*, top panels; *Figure 5C*, compare bars in mock treatment). Furthermore, a greater percentage of stromules were longer than 3 µm in *NbGCP4*-silenced plants (*Figure 5—figure supplement 1A*). We classified stromule movement into three types, smooth and constant movement, sudden and erratic movement, and side and tangential movement (*Figure 5—figure supplement 1B*) and found that, in *NbGCP4*-silenced plants, stromule movements were more constant than those in VIGS vector control (*Figure 5—figure supplement 1C*).

We recently reported that stromules are induced significantly during an immune response against bacterial and viral infections (*Caplan et al., 2015*). The nucleotide-binding domain leucine-rich repeat (NLR) immune receptor N recognizes p50 effector from Tobacco Mosaic Virus (TMV) and activate immune response to limit TMV to the infection site (*Whitham et al., 1994*). The stromules are significantly induced during N NLR-mediated immunity to TMV (*Caplan et al., 2015*). Therefore, we tested if N NLR-mediated activation of immune response could further increase stromule number and length in *NbGCP4*-silenced plants. For this, we silenced *NbGCP4* in transgenic *N. benthamiana* expressing N NLR and NRIP1-Cerulean (stromule marker) for 3 days and then infiltrated with p50 and 24 hr later the observations were recorded. As shown before (*Caplan et al., 2015*), the number of stromules significantly increased in p50-treated VIGS vector control plants compared to mock-treatment (*Figure 5A*, compare left panels and 5B, compare green bars). The average length (*Figure 5C*, green bars) and percentage of stromules longer than 3 µm also increased during an immune response (*Figure 5—figure supplement 1A*). Interestingly, the increase in stromules in mock-treated *NbGCP4*-silenced plants and a p50-induced immune response in VIGS vector control were remarkably similar (*Figure 5A*, compare top right panel with bottom left panel; *Figure 5B*, compare mock-treated magenta bar with p50-treated green bar). There was no significant change in stromule number in p50-treated *NbGCP4*-silenced plants compared to the mock-treated *NbGCP4*-silenced plants (*Figure 5A*, right panels and 5B, compare magenta bars).

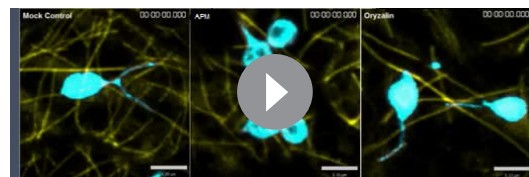

**Video 5.** Stromule dynamics are disrupted by APM or Oryzalin MT inhibitor treatment. Disruption of MT (yellow) by APM (20 µM) resulted in a loss of stromule extensions (middle) compared to the mock treatment control (left). Disruption of MT (yellow) by Oryzalin (300 µM) resulted in a loss of stromule extension compared to the mock treatment control (left). Maximum intensity projections of time-lapse z-stacks were taken every 40 s for Mock and Oryzalin and every 29 s for APM. Videos are 143 fames displayed at 15 frames/s. The video was used to generate *Figure 3A*.
DOI: https://doi.org/10.7554/eLife.23625.013

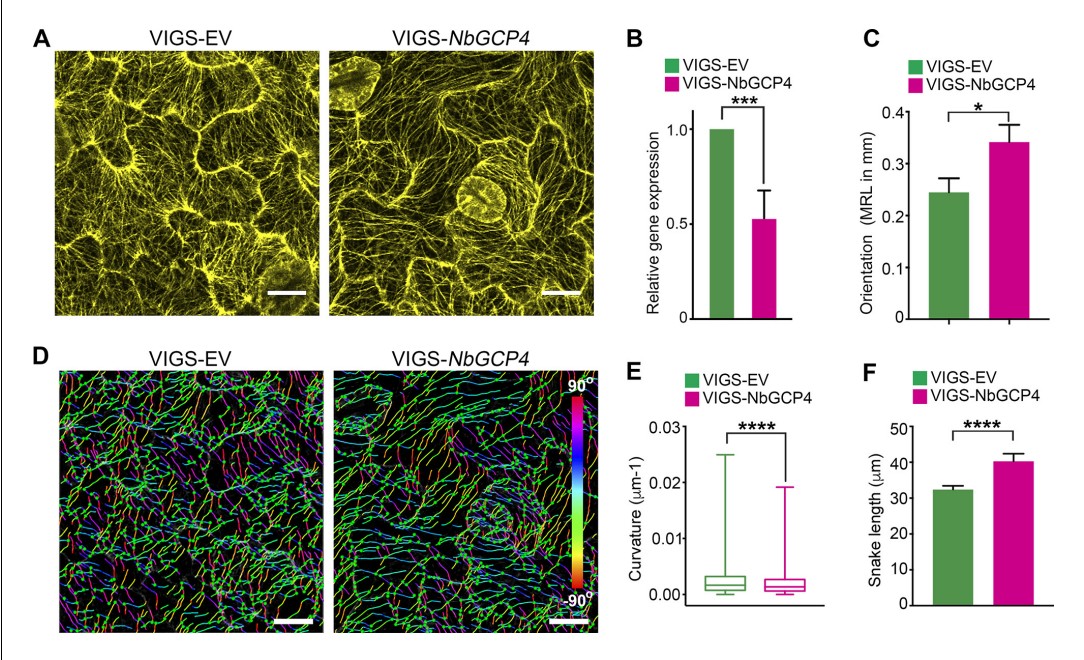

**Figure 4.** Alterations of MT is correlated with NbGCP4 silencing. (A) In GFP-TUA6 transgenic *N. benthamiana* leaves that mark MTs (yellow), *NbGCP4* silencing resulted in hyper-parallel and occasional bundling of MT (VIGS-*NbGCP4*, right) compared to the VIGS vector control (VIGS-EV, left). Images are maximum intensity projections of confocal z-stacks. Scale bar represents 40 μm. (B) qRT-PCR analysis of transcript levels of *NbGCP4* in VIGS-EV control and VIGS-*NbGCP4* plants showed a significant decrease in *NbGCP4* transcript level 4 days after silencing by VIGS-*NbGCP4* compared to the VIGS-EV control. Data represented as the mean +standard error of the mean (SEM), n = 12, ***p<0.001 (Student's t-test). (C) Azimuthal angles were analyzed by converting angles to mean resultant length (MRL) by converting the MT angles into individual vectors, adding the vectors together, and calculating the mean. MRL values are between 0 and 1, with 0 indicating that MT angles are random and one indicating all MT angles are the same and completely aligned. Data represented as the mean + standard error of the mean (SEM), *p<0.01 (student's t-test). (D) SOAX analysis was conducted on the images in (A). MT filaments are color-coded based on the azimuthal angle so that parallel MTs are the same color. (E) Curvature analysis that measures the rate of change of tangent vectors shows MTs in VIGS-*NbGCP4* have less curvature. Box covers from first to third quartiles while a bar in the middle of the box indicates median. Whiskers show from minimum to maximum. ****p<0.0001 by Mann-Whitney test. (F) Analysis of the snake length computed by SOAX analysis showed an increase in MTs length in VIGS-*NbGCP4*. Data represented as median and 95% confidence interval. ****p<0.0001 by Mann-Whitney test.

DOI: https://doi.org/10.7554/eLife.23625.014

The following source data and figure supplement are available for figure 4:

**Source data 1.** Statistics of the quantitative RT-PCR to validate *GCP4* gene silencing.
DOI: https://doi.org/10.7554/eLife.23625.016
**Source data 2.** Statistics of the Azimuthal angles of the MT in VIGS-*NbGCP4* plants.
DOI: https://doi.org/10.7554/eLife.23625.017
**Source data 3.** Statistics of the curvature of MTs in VIGS-*NbGCP4* plants.
DOI: https://doi.org/10.7554/eLife.23625.018
**Source data 4.** Statistics of the SOAX analysis of MTs in VIGS-*NbGCP4* plants.
DOI: https://doi.org/10.7554/eLife.23625.019
**Figure supplement 1.** Plant morphology at different days post-silencing of NbGCP4 in NRIP1-Cerulean expressing *N. benthamiana* plants.
DOI: https://doi.org/10.7554/eLife.23625.015

Mock-treated *NbGCP4*-silenced plants also showed longer stromules compared to mock-treated VIGS vector control plants (*Figure 5C*). This increase was similar to that of in p50-treated VIGS vector control plants that showed significantly longer stromules compared to mock-treated plants (*Figure 5C*, compare green open bars). However, there was no significant difference in stromule length in p50- and mock-treated *NbGCP4*-silenced plants (*Figure 5C*, compare magenta open bars). Collectively, these results indicate that the activation of immune response does not further increase stromule number and length in *NbGCP4*-silenced plants that exhibit constitutive stromule induction.

The velocities of stromule extension and retraction were calculated as an indicator of stromule dynamicity and stability. The stromule extension and retraction velocities decreased in the *NbGCP4*-

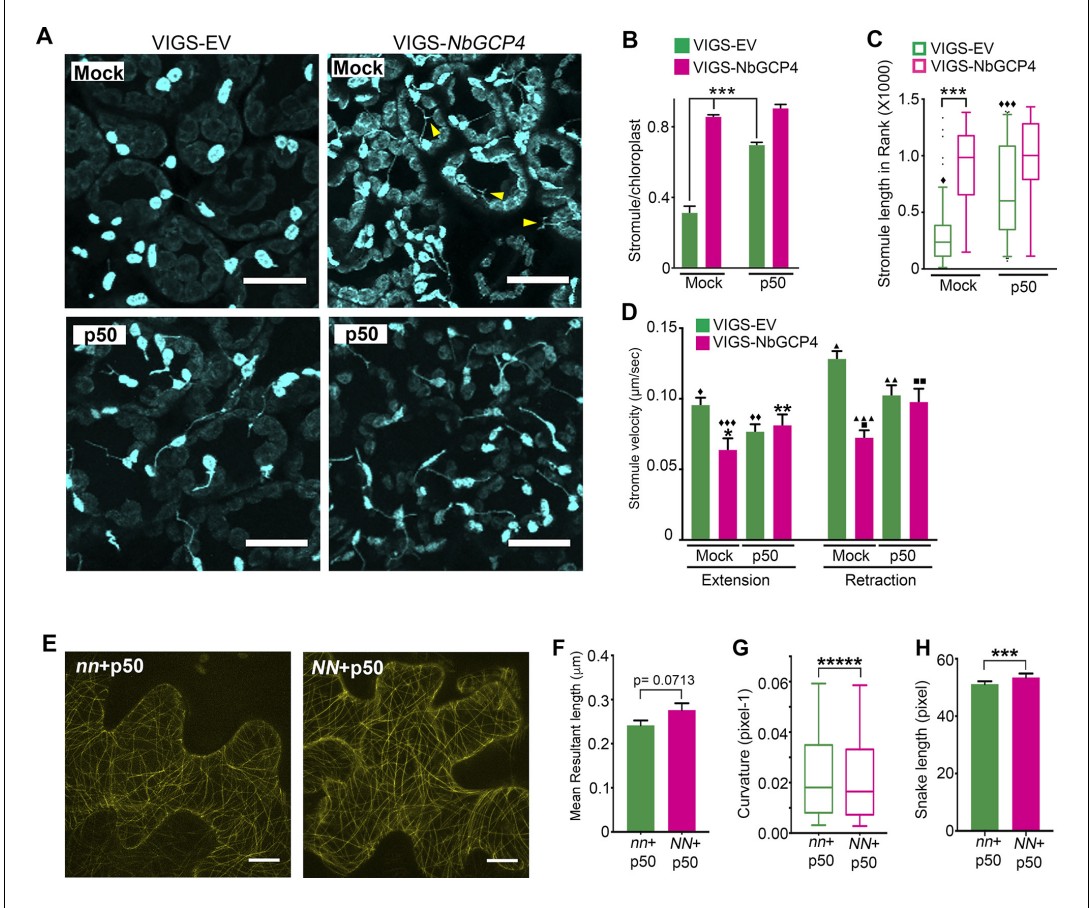

**Figure 5.** Microtubule stabilization induces stromules constitutively. (A) Stromules were induced 24 hr after TMV-p50 effector infiltration as described in *Caplan et al. (2015)* (bottom left) compared to a mock control (top left). Stromules were induced both in mock (top right) and p50 infiltrated (bottom right) *NbGCP4*-silenced plants indicating that *NbGCP4* silencing induces stromules constitutively. Stromules were often long and branched in mock-treated *NbGCP4*-silenced plants (top right, yellow arrowheads) compared to the VIGS-EV control (top left). Scale bar represents 40 μm. (B) Quantitative representation of stromules from A. Stromules were significantly induced in mock-treated *NbGCP4*-silenced plants compared to the mock-treated VIGS-EV control (compare left green bar and magenta bar). Stromules increased significantly in VIGS-EV control plants treated with p50 effector compared to the mock (compare green bars). No significant difference in stromule number was observed in *NbGCP4*-silenced mock-treated plants compared to the p50-treated plants (compare magenta bars). Four images per leaf were generated for quantification from total of 12 leaves for each condition. Data represented as the mean +standard error of the mean (SEM), ***p<0.001 (Student's t-test with Welch's correction). (C) Stromule length was significantly increased in mock-treated VIGS-*NbGCP4* plants compared to the VIGS-EV control (left open bars). p50-effector-induced immune response increased stromule length in VIGS vector control plants compared to the mock-treated VIGS-EV control plants (compare green open bars). No significant change was observed between p50-treated and mock-treated VIGS-*NbGCP4* plants (compare red open bars). Box and whisker plot was drawn with rank transformation. Box cover from first to third quartiles while a bar in the middle of box indicates median. Whiskers show from 5% to 95% of ranking. ***p<0.001, ♦ comparison with VIGS-EV control, p<0.001 by Mann-Whitney test. Dots in the graph indicate outliers. (D) The velocity of stromule extension and retraction in VIGS-EV control and VIGS-*NbGCP4* with or without TMV-p50-induced immune response. Data represented as the mean + standard error of the mean (SEM). Symbols at the top of bars indicate significant differences according to Dunnett's multiple comparison test. Single symbol (*,♦, ■), control set for each comparison; two-symbols (**,♦♦, ■■), p<0.05 and three-symbols (***,♦♦♦, ■■■), p<0.001. Scale bars equal 40 μm. (E) TMV-p50-induced immune response resulted in hyper-parallel MTs (*NN* + p50, right) compared to the control (*nn* +p50, left) in transgenic *N. benthamiana* leaves that mark MTs (yellow). Images are maximum intensity projections of confocal z-stacks. Scale bar represents 20 μm. (F) Azimuthal angle differences of MT filaments were measured by the length of the arc. Data represented as the mean +standard error of the mean (SEM), p=0.0713 (Student's t test with Welch's correction). (G) Curvature analysis that measures the rate of change of tangent vectors shows MTs in *NN* + p50 have less curvature. Box covers from first to third quartiles while a bar in the middle of box indicates median. Whiskers show from minimum to maximum. *****p<0.00001 (Mann-Whitney test). (H) Analysis of the snake length computed by SOAX analysis showed an increase in MTs length in *NN* + p50. Data represented as median and 95% confidence interval. ***p<0.001 by Mann-Whitney test.

DOI: https://doi.org/10.7554/eLife.23625.020

The following source data and figure supplements are available for figure 5:

**Source data 1.** Statistics of the stromule dynamics in VIGS-*NbGCP4* plants.

*Figure 5 continued on next page*

*Figure 5 continued*

DOI: https://doi.org/10.7554/eLife.23625.023
**Source data 2.** Statistics of the MT alteration by TMV-p50 induced immune responses.
DOI: https://doi.org/10.7554/eLife.23625.024
**Figure supplement 1.** Quantitative analysis of stromule length and movements.
DOI: https://doi.org/10.7554/eLife.23625.021
**Figure supplement 1—source data 1.** Statistics of quantitative analysis of stromule length and movement in VIGS-*NbGCP4* plants.
DOI: https://doi.org/10.7554/eLife.23625.022

silenced plants compared to the VIGS vector control (*Figure 5D*), suggesting that stromules were less dynamic and more stable. These results indicate that specific alterations of MTs are correlated with change in stromule dynamics and further support a role for MTs in regulating stromules. Interestingly, p50-treated VIGS vector control compared to the mock treatment reduced the velocities of stromule extension and retraction (*Figure 5D*, compare green bars) suggesting that stromules are less dynamic and more stable during active immune response. To test if p50-induced immunity alters MT organization resulting in alteration in stromule dynamics, we observed MT dynamics upon TMV-p50 treatment. For this, MT marker TagRFP-MAP-CKL6 was infiltrated into transgenic *N. benthamiana* plants expressing N NLR and NRIP1-Cerulean (*NN*) or expressing only NRIP1-Celulean without N NLR (*nn*). 12 hr later, p50 was infiltrated into the same spot to induce an immune response. After 48 hr of TagRFP-MAP-CKL6 expression and 36 hr of p50 expression, the MT cytoskeleton was imaged and then analyzed by SOAX. Visible differences in MTs between immunity-induced plants (*Figure 5E*, *NN* + p50) and non-immunity-induced plants (*Figure 5E*, *nn* +p50) were difficult to observed in the images, but interestingly, SOAX analysis revealed that p50-induced immunity altered MT morphology (*Figure 5F–H*). Specifically, there were minor differences in orientation (*Figure 5F*), curvatures were significantly smaller (*Figure 5G*) and snake lengths were larger (*Figure 5H*) in *NN* + p50 compared to *nn* +p50. Collectively, these results indicate that changes in MT organization caused by *NbGCP4*-silencing plants or during p50-induced immunity are correlated with changes in stromule dynamics, indicating a possible direct or indirect role for MT organization in modulating stromule dynamics.

## Actin filaments serve as anchor points but not as tracks for stromule extension

Since AFs were previously shown to regulate chloroplast movement and stromule morphology (*Kwok and Hanson, 2003*; *Kwok and Hanson, 2004a*), we tested if stromules extend along AFs. We expressed Lifeact-TagRFP that labels AF (*Era et al., 2009*; *Riedl et al., 2008*) in transgenic *N. benthamiana* plants expressing NRIP1-Cerulean that marks stromules (*Caplan et al., 2015*; *Caplan et al., 2008*). Out of 73 stromule tip extension events from 34 cells, the vast majority (93%) of stromule tip extensions were not observed along AFs. Stromules were occasionally observed to be aligned with AF (*Figure 6A*, asterisk), but high-resolution examination showed that they were not co-localizing (*Figure 6—figure supplement 1*). Instead, in many cases, stromules interacted at restricted foci (*Figure 6A*, arrowheads) that often corresponded with a kink in the stromule. We verified these interactions using TEM and found an AF bundle in close proximity to the apex of a stromule kink (*Figure 6—figure supplement 2A–B*). Stromule tips often reached actin filaments (*Figure 6A*, arrows); however, time-lapse videos showed stromules interacting with AFs, but not extending along AFs (*Video 6*).

To determine if actin plays another role in stromule dynamics, we performed time-lapse studies in epidermal cells expressing the actin marker Lifeact-TagRFP. Stromules appeared to interact statically, and not dynamically, with AFs, suggesting there are actin anchor points along stromules. Interactions were observed at the tips or at kink points (*Figure 6B*). Kymographs and time lapsed video show that retracting stromules paused for multiple, consecutive time frames or stopped completely at AFs (*Figure 6C*, *Figure 6—figure supplement 1*, *Video 7*). Due to the density of the AF network, stromules are often seen intersecting with AFs. To indirectly determine if those points of intersection are potential AF anchor points, we examined stromule retraction events. 19.4% of stromules retracted fully back to the body of the chloroplast without any pausing, often passing intersections with AFs. 77.1% of retracting stromule tips paused for multiple, consecutive frames and showed

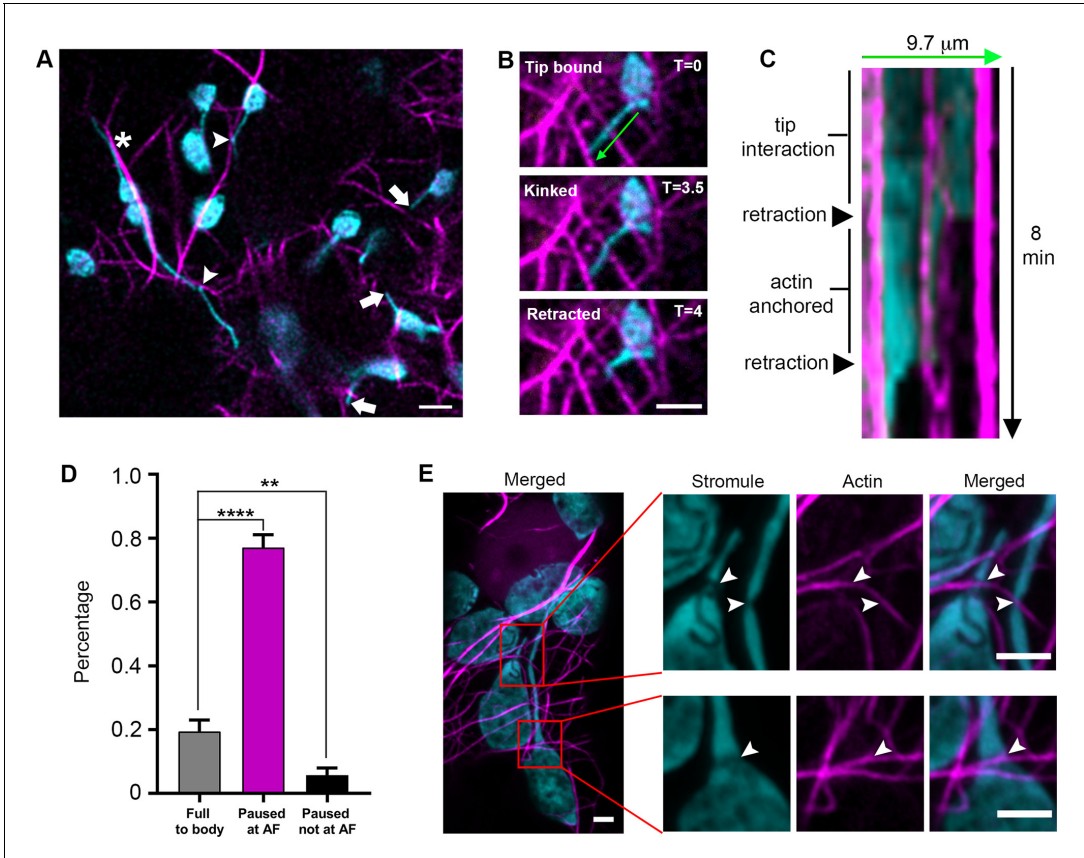

**Figure 6.** Stromule are anchored to the actin microfilament network. (**A**) AFs were marked by expressing Lifeact-TagRFP (magenta) in *N. benthamiana* transgenic lines expressing NRIP1-Cerulean that marks stromules (cyan). Stromules were seen interacting with AFs at kink points (arrow heads). Tips were commonly seen not associated with AFs (Arrows). Stromules were occasionally seen oriented along AFs but not overlapping with AFs (*). Scale bar equals 5 μm. (**B**) An extended stromule with a tip in close proximity to an AF, became kinked at a point overlapping with an AF near the midpoint of the stromule, and retracted back to the kink point interacting with the AF. (**C**) A kymograph was created along the stromule adjacent to the 9.7 μm green line in panel B over 8 min. The stromule tip that was in close proximity to an AF and then rapidly retracts to an actin anchor. It remained attached to the actin anchor point for an additional 4 min before retraction to the body of the chloroplast. (**D**) The percent of stromules pausing at AFs during retraction events was quantified. No pausing resulted in a full retraction back to the body of the chloroplast (grey bar). Stromule retractions that did not retract completely and paused for multiple time frames showed a correlation of the paused stromule tip with an AF (magenta bar) or no correlation with an AF (black bar). Data was collected from 22 biological replicates spanning eight different experimental replicates. Eighty-two retraction events were quantified from 30 different cells. Data represented as the mean +standard error of the mean (SEM), **p<0.001, ****p<0.00001 (Student's t test with Welch's correction). (**E**) AFs were marked by expressing Citrine-mTalin (magenta) in *N. benthamiana* transgenic lines expressing NRIP1-Cerulean that marks stromules (cyan). High-resolution airyscan confocal micrographs revealed thinning points of stromule or chloroplast interactions with AFs (arrowheads). Scale bar equals 2 μm.

DOI: https://doi.org/10.7554/eLife.23625.025

The following source data and figure supplements are available for figure 6:

**Source data 1.** Statistics of the quantification of stromules pausing at the AFs.
DOI: https://doi.org/10.7554/eLife.23625.030

**Figure supplement 1.** Chloroplast stromules statically interact with actin microfilaments.
DOI: https://doi.org/10.7554/eLife.23625.026

**Figure supplement 2.** Characterization of actin microfilaments associated with stromules and the body of chloroplasts.
DOI: https://doi.org/10.7554/eLife.23625.027

**Figure supplement 3.** Disruption of actin filaments does not affect stromule number.
DOI: https://doi.org/10.7554/eLife.23625.028

**Figure supplement 3—source data 1.** Statistics of the stromule frequency after CTD treatment.
DOI: https://doi.org/10.7554/eLife.23625.029

colocalization with an AF. 5.7% of retracting stromule tips paused for multiple, consecutive frames,

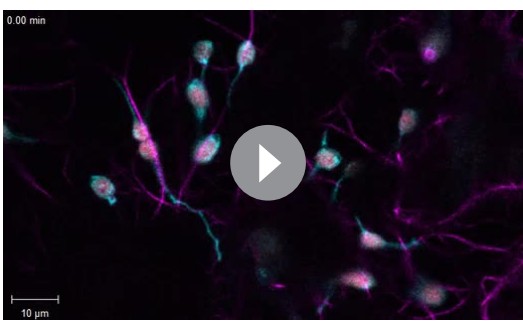

**Video 6.** Stromules do not extend along AFs. Stromules (cyan) interacted with AFs (magenta) at points along the stromule, but stromules did not directly extend along AFs. Sixty images were captured by time-lapse confocal microscopy every 18 s and displayed at 8 frames/s. The video was used to generate *Figure 6A*.
DOI: https://doi.org/10.7554/eLife.23625.031

but did not colocalize with AF (*Figure 6D*). The pausing of retracting stromules at AF cannot be explained by chance alone because the density of AFs and the colocalization of stromules with AFs observed appeared to be much less than 77.1% (*Figure 6A*; *Figure 6—figure supplement 1*; *Video 7*). Therefore, this data suggests that there are actin anchor points along stromules. We further examined the interaction of stromules and AFs by expressing mTalin-Citrine in NRIP1-Cerulean *N. benthamiana* plants and then examining by high-resolution airyscan confocal microscopy (*Figure 6E*). AF marker mTalin-Citrine has been used previously to detect chloroplast-associated actin (cp-actin) (*Kadota et al., 2009*). We could observe clear interactions of AF with chloroplast bodies (*Figure 6E*). Interestingly, we observed a clear thinning or constriction at the site of stromule-to-actin interaction points along the length and across the body of the chloroplast (*Figure 6E*; arrowheads). Three-dimensional modeling shows grooves across the body of the chloroplast that correlate with AFs (*Figure 6E*, *Figure 6—figure supplement 2C–D*). Although the mechanism of stromule thinning at actin interaction sites is unknown, collectively these results indicate that AFs provide anchor points for stromules but not tracks for stromule extension.

To determine the effect of AF disruption on stromule formation, we expressed mTalin-Citrine in NRIP1-Cerulean transgenic lines and applied 200 μM Cytochalasin D (CTD) to depolymerize AFs for 30 min (*Figure 6—figure supplement 3*). Since CTD treatment disrupted the actin network only in a fraction of the cells, only cells with a disrupted actin network were examined. Stromules were still present in cells where actin network was disrupted (*Figure 6—figure supplement 3A*) and the stromule number was similar between the CTD- and mock-treatments (*Figure 6—figure supplement 3B*).

## Microtubules and actin filaments contribute to stromule dynamics

Several studies suggested that MT and AF networks might work cooperatively for maintaining cell structure and physiology in eukaryotic systems (reviewed in [*Takeuchi et al., 2017*]). Although stromule formation and extension is primarily associated with MTs, AFs might have a role in stromule dynamics. To examine the role of each cytoskeletal filament, we treated transgenic *N. benthamiana* plants expressing GFP-TUA6 that marks MTs and FABD2-GFP that marks AFs with longer treatments of low concentrations of cytoskeleton inhibitors that specifically disrupt one cytoskeleton component, but not other. These experiments are in contrast to shorter treatments of higher concentrations (*Figure 3*, *Figure 6—figure supplement 3*) that are not optimal for time lapsed acquisition of stromule dynamics. We found that treatment with 1 μM of oryzalin (ORY) treatment for 1 hr partially disrupted MTs and had no significant, visible effect on the AF network (*Figure 7—figure supplement 1*, middle panels). Next, we tested CTD treatment concentrations to disrupt AFs. 10 μM

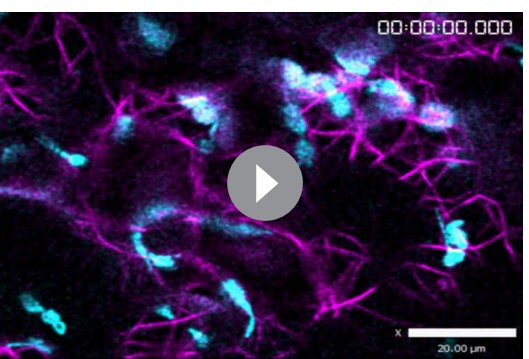

**Video 7.** Stromule retraction to actin anchor points. Two stromules (cyan) retracted to multiple actin anchor points (arrows) along AFs (magenta). Thirty-seven images were captured by time-lapse confocal microscopy every 18 s and displayed at 12 frames/s. The video was used to generate *Figure 6B–C*.
DOI: https://doi.org/10.7554/eLife.23625.032

of CTD treatment for 1 hr fully disrupted actin filament AF network showing bright puncta of GFP-FABD2, but only had a mild effect on MT organization (*Figure 7—figure supplement 1*, right panels); therefore, we used 10 μM of CTD. NRIP1-Cerulean transgenic plants were treated with either 10 μM CTD or 1 μM ORY for 1 hr to disrupt AFs or MTs, respectively, and then stromule length and dynamics were analyzed (*Figure 7A*). Interestingly, although average stromule length in both drug treatments were not significantly different (*Figure 7B*), velocity of stromule extension was increased significantly in ORY treatment compared to DMSO control (*Figure 7C*). Furthermore, CTD treatment resulted in significant reduction in velocity of both stromule extension and retraction compared to the control (*Figure 7C*). Interestingly, CTD treatment increased constant and smooth movements of stromules and reduced sudden and erratic movements of stromules, suggesting that CTD treatment stabilizes stromule dynamics (*Figure 7D*). Together, these results indicate that both types of cytoskeletal filaments regulate stromule dynamics.

## Stromules direct chloroplast movement

While analyzing time-lapsed images of stromule movements in *N. benthamiana* transgenic plants expressing NRIP1-Cerulean, we observed the movement of chloroplasts in the direction of stromules (*Figure 8A*; *Video 8*) or toward stromule kinks that are correlated with anchor points (*Figure 6*, *Figure 8—figure supplement 1*). This observation suggests that stromules might direct or guide chloroplast movement. To examine if this movement is correlated with the interactions with the cytoskeleton, we co-expressed Lifeact-TagRFP that marks AFs and NRIP1(cTP)-TagBFP that marks stromules in *N. benthamiana* transgenic plants expressing GFP-TUA6 that marks MTs (*Figure 8B*). Stromules were anchored to AF and connected to MT for extension at 0 min. Stromules extend along MT at 1 min and retracted to the actin anchor point at 3 min. Stromule reextend on MT at 8 min. Retraction of stromule at 9 min led to movement of chloroplast body toward the direction of the stromule movement (*Figure 8B*; *Video 9*).

Next, we investigated if the stromule angle and the angle of chloroplast movement are significantly correlated and changed by ORY or CTD treatment. Since CTD treatment resulted in a complete disruption of chloroplast movement (*Video 10*), it was not analyzed. Chloroplast movement was first identified as any movement larger than the radius of the chloroplast body and the direction of the movement was measured as the angle from the start and end points of each movement events. If a chloroplast changed direction, that was considered a separate movement event. Only chloroplasts containing one or more stromules were used for this analysis because it depends on comparing paired measurements of the angle of the stromule from the chloroplast body attachment point to the tip and the angle of chloroplast movement. We compared 33 pairs for DMSO control and 47 pairs for ORY to calculate a circular correlation coefficient, r(FL) (*Fisher and Lee, 1983*). An r (FL) value of 1.0 would indicate that the stromule angle and the angle of chloroplast movement are always identical, an r(FL) of −1.0 would mean the paired angles differ by 180 degrees, and if the angles are randomly matched the r(FL) will be close to zero. The r(FL) values for DMSO and ORY were 0.76 and 0.85, respectively. To test the statistical significance, each data set was randomly shuffled 10,000 times and the r(FL) calculated for each randomization; the observed f(FL) values were greater than all the randomized r(FL) values, so for both DMSO and ORY the stromule angle and the angle of chloroplast movement were significantly correlated (p<0.0001). Standard errors of the r(FL) values were calculated using the jackknife method (*Sokal and Rohlf, 1995*), and used in a two-sample t-test; the r(FL) values for DMSO and ORY were not significantly different from each other (p=0.52). We generated a scatter plot of chloroplast movement angles and stromule angles, which shows a fairly linear relationship compared to the randomized control (*Figure 8—figure supplement 1*). To further visualize these data, we calculated the difference between the two angles and plotted the frequency (*Figure 8C*). If the chloroplast movement angle and stromule angle are equal, then the difference will be zero. We observed a higher frequency around zero compared to the randomized control. An examination of only angle pairs with less than ±30 degree difference were highly correlated and had an r(FL) value of 0.95; we therefore defined a stromule-directed movement event as being within ±30 degrees. The circular correlation calculation requires paired stromule angles and chloroplast movement angles, and excludes chloroplasts that move but do not have stromules. Using the ±30 degree criteria for stromule directed movement, we were able to compare the percent of stromule driven movement compared to total movement events, which includes chloroplasts without stromules This analysis shows that ORY treatment decreased stromule-directed chloroplast

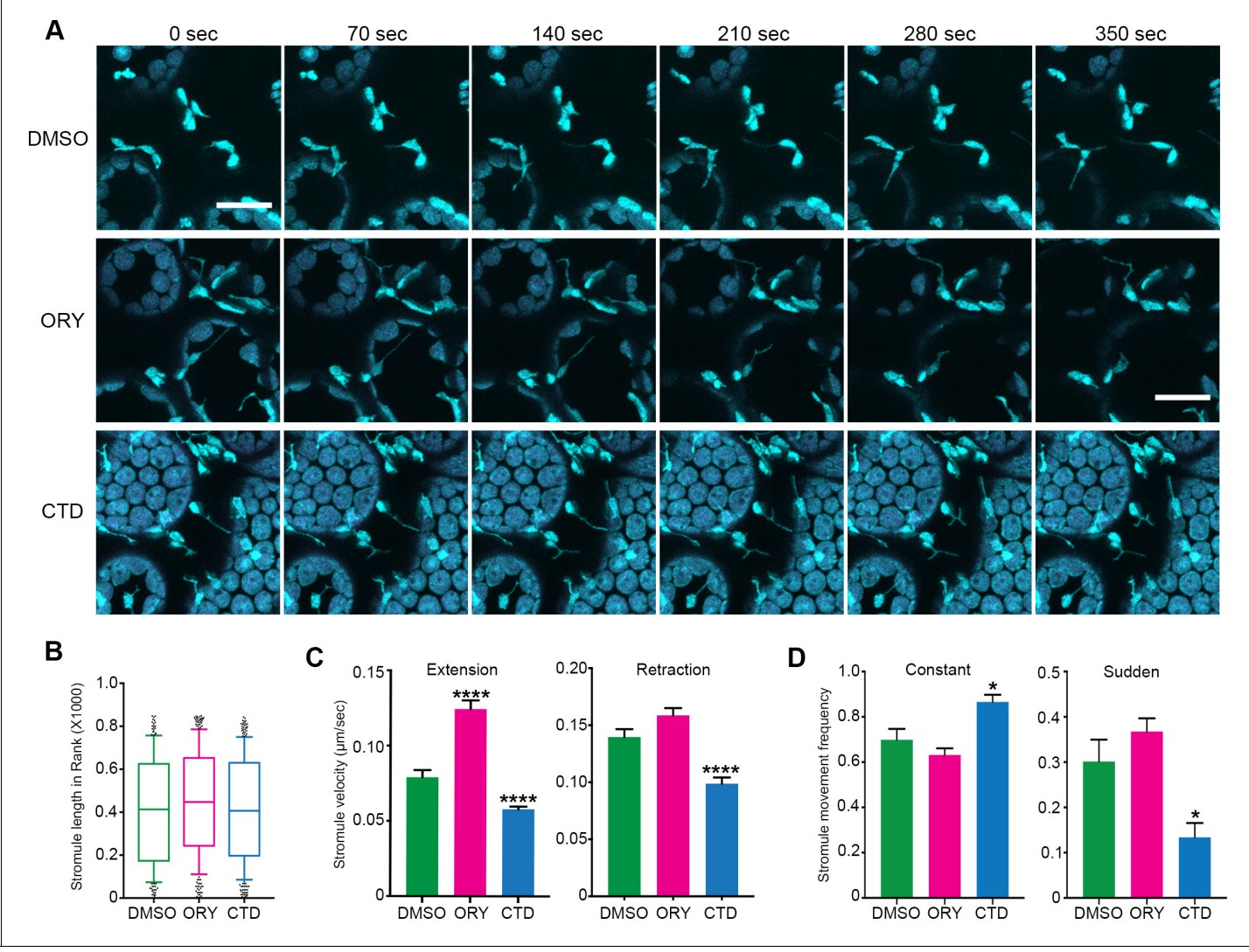

**Figure 7.** Disruption of cytoskeleton change stromule dynamics. (**A**) Time lapsed images of stromules 1 hr post-treatment with 1 µM of MT inhibitor oryzalin (ORY) or with 1 µM of actin inhibitor cytochalasin D (CTD) on the leaf of the NRIP1-Cerulean transgenic *N. benthamiana*. At these concentrations, ORY disrupt MT organization slightly while no visible effect on actin cytoskeleton. On the other hand, CTD showed significant disruption of the actin filament while no significant effect on MT organization (see *Figure 7—figure supplement 1*). The experiments were repeated three times with six replicates per treatment. Scale bar equals 20 µm. (**B**)-(**D**) Quantification of stromule dynamics in A. (**B**) Stromules length did not change significantly upon inhibitor treatments. (**C**) ORY treatment increased stromule extension velocity (magenta bars compared to green bars), while CTD treatment reduced the velocity of both stromule extension and retraction (blue bars compare to green bars). Data represented as the mean +standard error of the mean (SEM), ****$p < 0.0001$ (Dunn's multiple comparison test). (**D**) The frequency of constant, smoothly extending stromules was increased (left panel) and the frequency of sudden extending stromules decreased (right panel) with CTD treatment. ORY treatment showed no significant difference. Data represented as the mean +standard error of the mean (SEM), *$p < 0.05$, (Mann-Whitney test).

DOI: https://doi.org/10.7554/eLife.23625.033

The following source data and figure supplement are available for figure 7:

**Source data 1.** Statistics of the stromule dynamics.

DOI: https://doi.org/10.7554/eLife.23625.035

**Figure supplement 1.** Effects of inhibitor treatments on the disruption of cytoskeleton.

DOI: https://doi.org/10.7554/eLife.23625.034

movement and CTD disrupted nearly all chloroplast movement, including stromule-directed (*Figure 8D*). It is possible that stromule extension or retraction may provide the driving force for stromule-directed movement. Therefore, we quantified how many times stromule extension and retraction events occur in 10 mins after 1 hr of drug treatment. Interestingly, ORY treatment

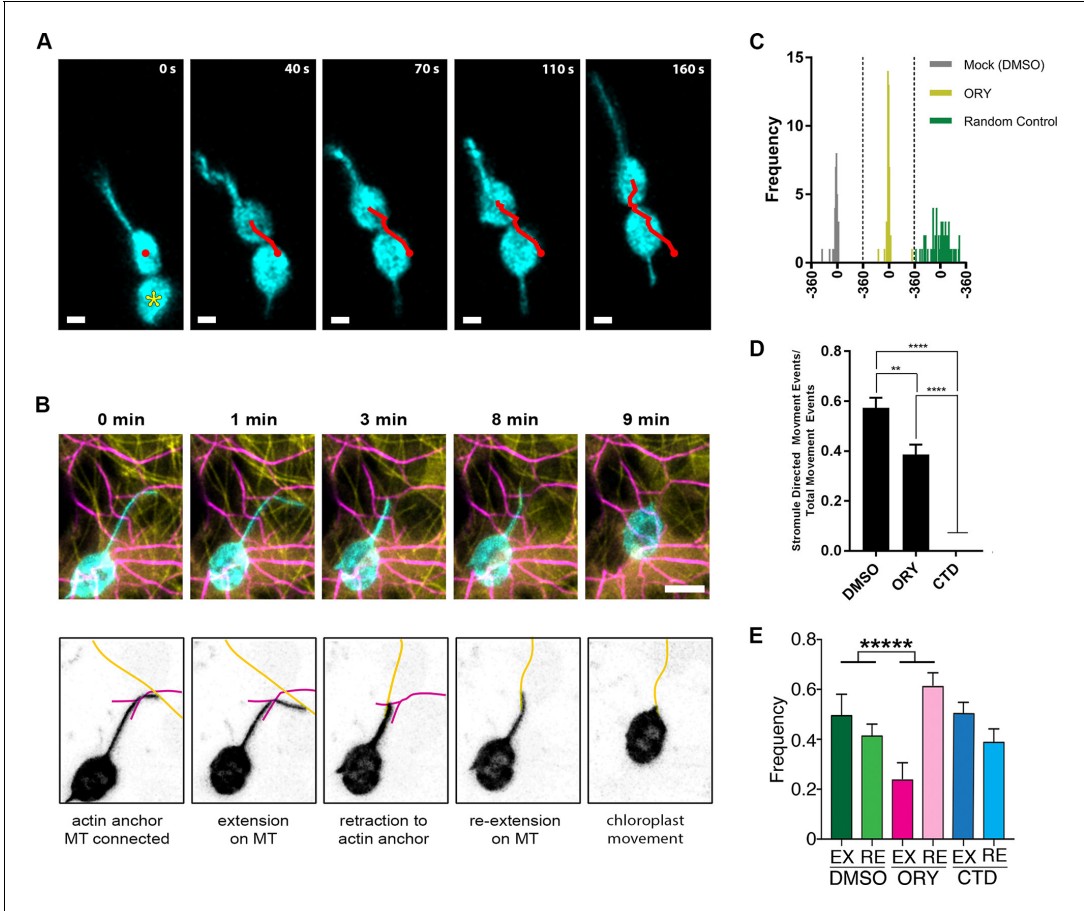

**Figure 8.** Stromule directed chloroplast movement. (**A**) Stromules and chloroplast movement events were imaged in NRIP1-Cerulean *N. benthamiana* transgenic plants. A chloroplast was tracked (red line) in time lapsed images. The direction of movement was correlated with the direction of the stromule. A connected chloroplast (asterisk) moved with the stromule directed chloroplast. Time is measured in seconds and scale bars equal 2 µm. (**B**) AFs and stromules were marked by transiently expressing Lifeact-TagRFP (magenta) and NRIP1(cTP)-TagBFP (cyan) in *N. benthamiana* transgenic plants expressing GFP-TUA6 that marks MTs (yellow). The top row shows the merged images and the bottom row is an illustration highlighting the MT and actin dynamic events. At 0 min, the stromule tip is bound to a MT and a branch point is bound to an AF. At 1 min, the stromule extended along the MT. At 3 min, the stromule retracted to an actin anchor point. At 8 min, the stromule re-extended along a MT. At 9 min, the stromule retracted and correlated with chloroplast movement. Scale bar equals 5 µm. (**C**) The direction of the stromule connected to the chloroplast body and the direction of chloroplast movement were measured in FIJI ImageJ. The difference in angle was calculated and plotted. Both Mock and Oryzalin (ORY), showed a high frequency of values close to 0. Randomly generated values were used as a control. (**D**) The percent of chloroplast movement that were stromule directed movements were quantified. Cytochalasin D (CTD) treatment resulted in a complete halt of movement. Oryzalin (ORY) treatment caused a decrease in stromule directed movement compared to the DMSO vehicle control. Data represented as the ±SEM, ****$p < 0.0001$, **$p < 0.01$ (one-way ANOVA). (**E**) Stromules retract more frequently (ORY; pink bar) and extend less frequently (ORY; magenta bar) with oryzalin treatment. Data represented as the mean +standard deviation (SD), *****$p < 0.00001$ by Mann-Whitney test.

DOI: https://doi.org/10.7554/eLife.23625.036

The following source data and figure supplements are available for figure 8:

**Source data 1.** Statistical analyses of the stromule-directed movements.
DOI: https://doi.org/10.7554/eLife.23625.040
**Figure supplement 1.** Stromule-directed movement.
DOI: https://doi.org/10.7554/eLife.23625.037
**Figure supplement 1—source data 1.** Data of the stromule-directed movement.
DOI: https://doi.org/10.7554/eLife.23625.038
**Figure supplement 2.** Stromule-directed chloroplast movement during partial actin disruption.
DOI: https://doi.org/10.7554/eLife.23625.039

significantly increased retractions and reduced the number of extensions (*Figure 8E*); however, the

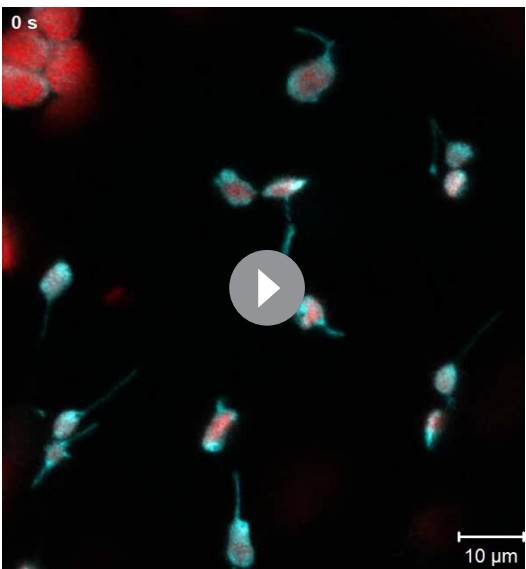

**Video 8.** Stromule-directed movement. Chloroplast bodies and stromules were visualized in a NRIP1-CFP transgenic N. benthamiana plants, in combination with chlorophyll autofluorescence. Time-lapse confocal microscopy was used to acquire images every 10 s for 600 s, displayed at 8 framess. In upper right corner, an instance of stromule-directed movement can be observed, which was used to generate *Figure 8A*. The chloroplast located in the bottom-center of the video, shows another instance of stromule-directed movement.

DOI: https://doi.org/10.7554/eLife.23625.041

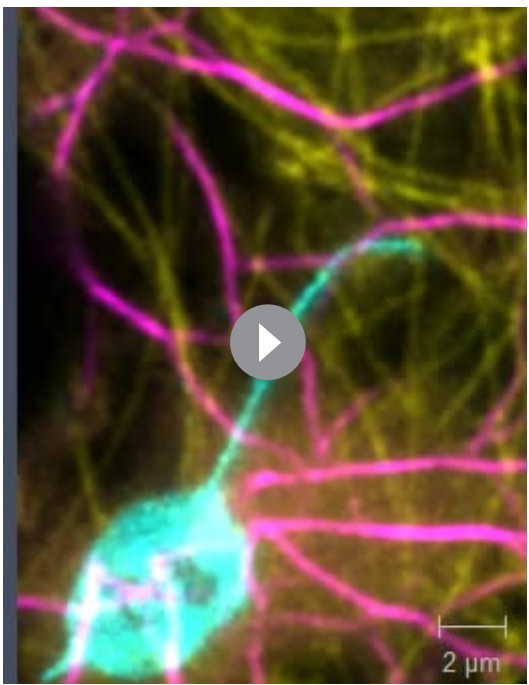

**Video 9.** Dynamic control of stromules by extension along MTs and anchoring to AFs. AFs and stromules were marked by transiently expressing Lifeact-TagRFP (magenta) and NRIP1(cTP)-TagBFP (cyan) in *N. benthamiana* transgenic lines expressing GFP-TUA6 that marks MTs (yellow). Sixteen maximum intensity projections of time-lapse z-stacks taken were taken every 24.5 s and displayed at 4 frames/s. The video was used to generate *Figure 8B*.

DOI: https://doi.org/10.7554/eLife.23625.042

remaining stromules extension showed a higher velocity (*Figure 7C*) suggesting the frequency rather than the velocity of stromule extension is with correlated chloroplast movement. Overall, these results suggest that ORY treatment caused the reduced stromule-directed chloroplast movement due to less extension events. Our data show that stromules may direct chloroplast movement in epidermal pavement cells; however, it remains unknown if stromules provide a driving force or only guide chloroplast movement.

The longer CTD treatment resulted in a complete disruption of AFs and nearly all chloroplast movement. Since chloroplasts are anchored to the AF network, we aimed to partially disrupt the AF network without fully abrogating all AF function. Treatment with CTD resulted in discontinuous AFs (*Figure 8—figure supplement 2A*, magenta) while the MTs were intact (*Figure 8—figure supplement 2A*, yellow). Examination of time lapsed maximum intensity projections of confocal micrographs showed that stromules were still present at 3 min and then briefly absent at approximately 8 min after CTD treatment

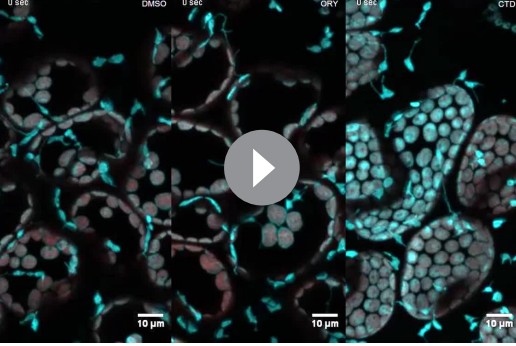

**Video 10.** Chloroplasts and stromule dynamics following ORY and CTD treatments. Chloroplast bodies and stromules were visualized in a NRIP1-CFP transgenic *N. benthamiana* plants, in combination with chlorophyll autofluorescence. Time-lapse confocal microscopy was used to acquire images every 10 s for 350 s, displayed at 16 frames/s. Displays chloroplast and stromule activity within plants subjected to either DMSO vehicle control (left), 1 µm of ORY (center), or 1 µm of CTD (right) drug treatments.

DOI: https://doi.org/10.7554/eLife.23625.043

(*Figure 8—figure supplement 2A*, *Video 11*). This brief disruption further supports that stromules are stabilized by AF anchors and disruption of AFs results in rapid retraction of stromules. However, despite the initial disruption, stromules re-extended along MTs and multiple stromules were observed after 20 min (*Figure 8—figure supplement 2A*). These observations explain why the disruption of stromules by CTD was missed during 30-min treatment (*Figure 6—figure supplement 3*). Tracking the stromule and chloroplast movement (*Lu et al., 2017*) showed that stromules can still direct chloroplast movement if AFs are only partially disrupted. One chloroplast (Cp1) had restricted movement and colocalized with an AF fragment (*Figure 8—figure supplement 2A*). Stromules were observed extending in opposite directions (*Figure 8—figure supplement 2B*). However, the second chloroplast (Cp2) did not co-localize with AF fragments (*Figure 8—figure supplement 2A*; *Video 11*). The stromule of this chloroplast not only extended, but its extension along the MTs facilitated a rapid pulling of the body of down the viewing plane (*Figure 8—figure supplement 2C*).

## Actin microfilaments mediate perinuclear chloroplast clustering during plant immune response

Our previous findings indicate that N NLR immune receptor-triggered immunity to the TMV p50 effector resulted in stromule induction, stromule-to-nuclear connections and eventual perinuclear clustering of chloroplasts (*Caplan et al., 2015*). Electron microscopy results in our previous studies indicated that the chloroplast and nuclear membranes do not directly interact (*Caplan et al., 2015*), suggesting other cytoplasmic components are required for this interaction. To study the importance of cytoskeleton during the process of perinuclear chloroplast clustering, we expressed TMV-p50 to induce an immune response in N-containing NRIP1-Cerulean *N. benthamiana* transgenic plants (*Caplan et al., 2015*; *Caplan et al., 2008*). Since stromules extend along MTs, initially, we marked MTs and looked at stromules to nuclear connections, but we were unable to find significant connections of stromules to MTs around nuclei. Therefore, we next marked AFs with Lifeact-TagRFP and found connections between stromules and AFs surrounding nuclei (*Figure 9*). Time-lapse studies showed long stromules stably connecting to an AF attached to a nucleus for approximately 18 min (*Figure 9A*, arrowheads). After 18 min of continuous imaging, a long stromule retracted, bringing the chloroplast body close to the nucleus (*Figure 9A*, arrows; *Video 12*). We verified these results with another AF marker, mTalin-Citrine (*Figure 9—figure supplement 1*). We also observed that when the bodies of chloroplasts were in contact with nuclei, there were connections with AFs (*Figure 9B–C*, arrows).

Since p50-induced immunity leads to vigorous stromule induction (*Caplan et al., 2015*; *Figure 5*), we hypothesized that more chloroplasts might move toward nucleus by stromule-directed movement of chloroplast body. Therefore, we quantified the perinuclear chloroplast clustering during TMV-p50-induced immune response in N-containing NRIP1-Cerulean transgenic plants in a time course (*Figure 10A–B*). Although majority of nuclei had a low number of interacting chloroplasts in the control (*Figure 10A*, left panels), we observed a significantly higher number of chloroplasts around nuclei in TMV-p50-treated samples (*Figure 10A*, right panels). More than 80% of nuclei (85 out of 105) were surrounded by more than two chloroplasts in TMV-p50-treated samples compared to 50% of observed nuclei (56 out of 120) were surrounded by none or single chloroplast in the control (*Figure 10A* and *Figure 10—figure supplement 1A*). The ratio of nuclei-clustered with more than four chloroplasts was significantly higher in TMV-p50 treatment compared to the control (*Figure 10B* and *Figure 10—figure supplement 1A*). These results indicate significant induction of perinuclear chloroplast clustering during an immune response.

To determine, if AF anchoring plays a role in the immunity-induced perinuclear clustering of chloroplasts, we treated plants with CTD and

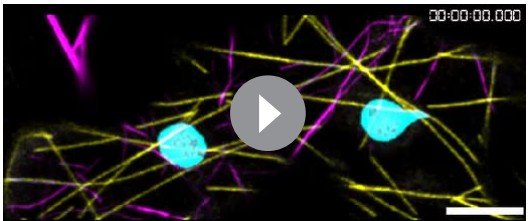

**Video 11.** Stromule directed movement in Cytochalasin-D-treated cell. Stromules (cyan) from two chloroplasts are disrupted after Cytochalasin D treatment that fragmented AFs (magenta). Stromules re-extended along MTs (yellow) resulting in the movement of chloroplasts. 105 maximum intensity projections of time-lapse z-stacks were taken every 39 s and displayed at eight frames per second. The video was used to generate *Figure 8—figure supplement 2*.
DOI: https://doi.org/10.7554/eLife.23625.044

ORY. Remarkably, CTD treatment significantly reduced the number of chloroplasts interacting with nuclei compared to the control and ORY treatment (*Figure 10C–D* and *Figure 10—figure supplement 1B*). These results support that anchoring of stromules to the AFs at the nucleus or more generally chloroplast movement is important for perinuclear clustering of chloroplasts during plant immune response. In conclusion, we propose a model in which perinuclear clustering of chloroplasts involves stromule anchoring to AFs surrounding nuclei and stromules guide chloroplasts toward nuclei during an immune response.

## Discussion

Cytoskeletal elements in plant cells support several cellular functions, including cytoplasmic streaming, cell division, cell elongation, polar growth, vesicle trafficking, nuclear positioning and morphogenesis (*Cai et al., 2015*; *Higa et al., 2014*; *Li et al., 2015*). In this study, we show that dynamic stromules extend along MTs and AFs stabilize stromules and chloroplast-to-nuclear connections during innate immune response. Stromules have the ability to direct chloroplast movement, and AF anchoring of stromules may guide perinuclear chloroplast clustering during innate immunity.

Previous studies in non-green hypocotyls indicated a role for AFs and MTs during stromule formation (*Kwok and Hanson, 2003*, *2004a*). The initial study (*Kwok and Hanson, 2003*) used cytoskeletal inhibitors to implicate AFs and MTs during stromule formation, suggesting that AFs promote while MTs restrict stromule and plastid movement. Stromules visualized by differential interference contrast were observed interacting directly with AFs labeled with GFP-hTalin and rearrangements of the AF network changed stromule morphology (*Kwok and Hanson, 2004a*). Movement along AFs was indirectly implicated by the identification of myosin XI cargo domain and a small tail domain that localize to chloroplasts (*Natesan et al., 2009*; *Sattarzadeh et al., 2009*). Knockdown of myosin XIs qualitatively disrupted stromules (*Sattarzadeh et al., 2009*) or quantitatively decreased the percent of plastids with stromules (*Natesan et al., 2009*). However, the dynamics of stromules moving along AFs were not examined in these studies (*Natesan et al., 2009*; *Sattarzadeh et al., 2009*). Furthermore, longer myosin XI tail domain (*Reisen and Hanson, 2007*) and full-length myosin XI (*Avisar et al., 2008*) do not localize to the chloroplasts. These studies prompted us to conduct a detailed time-lapse confocal microscopy of the dynamics of stromules and AFs. Surprisingly, our extensive investigations were unable to show extension of stromules along AFs. Instead, we discovered stromules were statically anchored to the AF network. Stromules were previously shown to actively move beyond AF attachment points via an unknown mechanism that was proposed to be either collisions with other components of the cytoplasm or interactions with very fine AFs (*Kwok and Hanson, 2004a*). Here, we have revealed that this unknown mechanism to be stromule extension along MTs by simultaneously monitoring MTs labeled with GFP-TUA6, AFs labeled with Lifeact-TagRFP and stromules labeled with NRIP1(cTP)-BFP via time-lapse confocal microscopy (*Figure 8B*). When our third revision version of the manuscript was under review, another report has proposed a model in which both stromule extension and slow anchoring occurs on MTs and rapid extension occurs on AFs (*Erickson et al., 2017b*). Our high-resolution imaging data clearly indicate that stromules do not extend along AF. Our data shows that static stromule anchoring is associated with the AF network and dynamic movement occurs along MTs. We used both Lifeact and mTalin, since they may label different pools of AFs. Lifeact results in even labeling of fine AF network and is accepted as one of the best markers for AF (*Riedl et al., 2008*). However, mTalin was used previously for labeling cp–actin interacting with chloroplasts and required for blue-light mediated movement (*Kadota et al., 2009*). We have found that each marker has its own advantages and disadvantages, and no single marker is perfect. The actin inhibitor CTD disrupted the actin network, briefly destabilizing stromules which then dynamically re-extended along MTs. It is possible that myosin XI silencing is causing a similar effect, since knockout of myosin XI in *Arabidopsis* resulted in inhibiting distribution and dynamics of actin network (*Cai et al., 2014*; *Park and Nebenführ, 2013*).

Early studies examining the role of MTs during stromule formation were conducted with MT inhibitors, APM or Oryzalin, leading to the conclusion that MTs have a limited role, because disruption resulted in either a 25% reduction in stromule length in hypocotyls treated with 5 µM of APM (*Kwok and Hanson, 2003*) or no alteration of stromules in *Nicotiana* leaves treated with 36 µM of oryzalin (*Natesan et al., 2009*). A recent study also shows that stromules remained extended after 100 µM Oryzalin treatment (*Erickson et al., 2017b*). We show that both 20 µM APM and 300 µM

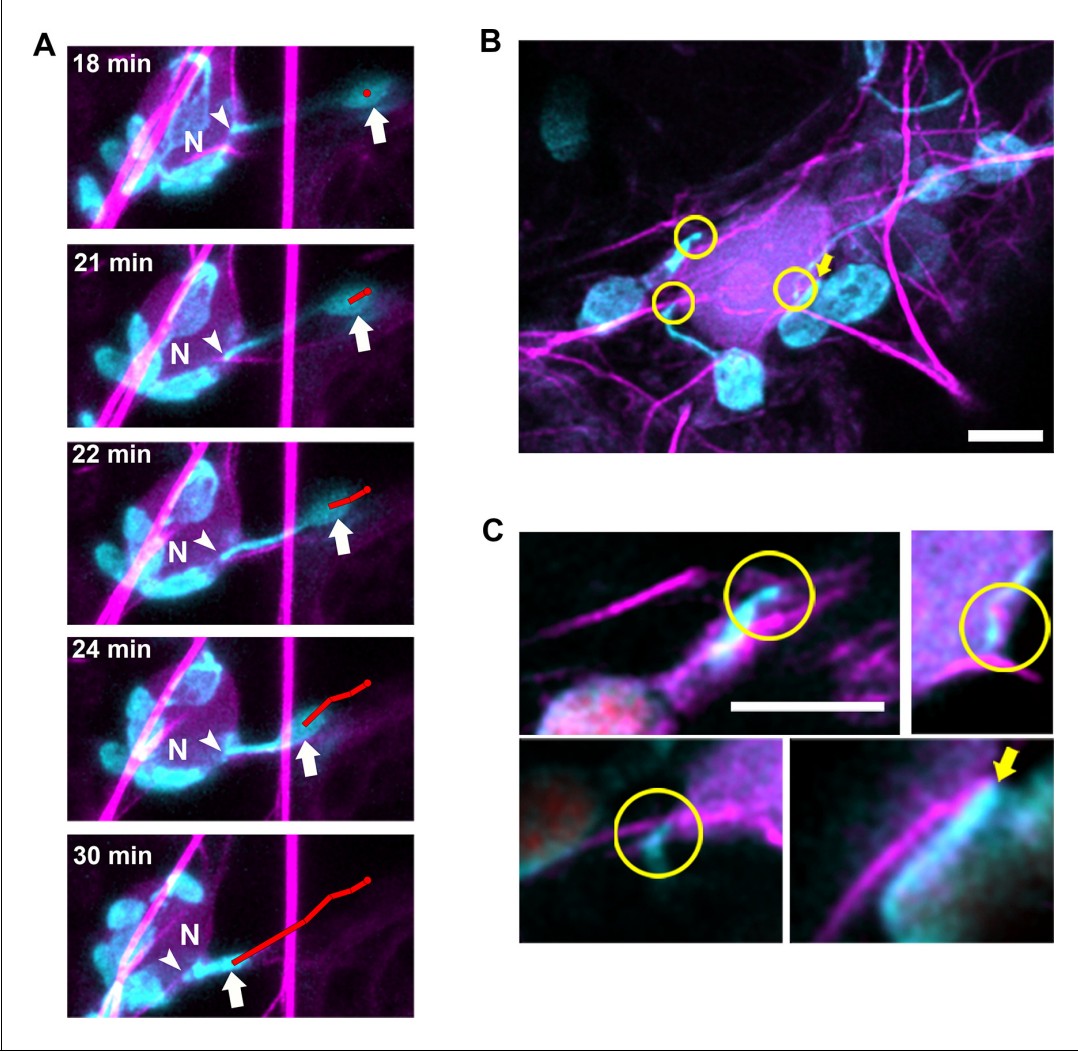

**Figure 9.** Chloroplasts and stromules positioning during perinuclear chloroplast clustering during immune response. (A) TMV-p50 effector and Lifeact-TagRFP (magenta; AFs) were expressed in transgenic NRIP1-Cerulean *N. benthamiana* plants that marks stromules (cyan). Time lapse images of stromule retraction toward the nucleus (N) after chloroplast positioning around the nucleus. A stromule tip (arrow head) remained stably associated with an AF associated with a nucleus and the body of the chloroplast was anchored away from the nucleus for 18 min. Stromule retraction from 18 to 30 min brought the chloroplast body (arrow) in close association to the nucleus. The body of the chloroplast was tracked (red line). Arrows indicate retracting stromule. Scale bar equals 5 μm. (B) TMV-p50 effector and Lifeact-TagRFP (magenta; AFs) were expressed in transgenic NRIP1-Cerulean *N. benthamiana* plants and then fixed as described previously (*Caplan et al., 2015*). Three interaction points of stromules with actin surrounding nuclei were detected (circles). The body of a chloroplast was also associated with perinuclear AFs (arrow). Image is a deconvolved maximum intensity projection of a confocal microscopy z-stack. (C) Enlargements of individual xy slices of the z-stack show connections of stromule tips (left) and a stromule kink point (top right) with AFs.

DOI: https://doi.org/10.7554/eLife.23625.045

The following figure supplement is available for figure 9:

**Figure supplement 1.** Stromule association with perinuclear actin microfilaments.

DOI: https://doi.org/10.7554/eLife.23625.046

Oryzalin can disrupt stromules in *Nicotiana* leaves; and propose that the difference between these studies may be caused by differences in either cell type or inhibitor concentration. The plastids in epidermal pavement cells in *Nicotiana* are chloroplasts (*Barton et al., 2017*), compared to dark grown hypocotyls that lack chlorophyll-containing plastids (*Kwok and Hanson, 2003*). In general, the formation of stromules may vary based on differences in cell, plastid, or stimulus type. We found that 20 μM of APM or 300 μM of oryzalin MT inhibitor was required to observe a more complete disruption of MTs. By monitoring MTs and stromules with fluorescently tagged markers, we were able

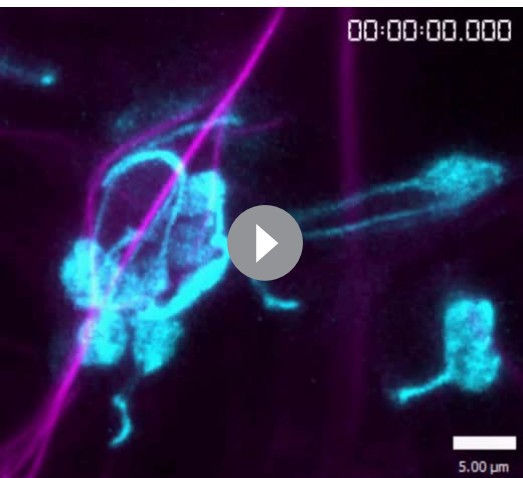

**Video 12.** Stromule retraction resulted in the movement of a chloroplast to a nucleus. A stromule (cyan) attached to an AF (magenta) pulls the chloroplast body to the nucleus by retracting. 100 maximum intensity projections of time-lapse z-stacks were taken every 38 s and displayed at 8 frames/s. The video was used to generate *Figure 9A*.
DOI: https://doi.org/10.7554/eLife.23625.047

to directly observe the effect of inhibitors on MT formation and found that stromules maintained interactions with small fragments of MTs, but retracted after complete disruption of MTs. The study using 100 µM Oryzalin also observed that stromules would remain associated to small fragments of MT (*Erickson et al., 2017b*). Although they did not quantify changes in stromule frequency or dynamics like we describe here, they qualitatively observed more fast moving, short-lived stromules. This is consistent with the overall increased stromule extension velocity that we measured after 1 µM Oryzalin treatment. The role of MTs during stromule extension is also supported by time-lapse confocal microscopy that shows stromule tips interacted and dynamically extended along MTs. This is consistent with the recent study also showing stromule extension along MTs using mOrange2-MAP4 (*Erickson et al., 2017b*). The involvement of MTs was unexpected because of the previously implicated role of AFs; therefore, we repeated these experiments with three independent MT markers, GFP-TUA6, EB1-Citrine, and TagRFP-MAP-CKL6.

To rule out potentially indirect effects of MT drugs, we further examined the mechanistic function of MTs during stromule formation by stabilizing MTs either chemically or genetically. Taxol, which stabilize MTs (*Schiff and Horwitz, 1980*), doubled the average number of stromules per chloroplasts. Furthermore, we altered MTs genetically by silencing *NbGCP4*. The γ-tubulin forms a complex with γ-tubulin complex protein (GCP) such as GCP2-GCP4 to form γ-tubulin ring complex (γ-TuRC) that plays an important role in MT nucleation and organization (*Moritz and Agard, 2001*). GCP4-GCP6 subunits are not essential for γ-tubulin complex (*Vinh et al., 2002*), but these subunits are important for stabilizing the ring complex (*Guillet et al., 2011*). Knockdown of GCP4 in *Arabidopsis* leaf pavement cells resulted in hyper-parallel bundles of MT (*Kong et al., 2010*). Our results showed that *NbGCP4* silencing in *N. benthamiana* leaves exhibited similar changes in MT organization via SOAX analysis. The change in MT structure induced by *NbGCP4* silencing was sufficient to induce stromules constitutively. Increased stromule length in *NbGCP4*-silenced plants could be due to less dynamic stromules, since extension and retraction velocities decreased. It is possible that the decrease in stromule dynamicity is caused by a disrupted balance between MT branching and MT bundling in *NbGCP4* silenced plants. These findings support that MT dynamics are a key regulator of stromule formation and dynamics.

It has been proposed that stromules may extend via an internal force, and not along MTs or AFs. Early studies have found filament-like structures in plastids and stromules, which potentially could provide an outward force (*Bourett et al., 1999*; *Lawrence and Possingham, 1984*). Recently, stromules were shown to form in vitro from isolated chloroplasts (*Brunkard et al., 2015*); but, clean chloroplast preparations resulted in only 1.1% of chloroplasts having short, spontaneous stromules and a 40-fold increase in stromules after the addition of cell extracts (*Ho and Theg, 2016*). We also have observed rapidly moving beak-like or small protrusions (*Video 2*, red dots) that do not interact with MTs. They also resemble 'chloroplast protrusions' observed in alpine plants that form independently of MTs (*Buchner et al., 2007*; *Holzinger et al., 2007a*; *Moser et al., 2015*). It was recently proposed that the small, fast moving stromules moved along actin because their rate of extension was similar to myosin motors (*Erickson et al., 2017b*). However, when AFs were marked with Lifeact-TagRFP, we did not observe a correlation of these stromules extending along AFs (N = 73). These studies combined suggest that there may be an alternative mechanism for stromule initiation that may depend an internal force. Another alternative to cytoskeleton driven stromule formation is force derived from membrane contact sites (MCS) with the ER (*Schattat et al., 2011*). Stromule and

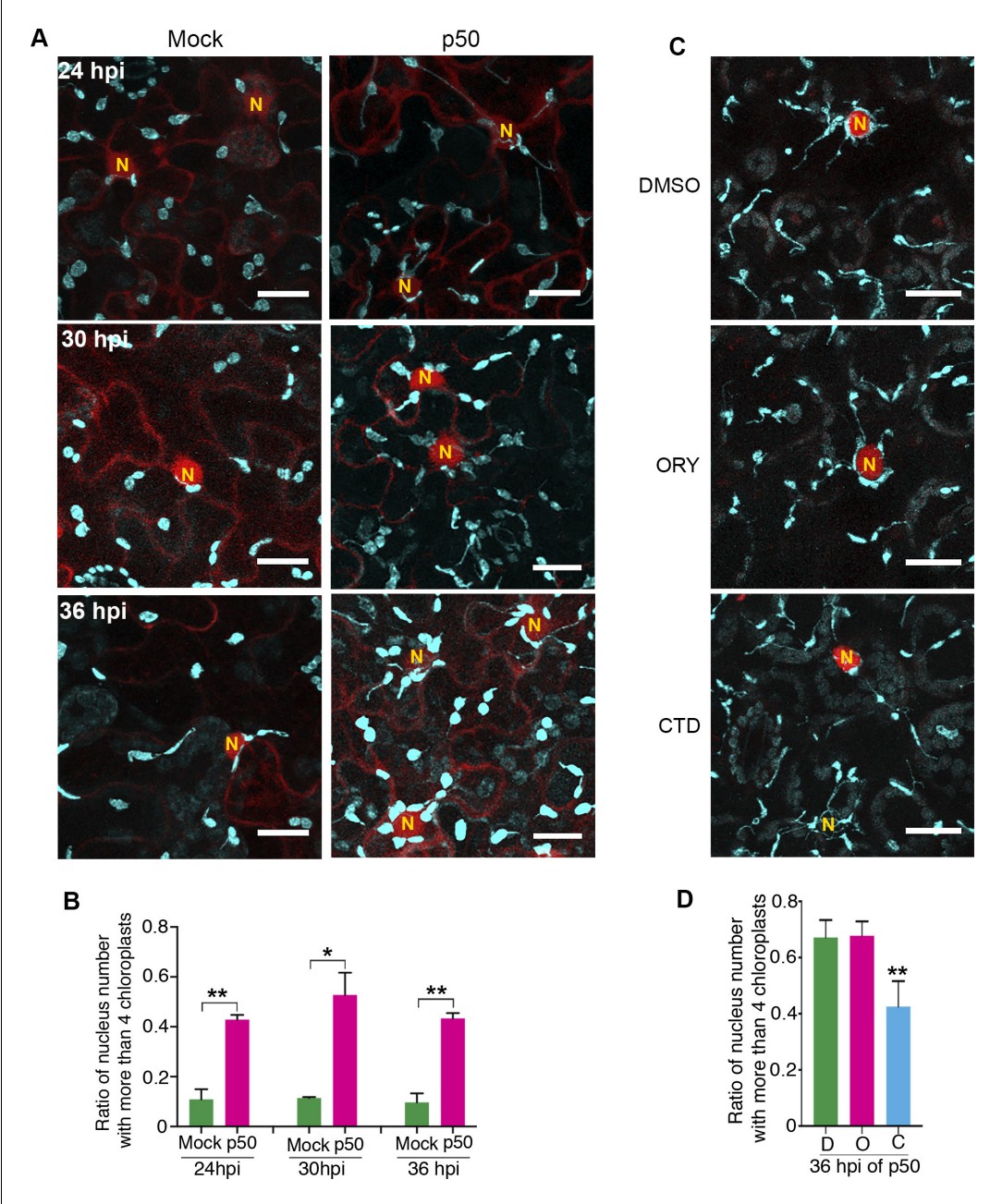

**Figure 10.** Perinuclear chloroplast clustering during immune response. (**A**) Time course images of perinuclear chloroplasts during TMV-p50 effector induced immune responses compared to mock control. Chloroplasts and stromules marked by NRIP1-Cerulean were pseudocolored cyan while nuclei and cytoplasm were pseudocolored red. N, nucleus. Scale bar equals 20 μm. (**B**) Quantification of perinuclear chloroplasts during TMV-p50 effector induced immune response compared to mock control shown in A. Ratio of nuclei associated with more than four chloroplasts in TMV-p50 infiltrated cells (magenta bars) are compared to those in control cells (green bars). More than 92 nuclei were observed for each condition from 48 images. Experiments were repeated three times with four plants each. Data represented as the mean +standard deviation (SD), **p<0.001, *p<0.01 (Student's t test with Welch's correction). (**C**) Actin cytoskeleton disruption by treatment with cytochalasin D (CTD) lead to dissociation of chloroplasts near the nucleus while microtubule disruption by treatment with oryzalin (ORY) does not affect the chloroplast positioning. Perinuclear clustering of chloroplasts was induced by TMV-p50 effector. Images were acquired 36 hr after induction. Cytoskeleton Inhibitors were treated 35 hr after induction. Chloroplasts and stromules presented by NRIP1-Cerulean were pseudocolored cyan, while nuclei were pseudocolored red. N, nucleus. Scale bar equals 20 μm. (**D**) Quantification of cells associated with more than four chloroplasts in C. More than 60 nuclei were observed for each condition from 12 images. Experiments were repeated three times with two plants each (total 6 plants). Data represented as the mean +standard deviation (SD), **p<0.001 (Student's t test with Welch's correction). D, DMSO; O, oryzalin; C, cytochalasin D.

DOI: https://doi.org/10.7554/eLife.23625.048

*Figure 10 continued on next page*

*Figure 10 continued*

The following source data and figure supplements are available for figure 10:

**Source data 1.** Statistics of the perinuclear clustering of the chloroplasts in TMV-p50 induced plant immune response and effect of cytoskeleton disrupting drugs.

DOI: https://doi.org/10.7554/eLife.23625.051

**Figure supplement 1.** Perinuclear clustering of chloroplasts in TMV-p50 effector induced immune response.

DOI: https://doi.org/10.7554/eLife.23625.049

**Figure supplement 1—source data 1.** Data of the chloroplast clustering to nucleus in TMV-p50 induced immune response.

DOI: https://doi.org/10.7554/eLife.23625.050

ER dynamics are correlated and it is possible that MCS stabilize stromules similar to actin anchors. They propose a model in which ER MCSs or the underlying cytoskeleton dictate stromule dynamics (*Schattat et al., 2011*). Our findings strongly supports a role for the cytoskeleton in which stromules require a combination of MT and AF interactions. Nonetheless, the function of the MCS between stromules and ER is intriguing and may assist in transfer of proteins, lipids or small molecules.

Blue-light-induced-chloroplast movement in plants is driven by chloroplast actin filaments (cp-actin) (*Kadota et al., 2009*). The chloroplast unusual positioning 1 (CHUP1) protein recruits actin to the leading edge of chloroplasts and is required for movement. Interestingly, the N-terminal coiled-coiled domain of CHUP1 is also required to anchor chloroplasts to the plasma membrane, revealing a complex, dual role of actin during chloroplast movement and anchoring (*Oikawa et al., 2008*). Chloroplasts are held by a cage of AFs (*Kandasamy and Meagher, 1999*) and additional recruitment of cp-actin via CHUP1 potentially may inhibit stromules by forming a physical constraint. Consistent with this hypothesis, CTD treatment disrupts actin around chloroplasts causing them to lose their ellipsoid shape to become round (*Figure 8—figure supplement 2*). Once released, chloroplasts moved in the direction of stromules extending along MTs. However, complete disruption of actin using longer treatments of CTD resulted in a complete disruption of all chloroplast movement, including stromule-directed movement. Stromule-guided movement was also seen without CTD treatment (*Figure 8*) and appears to be a novel type of organellar movement along MTs. Over 50% of all the chloroplast movement in steady-state epidermal pavement cells was stromule-directed, suggesting that this type of movement may significantly contribute to chloroplast movement and positioning. Interestingly, stromules in the green algae, *Acetabularia*, also have been implicated in chloroplast movement (*Menzel, 1994*), suggesting that both cp-actin and stromule-directed chloroplast movement are conserved between land plants and green algae (*Suetsugu and Wada, 2016*).

We have recently shown that stromules play an important role during innate immunity and programmed cell death (*Caplan et al., 2015*). During an immune response, chloroplasts move toward the nucleus and different types of chloroplast stromule-to-nuclear connections are established (*Caplan et al., 2015*). However, the mechanism behind perinuclear chloroplast clustering and chloroplast stromule-to-nuclear interactions is unknown. Our results described here using TMV-p50-induced, effector-triggered immunity indicate a role for AFs and MTs during perinuclear clustering. MTs promote stromule extensions, contributing to more stromule movement, while AFs provide anchors to position chloroplasts towards the nucleus. These results reinforce the role for AFs as anchor points for stromules that were also previously shown to exist in *Arabidopsis* hypocotyls (*Kwok and Hanson, 2004b*), and a recent study showing that stromules are involved in maintaining contact with nuclei (*Erickson et al., 2017a*). Overall, our results invoke a model in which, during effector-triggered immunity, MTs facilitate stromule extensions and stromules bind tightly to AFs around nuclei. The role of MTs during the formation of stromule-to-nuclei connections requires further studies. However, our data suggests that once those connections are formed, stromules may guide or pull chloroplasts toward the nucleus, which then results in perinuclear clustering of chloroplasts.

Results described here show that MT-mediated stromule extension and AF-mediated stromule anchoring are two complementary activities during stromule formation and movement. We provide mechanistic insights into how interactions with the cytoskeleton form and stabilize stromules. Furthermore, we describe a new type of organellar movement along MTs that is stromule-directed and reveal a mechanism for perinuclear clustering during innate immunity. In the future, it will be

interesting to investigate the molecular components required for stromule dynamics and stromule-directed movement and importance of perinuclear chloroplast clustering during innate immunity.

# Materials and methods

## Key resources table

| Designation | Source or reference | Identifiers | Additional information |
|---|---|---|---|
| *NRIP1* | PMID: 18267075; 26120031 | | |
| *GCP4* | This paper | | |
| *p50* | PMID: 18267075; 26120031 | | |
| GV2260 | PMID: 4022773 | | |
| GV3101 | https://doi.org/10.1007/BF00331014 | | |
| *NRIP1-Cerulean* transgenic plant | PMID: 18267075; 26120031 | | |
| *GFP-TUA6* transgenic plant | PMID:12084822 | | |
| *FABD2-GFP* transgenic plant | PMID:12084822 | | |
| *NRIP-Cerulean*; *N* double transgenic plant | PMID: 18267075; 26120031 | | |
| TRV1 | PMID: 14501071 | | |
| TRV2-EV | PMID: 14501071 | | |
| *Nicotiana benthamiana* | | Taxonomy ID: 4100 | |
| mTalin-Citrine | This paper | | ask construct named 'SPKD2681' |
| Lifeact-TagRFP | This paper | | ask construct named 'SPKD2209' |
| TagRFP-MAP-CKL6 | This paper | | ask construct named 'SPKD2386' |
| NRIP1(cTP)-TagBFP | This paper | | ask construct named 'SPKD3168' |
| TRV-*NbGCP4* | This paper | | ask construct named 'SPKD3111' |
| Citrine-p50-U1 | This paper | | ask construct named 'SPKD1939' |
| Citrine | This paper | | ask construct named 'SPKD914' |
| p50-2xHA | PMID: 18267075; 26120031 | | |
| NLS-mCherry | PMID: 28619883 | | |
| Cytochalasin D | Sigma-Aldrich, St. Louis, MO | C8273 | |
| Aminoprophos-methyl (APM) | Sigma-Aldrich, St. Louis, MO | 03992 | |
| Oryzalin | Sigma-Aldrich, St. Louis, MO | 36182 | |
| Paclitaxel-BODIPY | Thermofisher Scientific, Waltham, MA | P7501 | |
| DMSO | Sigma-Aldrich, St. Louis, MO | D8418 | |
| PRISM7 | GraphPad, La Jolla, CA | | |
| Stromule detection and tracking algorithm | http://sigport.org/1807 | | |

## Plasmids

Plasmids used in this study includes mTalin-Citrine (SPKD2681), Lifeact-TagRFP (SPKD2209), TagRFP-MAP-CKL6 (SPKD2386), NRIP1(cTP)-TagBFP (SPKD3168), TRV-*NbGCP4* (SPKD3111), Citrine-p50-U1 (SPKD 1939), Citrine (SPKD914), and p50-2xHA (TBS44), NLS-mCherry. These were constructed by PCR and standard cloning methods. Details of constructions are available upon request.

## Transgenic marker lines and transient expression by agroinfiltration

Transgenic *N. benthamiana* plant expressing the NRIP1-fused to Cerulean is described in (*Caplan et al., 2015*; *Caplan et al., 2008*). Transgenic *N. benthamiana* plants expressing GFP-TUA6

and FABD2-GFP were gifts from Drs. Manfred Heinlein and Karl Oparka and described in *Gillespie et al. (2002)*. The plants were grown under continuous light at 20°C on growth carts for 4–5 weeks as described in *Caplan et al. (2015)*, *(2008)*. Cultures of GV2260 *Agrobacterium* containing the recombinant plasmids were grown on plates containing Streptomycin (50 mg/L), rifampicin (25 mg/L), and carbenicillin (50 mg/L) and spectinomycin (100 mg/L) antibiotics. Agrobacterium was resuspended in infiltration media containing 10 mM MgCl$_2$, 10 mM 2-Morpholinoethanesulfonic acid (MES) and 200 µM acetosyringone and induced for at least 3 hr. Fully expanded leaves of 3- to 4-week-old *N. benthamiana* were used for agroinfiltration as described in *Caplan et al. (2015, 2008)*.

## Inhibitor treatment

Actin inhibitor Cytochalasin D (200 µM), microtubule inhibitors APM (20 µM) and Oryzalin (300 µM) and the microtubule stabilizing agent, Paclitaxel-BODIPY (0.8 nM) were prepared as 1M stocks in dimethyl sulfoxide (DMSO) and suspended at appropriate working concentrations in the infiltration medium prior to pressure infiltration for imaging. Working concentrations were determined after testing a range of concentrations of the respective inhibitors and agents. The concentrations that resulted in the microtubule depolymerization/stabilization without any lethal effect on the cells at the microscopic level were used further for experiments. Inhibitor treatments were performed by pressure infiltration. A small hole was made on the underside of the leaves with a razor blade. A 1-mL syringe was used to pressure infiltrate inhibitor solutions or a mock containing DMSO ($\leq$0.2%) in infiltration media Leaf excisions approximately 4 mm$^2$ were taken away from the infiltration point and mounted in a Nunc coverglass bottom chamber (Thermo Fisher Scientific). The center of the sample was imaged to minimize effects caused by excision-induced wounding. All time points started immediately following the pressure infiltration of the treatment. The 0–5 min time point after the respective treatments accounts for the time taken for sample preparation and mounting the samples after infiltration with the inhibitors and stabilization agents.

For 1 hr treatment, Cytochalasin D (10 µM) and Oryzalin (1 µM) as well as DMSO (0.1%) as a control were infiltrate in an area of about 3 cm diameter on the same leaf by needleless syringe infiltration. After 1 hr, around 4 mm$^2$ leaf disc away from the infiltrated point were excised and mounted in a Nunc coverglass bottom chamber.

## VIGS assay

NRIP1-Cerulean or GFP-TUA6 *N. benthamiana* transgenic plants were used for VIGS experiments as described in (*Dinesh-Kumar et al., 2003*). *Agrobacterium* culture containing pTRV1 was mixed with culture containing TRV2-EV, or TRV2-*NbGCP4* in 1:1 ratio to adjust an OD$_{600}$ to 0.5. Plants of 6 leaf stage were infiltrated and observed their stromules were observed in leaf epidermis 4 days after infiltration of VIGS vectors. In immune response experiments, *Agrobacterium* culture containing TMV-p50 effector was infiltrated on the third day after VIGS construct infiltration. A total of 48 images were taken from 12 plants by three independent experiments for each condition. Real-time RT-PCR was performed to determine the silencing efficiency. After imaging, RNA was extracted from leaves by plant RNeasy kit (Qiagen) and cDNA was generated by reverse transcription using Superscript III Reverse Transcriptase (Thermo-Fisher Scientific). Real time PCR was performed on a Bio-Rad CFX96 touch$^{TM}$ real-time PCR detection system (Bio-Rad) using iTaq Universal SYBR Green Supermix (Bio-Rad). GCP4-F-realtime 5'-GGATGGTTCATCTCATCAGC-3' and GCP4-R-realtime 5'- AACAACAAGC TGCCACAGAT-3' were used for *NbGCP4* gene expression while EF1α-F-Realtime 5'-CTGGTGTCC TCAAGCCTGGTATGG-3' and EF1α-R-Realtime 5'-TGGCTGGGTCATCCTTGGAGTTTG-3' were used as for control PCR.

## Perinuclear clustering of chloroplasts

To count chloroplast clustering under immune response, two leaves of N and NRIP1-cerulean transgenic *N. benthamiana* were infiltrated with agrobacterium containing citrine 48 hr prior to imaging. On the same leaf, non-recombinant cell or cells containing p50-HA were infiltrated 24, 30, or 36 hr before observation. 4 mm$^2$ leaf tissues away from the infiltrated point were excised and imaged by a confocal microscope. To examine MT structure during immune response, transgenic *N. benthamiana* plants containing *N and NRIP1-Cerulean* or without *N and NRIP1-Cerulean* were infiltrated with Agrobacteria containing p50-HA and TagRFP-MAP-CKL6 36 hr prior to imaging. For the

cytoskeleton inhibitor treatment after inducing immune response, transgenic *N. benthamiana* plants containing *N and NRIP1-Cerulean* were infiltrated with a mixture of Agrobacteria containing p50-HA and NLS-mCherry were infiltrated 35 hr before inhibitor treatment. Inhibitors were infiltrated one hour prior to the imaging.

## Confocal microscopy

*N. benthamiana* leaf sections (4 mm$^2$) away from the infiltrated point were excised, infiltrated with water and imaged on a Zeiss LSM 780 upright confocal microscope, LSM 710 inverted confocal microscope or LSM 880 inverted confocal microscope fitted with 40X C-Apochromat water immersion objective (NA = 1.2) (Carl Zeiss Inc, Thornwood, NY). The 405 nm, 458 nm, 488 nm, 514 nm, or 561 nm laser line was used for TagBFP, Cerulean, GFP, Citrine, or TagRFP, respectively. TagBFP and Cerulean were pseudo-colored cyan, Lifeact-TagRFP and mTalin-Citrine were pseudo-colored magenta, and GFP-TUA6, EB1-Citrine, and TagRFP-MAP-CKL6 were pseudo-colored yellow throughout the manuscript. In the perinuclear clustering experiment, Citrine for cytosol and nucleus diffusion was pseudo-colored blue and mCherry with nuclear localization signal was pseudo-colored blue for consistency of data presentation.

## Image processing

Huygens Professional (Scientific Volume Imaging, Hilversum, Netherlands) was used on the majority of images to deconvolve using a Classical Maximum Likelihood Estimation (CLME) restoration method, to remove drift using the object stabilizer algorithm, to correct photobleaching across time lapsed images and to equalize brightness and contrast. Noise was removed from images that were not suited for deconvolution using a 3 × 3 median filter. Volocity (PerkinElmer, Waltham, MA) was used to generate images, kymographs and videos.

## SOAX microtubule analysis

Bio-filament analyzing program SOAX (*Xu et al., 2015*), which utilizes multiple Stretching Open Active Contours (SOACs), was used in order to determine Curvature, Length and Azimuthal Angles for MT filaments within Maximum Intensity Projections (MIP) of epithelial leaf cells. High-resolution z-stacks were acquired on an LSM 880 confocal microscope or an LSM 710 confocal microscope and deconvolved in Huygens Professional batch conversion, with Regularization per channel decreased to a minimum of 2, and Quality Change Threshold changed to 0.05. Resulting images were then converted into MIPs using Fiji (ImageJ) and analyzed. Regions were selected from five maximum intensity projections for each treatment, toward the central region of epidermal pavement cells which poses clear MT network. Regions were uniform in a radius of 10 μm from the center point, and minor errors in regional snakes were corrected. Following this, each region was analyzed using curvature and snake length analysis. Points of high filament visibility and quality were analyzed within each cell. Curvature and Snake Length were then compiled, while Azimuthal Angles were converted to Mean Resultant Lengths for statistical analysis. All settings for SOAX analysis were kept at program defaults excluding Ridge Threshold, increased to a maximum of 0.04 with a minimum of 0.02, and Stretch Factor, increased to 1. Results were compiled and graphed with Prism 7 (GraphPad). Azimuthal angle color-coding was performed on SOAX analyzed images to display the orientation of MTs.

## Quantification of stromules

Stromules were manually counted using ImageJ (National Institutes of Health, Bethesda, Maryland, USA) from the maximum intensity projections of the confocal images. Mean stromule ratios were determined by counting the total number of stromules and then dividing by the total number of chloroplasts. To quantify perinuclear clustering, *Agrobacterium*-containing p35S::Citrine T-DNA vector was infiltrated into *N. benthamiana* NRIP1-Cerulean transgenic plant leaves as described in *Caplan et al. (2015, 2008)*. Twenty-four hours later, either *Agrobacterium*-containing TMV-p50 or empty vector control was infiltrated. Images in Z series were captured by confocal microscope as described in *Caplan et al. (2015)* at the indicated time points. Perinuclear chloroplasts were counted manually with the cell counter plugin in ImageJ. Experiments were repeated three times with similar results and graphed with Prism 7 (GraphPad).

## Automated stromule tracking

Matlab code was written to perform Fuzzy c-means clustering (FCM), active contour framework, contour smoothing, unit normal feature analysis and branch analysis. In the FCM, we utilize both spectral energy and spatial energy functions for clustering. A 5 by five window around each pixel is used to compute the spatial component. In our experiments, we clustered the spectral domain into eight clusters and compute coefficients for each pixel. Pixels having 30% coefficients as background and 70% as the foreground were then used in the active snake formulation. Matlab code is also written to perform tracking stromule. Segmentation is first performed in 3–4 layers from total 8 layers of z stack. Results are then projected into one image to perform nearest neighbor based tracking.

## Manual stromule tracking

Sixty frames of time-lapse z-stack images of stromule dynamics were acquired every 10 s in NRIP1-Cerulean transgenic plants silenced for *NbGCP4* or vector control with and without the TMV-p50 effector. Maximum intensity projections of time-lapse z-stacks were generated in Zen software (Carl Zeiss) and motion types including, extension, retraction, constant smooth, sudden erratic, and side tangential were manually counted. The maximum and minimum stromule lengths were manually measured using the FIJI version of ImageJ (*Schindelin et al., 2012*). The extension and retraction velocities were calculated using the Cell Counter plugin in FIJI ImageJ, via frame-by-frame analysis. This allows quantification of movement of stromules between frames of a temporal stack, in 2D and 3D.

## Transmission electron microscopy

Transmission electron microscopy was conducted as described previously in *Caplan et al. (2015)*. Leaf excisions were fixed with 2% paraformaldehyde and 2% gluaraldehyde in PHEM (60 mM PIPES, 25 mM HEPES, 10 mM EGTA, and 2 mM MgCl2, pH 6.9) buffer for 45 min) overnight at 4°C. Samples were washed with 0.1 M sodium cacodylate buffer (pH 7.4), postfixed with 1% osmium tetroxide in the same buffer for 2 hr, and then washed with buffer and water. Samples were dehydrated in an acetone series (25%, 50%, 75%, 95%, and twice in anhydrous 100% acetone; 30 min each step) and infiltrated with Quetol 651-NSA resin. Ultrathin serial sections were cut on a Reichert-Jung Ultracut E ultramicrotome and collected onto a film of 0.5% formvar using 2 × 1 single slot grids. Sections were post-stained with methanolic uranyl acetate and Reynolds' lead citrate and examined with a Zeiss Libra 120 TEM operating at 120kV. Images were acquired with a Gatan Ultrascan 1000 2k × 2 k CCD.

## Statistical analysis

Statistical analysis was performed using Microsoft Excel 2013 (Microsoft) and Prism 7 (GraphPad). Stromule counts were performed on 3–4 images obtained at the appropriate time points depending on the drug treatment. Experiments were repeated at least three times. For experiments involving the Paclitaxel-BODIPY treatment each image was considered a replicate and the experiment was repeated three times. Student's t-test with Welch's correction was performed to examine difference between treatments. For stromule frequency, the results passed the D'Agostino and Pearson's normality test. Thus, t-test with Welch's correction was used to evaluate the differences. For the stromule lengths, rank transformation was performed and Mann-Whitney test was used for comparison. Comparisons of velocities of stromule extension and retraction between all the conditions were done using Dunnett's multiple comparison. For the perinuclear clustering, non-parametric Mann-Whitney t-tests were performed to evaluate the differences. All graphs were formed with Prism 7 (GraphPad). Statistical analyses and graph generations were performed using Prism 7.

## Acknowledgements

We thank Drs. Jung-Youn Lee and Bo Liu for providing MAP-CKL6 and EB1 plasmids respectively. We thank Drs. Manfred Heinlein and Karl Oparka for providing GFP-TUA6 *N. benthamiana* transgenic seeds. The National Institute of Health R01 grant GM097587 to SPD-K and JLC, supported this work. Microscopy access was supported by grants from the NIH (P20 GM103446, S10 OD016361 and S10 RR027273).

## Additional information

### Funding

| Funder | Grant reference number | Author |
|---|---|---|
| National Institutes of Health | R01 GM097587 | Savithramma P Dinesh-Kumar Jeffrey Lewis Caplan |
| National Institutes of Health | P20 GM103446 | Jeffrey Lewis Caplan |
| National Institutes of Health | S10 OD016361 | Jeffrey Lewis Caplan |
| National Institutes of Health | S10 RR027273 | Jeffrey Lewis Caplan |

The funders had no role in study design, data collection and interpretation, or the decision to submit the work for publication.

### Author contributions

Amutha Sampath Kumar, Conceptualization, Data curation, Formal analysis, Visualization, Methodology, Writing—original draft; Eunsook Park, Conceptualization, Data curation, Formal analysis, Investigation, Visualization, Methodology, Writing—original draft, Writing—review and editing; Alexander Nedo, Conceptualization, Data curation, Validation, Visualization, Methodology; Ali Alqarni, Formal analysis, Validation, Investigation, Writing—review and editing; Li Ren, Software, Methodology; Kyle Hoban, Formal analysis, Investigation, Visualization, Writing—review and editing; Shannon Modla, Investigation, Methodology; John H McDonald, Formal analysis, Methodology, Writing—review and editing; Chandra Kambhamettu, Software, Formal analysis, Supervision, Methodology; Savithramma P Dinesh-Kumar, Conceptualization, Resources, Data curation, Formal analysis, Supervision, Funding acquisition, Validation, Investigation, Methodology, Writing—original draft, Project administration, Writing—review and editing; Jeffrey Lewis Caplan, Conceptualization, Resources, Data curation, Software, Formal analysis, Supervision, Funding acquisition, Validation, Investigation, Visualization, Methodology, Writing—original draft, Project administration, Writing—review and editing

### Author ORCIDs

Eunsook Park (ID) http://orcid.org/0000-0003-2984-3039
Savithramma P Dinesh-Kumar (ID) http://orcid.org/0000-0001-5738-316X
Jeffrey Lewis Caplan (ID) http://orcid.org/0000-0002-3991-0912

### Decision letter and Author response

Decision letter https://doi.org/10.7554/eLife.23625.054
Author response https://doi.org/10.7554/eLife.23625.055

## Additional files

### Supplementary files

• Transparent reporting form
DOI: https://doi.org/10.7554/eLife.23625.052

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
