## [Decision Letter]

[Editors’ note: this article was originally rejected after discussions between the reviewers, but the authors were invited to resubmit after an appeal against the decision.]

Thank you for submitting your work entitled "Stromule extension along microtubules coordinated with actin-mediated anchoring guides perinuclear chloroplast movement" for consideration by *eLife*. Your article has been reviewed by three peer reviewers, and the evaluation has been overseen by a Reviewing Editor and a Senior Editor.

Our decision has been reached after consultation between the reviewers. Based on these discussions and the individual reviews below, we regret to inform you that your work will not be considered further for publication in *eLife* in its present form (see details below).

The reviewers and reviewing editor discussed the manuscript and the reviews in detail. They concluded that, while there were some interesting observations, significantly more work would be needed to solidify the claims (see the reviews). A point that came up in the discussion is that some data are over-interpreted, such as the data to suggest that stromule extension requires MT's. The data presented indicates that application of MT depolymerizing drugs reduces the number of observed struggles per plastid. This observation indicates a possible role in stromule maintenance, not a requirement in their extension. Keeping with the *eLife* spirit of not inviting revision when the requested experiments will likely take longer than a couple of months, we are declining the work. However, we are interested in the topic. A thoroughly revised manuscript that addresses the reviewers' concerns would likely be reviewed again. While it will be treated as a new submission, we would try to solicit advice from the same reviewers.

*Reviewer #1:*

I have reviewed the manuscript by Kumar et al. on " Stromule extension along microtubules coordinated with actin-mediated anchoring guides peri-nuclear chloroplast movement." The researchers have used a number of probes to establish the role of microtubules and provided some excellent images. The portion dealing with the cytoskeleton is generally well done and I commend the researchers for it. However, I do have major concerns that must be addressed in order to make the work make a higher impact.

1) The researchers use 4mm^2^ portions of tobacco leaves (that have been previously pressure infiltrated with different solutions). I am concerned that wounding of the leaf tissue plays an important role in their conclusions. Although controls have been used and I am not questioning the observations as such, it would be a much more convincing observation if portions of intact leaves (or large wounding-free areas) were imaged.

2) The Materials and methods portion is very sketchy and does not provide sufficient detail to allow the experiments to be repeated and verified by other labs. The researchers/manuscript would greatly benefit from clear statements of how things were actually done in this study, rather than referring to earlier publications for methodology.

3) It is important to know the time at which leaves were infiltrated (how many hours after exposure to light/daylight)?

4) Results section

"Beaks extend into longer stromules. Longer stromules were seen associated as just the tips of stromules bound to MTs or the full-length of the stromule extended along MTs. More complex stromule structures were observed associated with MTs." What does the “or” in the statement mean? While the stromule-tip-association suggests a good link to stromule extension, what happens when a stromule is totally aligned with a MT. Is this a coincidence or is there more to it? What are the more complex stromule structures?

5) 11% of stromules were not associated with MTs. The authors explain this by resorting to complete speculation and "MT-independent mechanism of stromule formation ". What are these other mechanisms and where is the evidence for them and their link to the observations presented here? If all stromule formation and extension is not based on MTs then I do not see the big discovery here. The observations presented then just become a biased subset that is chosen to make a point while anything that does not fit in is neglected.

6) The authors wished to 'conclusively demonstrate that stromules extend along MTs' and therefore used two inhibitors to disrupt the MTs. Their evidence for MT depolymerization was that stromules disappeared after 15 minutes. How can a specific assay be used and the observations then used as evidence that the assay is working? I find the reasoning faulty and wonder if this not a case of circular reasoning?

The stromules disappeared 15 minutes after APM and Oryzalin treatment: is this 15 minutes after pressure infiltration? Is it 15 minutes after the 4mm^2^ leaf excision was taken for viewing? What does disappearance mean? Did they retract? Were they not formed? It must be made clear what disappearance means.

7) The concentrations of inhibitors used are, in general, extremely high and would completely disrupt cellular functions, especially after pressure infiltrations. The authors provide minimal information on this by stating "The concentrations that resulted in the microtubule depolymerization/stabilization without any lethal effect on the leaves were used further for experiments." It must be made clear in the Materials and methods section what was done to ascertain that the cells were actually surviving and functioning normally after the harsh treatments.

8) The VIGS assay creates a weak point in this study. The time frame for the experiments involving VIGS is well beyond the induction period for stromules. The researchers actually observed plants four days after VIGS and found numerous anomalies including significantly altered / bundled cortical MT organization in leaves. They also found twice the number of stromules compared to VIGS vector control. The conclusion of high stromules number after 4 days in NbGCP4-silenced plants is at best just a correlation; it is not a direct evidence suggesting a clear cause and effect relationship. Perhaps the greatly altered MTs changed a lot of other things that could then then also have affected stromule frequency. the plant was clearly no longer operating in a normal manner. The same reasoning applies to the further increase in stromule length and number upon expressing p50 subsequently. In this case the observations were taken 24 hours later. Could something more have happened to create a physiological change in the 24 hour period? What happened at 4, 8, 12 hrs? That is definitely sufficient time for stromule formation to have occurred (Brunkard et al., 2015).

9) There are some really obtuse statements:

“Stromules appeared to interact statically, and not dynamically, with AFs, suggesting there are actin anchor points along stromules.” What is meant by 'interact statically and not dynamically'? actions. Is the word 'interaction' itself not denoting a dynamic phenomenon?

Further, the stromules became thin at anchor points – “We observed a clear thinning or constriction at the site of stromule-to-actin interaction points along the length and across the body of the chloroplast". Did the actin anchor create this thinning? As I understand the authors are talking about an anchor point and not a band of F-actin. If not, then how is the thinning of a stromule explained ? What is meant by “across the body of the chloroplast”? Are we still talking about thinning?

10) In a similar manner the authors state "At 8 min, the stromule began to re-extend along the MT, between 8 min and 9 min, the body of the chloroplast was released and this resulted in stromule driven chloroplast movement." What are the authors trying to convey? The body of the chloroplast was released! From what (?) and how did this result in a conclusion of 'stromule driven chloroplast movement'? The authors did not actually establish 'stromule driven movement'. Did the stromule pull the body or push it? Did the stromule elongate further or did it retract in order to drive chloroplast movement?

11) It is well observed that stromules arise in many different areas of a chloroplast randomly. Does this always coincide with chloroplast movement? The authors address this issue: "To examine how disruption of stromule and chloroplast AF anchoring effects chloroplast movement, we followed two chloroplasts in an epidermal cell treated with CytD." As I understand from the manuscript only one chloroplast was actually observed to reach the conclusion since the 1st one had restricted movement. Interestingly this singular observation is supposed to have added to the evidence that stromules may direct chloroplast movement in the absence of intact AFs.

12) The weakest part of this manuscript is the proposed link to 'innate immunity'. This part of the study takes the manuscript into a highly speculative mode where the nice demonstrations of microtubule alignments and the presence of actin are undone by a lack of rigorous experiments. The authors point to a feature of innate immunity as 'eventual peri-nuclear clustering of chloroplasts'. It is notable that chloroplast clustering around nuclei happens routinely in leaves, even without bringing innate immunity into play. In the excised 4mm^2^ sections what is the baseline number of chloroplasts clumping together and how is it accounted for by the idea of innate immunity? Perhaps the wounding of tissue is responsible for innate immunity? This still does not explain the chloroplasts around nuclei in normal, unwounded (or in any untreated tissue). Clearly stromules have been observed by the authors under conditions that trigger the innate immunity but I fail to see an exclusive response where a cause and effect relationship is indicated with regard to stromules.

Some references are missing and should be added. Reference for the presence of chloroplasts in tobacco pavement cells is needed.

The Introduction as well as discussion of earlier work on actin and microtubules must be done more thoroughly rather than be given a cursory mention. The authors must clearly state how their results advance the field and are not a repetition of earlier inhibitor experiments with a rather speculative link to chloroplast movement and plant immunity.

The authors have only looked at chloroplasts in pavement cells. For this study to have a real impact the observations must take into account other kinds of plastids, including plastids in mesophyll tissue and leucoplasts, all of which are known to display stromules.

*Reviewer #2:*

In this study Amutha Sampath Kumar and colleagues investigated the dynamics of stromules generation, elongation and retraction. The authors’ major discoveries indicate that stromules extend along microtubules (MTs) but not actin filaments (AFs). Moreover, they show that stromules extension and number per chloroplast depend on MTs integrity and stability. They also show a possible role for AFs as anchor points for stromules and plastid bodies around nuclei. Interestingly, they also provide evidence that stromules extending along MTs drive the movement and clustering of plastids to the nuclei during defense responses. This stromules/plastid behavior was previously reported by the same research group as being important for resistance against viruses (Caplan et al., 2015).

This work provides new and a significant information about the formation/regulation and function of the understudied plastid stromules. Although one can argue that the work it is generally based on correlative results, I consider that this data will be a keystone for future studies in this topic. I think that the manuscript fits with *eLife*'s scope. However, I have some concerns about how the results are presented, described and its interpretation. Also, I consider that authors should support with more data the proposed model. Particularly, authors should address two things:

- The authors use the terms associated, bound, touching, interacting, etc. as "synonyms" but without specifying what they exactly mean. They should define these terms in a "precise" stromules-MTs distance(s) manner. Indeed, I do not think authors demonstrated interaction in any case. Authors should do 3D co-localization analyzes. In addition, they should consider doing FRET analysis if this is possible.

- Authors should analyze the dynamics changes of MTs and AFs networks after p50 defense induction. Are MTs stabilized after p50 expression? Are perinuclear AFs increased after p50? These results could be of great support for the proposed model. I believe the authors could do both analysis using the data that they already have.

The authors should also address the following comments:

Abstract:

1) Misspelling: effects should be affect.

2) Authors should revise and change the statement suggesting that AFs have no role in stromule extension. I think that providing "…anchor points for stromules to prevent their retraction while extending on MTs…" sounds like an important role.

Results:

1) Section "Microtubules are required for stromule extension". Authors should consider that the drugs used inhibit the increase of stromules per plastid associated to mock treatment instead of reducing them.

2) Section "Microtubule stabilization increases stromule number, length and stability", first paragraph: What are authors considering as MTs stabilization in Figure 2? The image showing cells treated with paclitaxel should be label indicating this MTs stabilization. Are stromules longer in cells treated with paclitaxel?

3) Section "Microtubule stabilization increases stromule number, length and stability". Figure 3: Authors point out that the stromules number increase observed in NbGCP4-silenced and in p50-induced plants are "remarkably similar". They should analyze the MTs stability during p50 induction to support this statement.

4) Section "Microtubule stabilization increases stromule number, length and stability". Figure 3: Authors should describe and discuss the extension/retraction results obtained after treatment with p50.

5) Section "Stromules direct chloroplast movement in the absence of actin filaments", second paragraph: Although authors mention that exist "opposing forces" and a "rapid pulling" provided by stromules, they do not have any measure of these traction forces. Authors should be cautious with results interpretation and re-phrase this paragraph.

6) Section "Actin microfilaments mediate perinuclear chloroplast clustering during plant immune response", second paragraph: authors should replace the sentence "…connections, but were unable to find…" by "…connections, but we were unable to find…"

Discussion:

1) Third paragraph: "…and propose the difference…" should be replace by "…and propose that the difference…"

2) Third paragraph: What do the authors mean with the statement "directly interacted"? I do not think they showed any interaction.

Figures:

1) Figure 1: How authors define tip connection or extension along MTs? How close are stromules and MTs? Are authors using z-stacks projections to analyze this distance? Authors should clarify this point in every image/quantification where an interaction/connection/touching, etc. suggestion is made.

2) Figure 1: Are these analyses using only z-stack projections or they consider 3D/4D dimensions? Figure 1: Is the algorithm calculating the 3D distance between stromules and MTs?

3) Figure 2: Authors should show if exist statistical differences between times/treatments.

4) Figure 2: What do the authors consider as stabilization of MTs in this image? They should do an analysis similar to the Figure 3.

5) Figure 2: Is there a positive correlation between the increase on stromules number and the stromules "associated" to MTs?

6) Figure 5: Authors should add the separated MTs and AFs channels to this figure.

7) Figure 3—figure supplement 2: Authors should add stats to this quantification. They should also consider using this data as a main figure.

*Reviewer #3:*

Higher plant plastids are observed to produce projections called stromules. It has been speculated that stromules may serve in exchange of metabolites or molecular signals to other organelles, including the cell nucleus. In a previous publication, the authors presented evidence that stromules may in fact aid in transport of proteins and reactive oxygen species to the cell nucleus in the context of innate immunity. In this study, the authors address cellular mechanisms by which stromules are extended, how they may interact with other components of the cell, and how stromule formation and dynamics may guide plastid movements and accumulation around the cell nucleus during innate immune response.

Previous studies have implicated both the actin and microtubule cytoskeletons in modulating stromule extension and dynamics, with actin emerging as the dominant player in stromule extension and plastid interaction with the cell periphery. Here, the authors acquire beautiful dynamic images of stromule dynamics showing compelling evidence for the guidance of stromule extension by cortical microtubules, conclude that such interactions contribute to plastid movement, extend previous observations of actin-implicated anchoring of plastid membranes at the cell cortex, and examine these mechanisms in the context of plastid accumulation at cell nuclei during the innate immune response. They propose a model whereby stromule extension mediated by microtubules helps to bring stromules to the nuclear periphery, where they are captured by surface-localized actin which can serve as attachment points for stromule mediated movement of plastids to the nucleus.

The Introduction and Discussion of the paper are well written, touching on the key literature. The microscopy is beautiful and sophisticated analyses are performed on some of the image data sets using tools that other plant biologists will be interested in knowing about and perhaps applying. A strength of the study are the time series images showing evidence for guidance of stromule dynamics by cortical microtubules, this result was nicely controlled for by using multiple markers for microtubules, making the possibility of an aberrant interaction between stromules and an overexpressed microtubule marker unlikely. However, there are issues with data interpretation that need to be addressed and I felt that a number of major and secondary conclusions were not supported well by the data presented.

A) The authors test the role of microtubules in stromule maintenance by applying depolymerizing and stabilizing drugs and by manipulating microtubule organization by genetic knock down of a nucleation complex protein. Previous studies had indicated that microtubules play a relatively minor role in stromule abundance, but here microtubule depolymerization resulted in a significant reduction in stromules per plastid compared to the mock control. This was the strongest section of the paper but I have a few comments.

1) The following point is not a major criticism, but an odd thing with the drug experiments was that stromules significantly increased in abundance in the mock control over 15 minutes. The reduction in stromule numbers with drug treatment observed at the same time were only slightly lower than the numbers observed at time 0-5 minutes after treatment. If just stromule numbers at time zero and 15 minutes were compared, one would not infer a strong effect of microtubules. The reason for the increase in stromule number in the mock is neither explored nor explained. A couple of possibilities are that stromule formation may be stimulated by stress caused by placing the tissue in the microscope slide mount, or perhaps that light used for the imaging stimulates stromule formation. Since it is an effect on the increased stromule numbers that is observed with drug treatment, it would be nice to have bit more insight into why these number are elevated over the course of the experiment. The two possibilities mentioned could be tested by not imaging at time point 1, and by determining if media exchange during imaging can mitigate the induced increase in stromules over time in the slide mount.

2) It was a good idea to manipulate microtubules by a secondary means, other than drug treatment. However, here the authors conclude that altered microtubule dynamics were responsible for observed reduction in stromule extension and retraction rates. Since microtubule dynamics were not measured, nor were observations shown of possible interaction between stromule tips and microtubule ends, it is not clear why this conclusion was reached. What is evident is that microtubule organization is altered in the GCP4 knock down, with an increase in parallel organization of microtubules in leaf epidermal cells, a result that has been shown previously for knockdown of GCP4 and also for GCP-WD/NEDD1. While the results are striking, it is not clear why a change in polymer organization would drive a change in stromule dynamics. My suggestion would be to keep the interpretation open and not ascribe it to microtubule dynamics. It is possible that the effect is indirect. For example, by altering levels of free MAPs due to reduced total lattice in these cells.

3) The authors conclude that microtubules play a direct role regulating stromules. While agree that the data shown are consistent with that idea, it is possible that the relationship is less direct, as the authors touch on in their Discussion when considering the ER and membrane contact sites.

4) The tip tracking method is very nice. It is kind of a shame though that it is not used to more rigorously quantify stromule tracking of microtubules. Only one example of such tracking is shown, with no summary measurements made over many cells, something that could be done with such a method. A further point here is that it should be defined more clearly how "on tube" vs. "not on tube" is determined, since the underlying molecular relationships are sub-resolution.

5) The SOACs analysis is used to assess features of polymer orientation, curvature and length. While this analysis produced a nice color-coded image of the orientation of segmented features, these data were not used to quantitatively assess patterns of polymer orientation, which again, seems a missed opportunity to show robustness of observations over many cells and samples. A second point, and an important one, is that by inspection of the raw data and the processed data, the method is not an appropriate means to assess polymer length, and those data should be dropped. It is not possible with such segmentation to tell if extended structures are composed a few long polymers or many shorter polymers, nor can it tease apart what happens were polymers meet and overlap at angles. Determining the length distribution in these arrays is an important goal in the field, but it is not trivial.

6) It is stated that GCP4 knock down results in greater bundling. This is not determined in the present manuscript.

B) The authors conclude that actin filaments do not extend along actin filaments but that actin filaments do serve as static anchor points for stromules, and also prevent stromule retraction.

1) The data for stromule extension along actin cables are not quantified, but presented as example images and a video. These images show little evidence for stromule extension along actin bundles, but to be robust, a method should be used to census the relationship of stromules and actin structures so that data from multiple cells and tissue samples can be assessed. Even so, it should also be at least discussed that actin features may be present that did not label with the probes used.

2) In the CytD experiments, why didn't stromule number increase from 0 to 15 minutes in the mock as was observed for the microtubule drug experiments? If this effect does not occur in every experimental setup, the microtubule drug experiments should be repeated under conditions where stromule number does not go up in the mock.

3) The evidence of static anchor points at actin filaments is weakly presented. A single kymograph is shown for analysis of stalling during stromule retraction, and in this single analysis it is not clear that the stall actually occurs at the position of actin signal. The signal corresponding to the stromule tip is offset from the actin signal. A much more rigorous analysis of retraction and actin overlap sites is required.

4) Likewise, the image sequence in Figure 5 and corresponding video are not convincing of retraction back to an actin-defined anchor point – the stromule tip appears to overshoot the "kink" position at the actin branch site. This is also a single image sequence example.

5) The analysis in Figure 5 looks more quantitative, but it is a less direct measure of interaction. Rather, it is measuring a predicted consequence if the proposed interactions exist. It is also poorly explained and it took some time to determine what the graph represents. I think the second bar was meant to be labeled "not fully retracted", rather than "actin anchored', which is a conclusion of function rather than an observation.

C) The authors conclude that loss of actin anchoring allows stromules to direct plastid movements.

1) Once again, these data consist of example images, not more global measurements across cells and tissue samples. In fact, just one image series example. It is a beautiful video, but it cannot be determined from this one example how generalized the phenomena shown are, nor how strong is the correlation of plastid movement with the direction stromule extension. A suggested means of analysis would be to assess each plastid displacement of x distance or greater in a set of time series acquired from multiple cells and leaves. For each displacement, stromule position and orientation would be determined and an orientational resultant would be estimated. The relationship between the predicted resultant angle of "pull" and the actual direction of movement could be plotted and analyzed.

2) A partial loss of actin structure by the CytB treatment is somewhat unsatisfying. The authors chose not to use LatB, which is more effective, because MT organization is also affected. However, I think these data should still be shown to ask if the observed movements of plastids are still seen when actin is more completely disrupted. It will be evident if stromule interaction with MTs (MT-dependent interaction sites) is also affected.

D) The authors conclude that actin anchoring mediate plastid accumulation at cell nuclei during the innate immune response and suggest a model by which plastid accumulation is facilitated by plastid movement driven microtubule- and stromule-based plastid motility, followed by actin mediated capture at the nucleus.

Another case of data by limited example. While the example shown in Figure 7 is compelling, more than this is needed. I was surprised that the hypothesis was not tested by actin disruption, for example. Further, it should be asked if plastid motility, and nuclear accumulation are significantly reduced if microtubules are depolymerized.

E) In general, the authors should be careful about distinguishing conclusion and interpretation from observation. This issue occurs throughout the manuscript.

[Editors’ note: what now follows is the decision letter after the authors submitted for further consideration.]

Thank you for submitting your article "Stromule extension along microtubules coordinated with actin-mediated anchoring guides perinuclear chloroplast movement during innate immunity" for consideration by *eLife*. Your article has been reviewed by two peer reviewers, and the evaluation has been overseen by a Reviewing Editor and Christian Hardtke as the Senior Editor. The reviewers have discussed the reviews with one another and the Reviewing Editor has drafted this decision to help you prepare a revised submission.

As you can see from the detailed reviews below, the reviewers very much appreciate your efforts to improve the initial manuscript. Yet they feel some additional adjustments are required before the paper can be published. These are mostly of organizational/semantic nature, and do not require additional experimentation. Apart from answering point by point to the reviews below, please pay particular attention to the following:

1) The reviewers are of the opinion that the analysis of retraction pausing with sites of actin filaments needs to be strengthened quantitatively. Currently, the data consist of example image sequences and an indirect measure involving pausing frequency, which is not explained very clearly. E.g., it is not obvious that this experiment correlates pausing with sites of possible actin action.

2) The reviewers believe that it would be a good idea to quantify tracking of microtubules by stromule tips rather than to show one example of such tracking.

3) Clarification is needed about how the circular correlation coefficient data relate to the results with the 30 degree cutoff angle. These data are new to the revised manuscript and need to be explained more thoroughly.

*Reviewer #2:*

The authors have addressed to most of my concerns. Also, they've clarified some other aspects raised by the other reviewers.

They included in this version new experiments improving the manuscript. Particularly, they added supporting results for the proposed model where they link the cytoskeleton, immunity and plastid movement/clustering around nuclei. I consider that this is new and significant information for the research field. In addition, authors included interesting quantification data for a more robust correlation between plastid movement and the direction stromule extension.

Nevertheless, one caveat is that authors are not showing quantitative data on the stromules tip tracking. The authors argue that the algorithm only works for the overexpressing EB1 construct, which could be changing the general MTs dynamics. I think this could also be true for the other OE constructs. Thus, authors should consider adding tip tracking experiment quantifications, and show the reproducibility for this method, at least for EB1.

Altogether, I think that although mostly based on correlative data, this is a foundational work for future investigations in this topic and should be published in *eLife*.

*Reviewer #3:*

All in all, the authors have addressed many of the concerns and comments made in the first review. In addressing these comments, a number of substantive additions to the manuscript have been made, such as quantifying the relationship of the angle of stromule extension and the direction of subsequent plastid movement. However, some concerns remain and there are also some new questions raised by the new datasets presented. The main issues concern the quantitation of actin anchoring of stromules (Figure 6), some follow-up questions on the new data quantifying the relationship between stromule extension angle and plastid movement, and the entire section on perinuclear clustering (should it be included?). Specifics are delineated below:

Point-by-point responses to authors' replies to initial review.

Point A1: In my first read, I missed the fact that DMSO treatment was associated with an increase in stromule number over time. This is perhaps because the media control was not shown in the figure for comparison. The revised text makes it more clear that DMSO causes an increase in stromule number over the media control, and that drug treatment reduces this increase. However, I am now wondering why the media controls are not shown in the figure. It would be useful to know if drug treatment also causes a reduction of stromules in the media control, even if there is no increase in time otherwise. If not, does this mean that induced stromules are somehow different from existing stromules in regard to microtubule depolymerization?

Point A2: The revised text addresses the concern raised about ascribing the changes in stromule dynamics to changes specifically in microtubule dynamics. However, I still think it would be appropriate to make clear that, while the manipulation here is pretty specific, the observed changes in stromule dynamics may be either a direct or indirect consequence of changes in microtubule organization. For example, the statement – "Collectively, these results indicate that change in MT organization in NbGCP4-silenced plants or during p50 induced immunity control stromule dynamics." – is stronger than I think the data warrant and should be restated. "Collectively, these results indicate that changes in MT organization caused by NbGCP4-silencing plants or during p50 induced immunity are correlated with changes in stromule dynamics, indicating a possible role, direct or indirect, for MT organization in modulating stromule dynamics."

Point A3:.The revised text addresses the stated concern.

Point A4:.Thanks for the clarification. I agree that the figure is useful for portraying a sense of dynamics in a static format. The original text did not make it clear that the figure was produced manually. Unfortunately, this is also not clear in the revised text. The manual method is described in the Materials and methods, but the text and figure legend do not indicate how this figure panel was made. I think a simple edit to the figure legend would fix this.

Point A5:.Revised text is an improvement.

Point A6:.OK.

Point B1: I think it is important to give a sense of how much space was searched through to say "almost never". For example, you could state that "at least x stromules were examined in at least y cells, and only n stromules showed any evidence of extending along actin filaments. Even in these cases…"

Point B2: OK.

Point B3: See comment for 5 below.

Point B4: See comment for 5 below.

Point B5: While more examples of stromules retracting to locations proximate to actin filaments, Figure 6 provides the only quantitation of this behavior. As indicated in the first review, it took me a while to figure out what I thought this panel represented. In reviewing this experiment again, I confess that I still find it unclear what exactly was measured here. The bar graph shows the percentage of fully retracted vs. partially retracted stromules. The figure legend states: "The percent of stromules containing actin anchors was quantified by monitoring retraction events in the samples described in panel A. Retracting stromules without actin anchors resulted in full retraction." However, in the text it states "70.3% + 0.02% (SEM) of retracting stromules partially retracted to one or more AF." So, was just the percent of partial retraction measured, or were retraction pause events also correlated with actin signal? If the latter, I might expect to see an analysis of how often retraction pausing was observed at sites of actin signal, but these data are not shown. If all partial retractions are indeed caused by actin anchors, then the percent of partial retraction alone would serve as an indirect estimate of the prevalence of stromules with actin anchors. However, is this known? Can stromules pause for other reasons? If they can, this experiment does not really probe prevalence of actin anchors well, nor does it test for the role of actin in pausing.

Point C1: The new data measuring the correlation of stromule angle with movement angle are an important addition to the manuscript. The angular correlation plot in the supplementary data is actually stunning. I would put it in the main figure. However, I think it should be stated more clearly how plastids were selected for this analysis. Were just plastids with one stromule selected? If so, how were they chosen? Also, a significant question is raised by the oryzalin experiments. In oryzalin treated cells, the circular correlation coefficient of stromule angle with the direction of plastid movement actually appears to go up (.85 vs.76 for no oryzalin). Yet when a 30 degree difference in stromule and movement angles is established as a cutoff for positive stromule directed movement, it is concluded that oryzalin treatment reduces stromule directed movement. It is hard for me to see how these two results are reconciled. If the correlation coefficient alone is considered, one would not conclude that oryzalin treatment reduces stromule directed movement in fact, it may even increase it.

Point C2: OK.

Point D1: It seems that the new drug tests during perinuclear clustering were useful. I agree that it may be challenging to interpret the results with cytochalasin D as it now turns out that actin is required for plastid movement and not just anchoring in these cells. Interestingly, this duality of actin being required for both movement and anchoring was also observed by Wada and colleagues for light driven plastid movement (it would be good to point this out in the Discussion). I am, however, a bit confused by continuing to propose a model where plastids are directed or guided during the early stages of the clustering response by MT's, since no effect on clustering by MT disruption was observed. In other words, the new data do not support all the particulars of the model. There seemed to be general agreement in the first review that this last experimental section was not a strength of the manuscript. I believe it may be a good idea to leave these studies out for now and have them be part of a future manuscript addressing specifically the mechanism of the clustering response.

Point E1: The text is generally improved in regard to clearly distinguishing between observation and interpretation.

Other comments and questions on the revised text:

Data questions:

1) Section “Microtubules are required for stromule formation and extension”: In relating the results that support a role for MT's in stromule extension, it is stated that stromules in the drug treated cells completely retract after 15 min. of treatment. If this is the case, can the authors comment on the data shown in Figure 3, which indicate only a modest drop in the number of observed stromules in the drug treated cells from time 0' to time 15'? Also, it is not clear that these differences are significant. The standard errors for APM look like they mutually overlap the means at 2 sigma (0' to 15'), It is not clear what is going on with the stats for the oryzalin data.

2) Section “Actin filaments serve as anchor points but not as tracks for stromule extension”, last paragraph: Can the authors comment on the observation that 70% of stromules were observed to only partially retract, a state that is attributed to actin interaction, whereas the number of observed stromules in cells treated with 200 µM CTD was the same as in mock treatments? If actin interaction is the primary means of preventing complete retraction of stromules, it seems that the number of observed stromules would be expected to be lower in cells where actin is disrupted.

[Editors' note: further revisions were requested prior to acceptance, as described below.]

Thank you for resubmitting your work entitled "Stromule extension along microtubules coordinated with actin-mediated anchoring guides perinuclear chloroplast movement during innate immunity" for further consideration at *eLife*. Your revised article has been favorably evaluated by Christian Hardtke (Senior Editor), a Reviewing Editor, and two reviewers.

The manuscript has been improved but there are some remaining issues that need to be addressed before acceptance. These are outlined below under reviewer #3. Additionally, since the work was in review, another paper appeared in the Plant Journal with similar results to yours. You should refer to the other study in your revised manuscript and point out any similarities/differences in the work (see doi: 10.1111/tpj.13813).

*Reviewer #3:*

The authors have addressed my previous concerns save one. At the end of the Results, the authors conclude:

"In conclusion, we propose a model in which perinuclear clustering of chloroplasts involves stromule extension along the MTs, stabilization of these extensions by anchor points to AFs surrounding nuclei and stromules guide chloroplasts towards nuclei during an immune response."

However, the authors reported that no connections with microtubules by stromules were observed during P50 induction of perinuclear clustering, and treatment with oryzalin had no effect on clustering. Thus, the experimental results do not support a role for guidance of stromules by microtubules in perinuclear clustering, indeed, they appear to contradict this possibility. The statement about microtubule guidance needs to be omitted from the proposed model for perinuclear clustering.

The description of the experiments addressing pausing of stromule retraction at AFs is now much more clear. However, the statistics in Figure 6 compare the rates of stromule pausing with the rate of full retraction. This comparison is relevant for asking if pausing is significantly more frequent than full retraction, but the main question here is whether pausing occurs at AFs more frequently than might be explained by chance alone. The rate of pausing at AFs by chance alone is a function of AF density; the higher the AF density, the more often pauses would be observed in association with them, even if there is no functional connection. Thus, AF density needs to be taken into account in a statistical test for the observed rate of AF-associated pausing (this can be a bit tricky due to the dynamics of the AFs). However, it seems pretty clear from the figures and the movies that the density of labeled AF's is lower than could easily explain a 77% rate of pausing at labeled AFs by chance alone (this would mean that, on average, about 3/4 of the stromule length would overlap AF signal). Even though the association is challenging to test formally, a statement to this effect might make more clear why the observed rate of 77% is a reasonable suggestion of an association.

---

## [Author Response]

[Editors’ note: the author responses to the first round of peer review follow.]

Reviewer #1:I have reviewed the manuscript by Kumar et al. on " Stromule extension along microtubules coordinated with actin-mediated anchoring guides peri-nuclear chloroplast movement." The researchers have used a number of probes to establish the role of microtubules and provided some excellent images. The portion dealing with the cytoskeleton is generally well done and I commend the researchers for it. However, I do have major concerns that must be addressed in order to make the work make a higher impact.1) The researchers use 4mm^2^ portions of tobacco leaves (that have been previously pressure infiltrated with different solutions). I am concerned that wounding of the leaf tissue plays an important role in their conclusions. Although controls have been used and I am not questioning the observations as such, it would be a much more convincing observation if portions of intact leaves (or large wounding-free areas) were imaged.

We thank the reviewer for bringing up this concern, since it is something we considered in our experiments. We have done other experiments looking at the edge close to the cut site, and observed wounding effects 3-4 cells into the tissue. Therefore, our treatments were done on intact leaves and a 4mm^2^ leaf section was excised away from the infiltrated point (no damage on the leaf section was observed), mounted in the observation chamber, and only the center was imaged immediately. This approach minimized variation due to wounding.

We clearly defined these conditions in the Materials and methods subsections “Inhibitor treatment”,” Perinuclear clustering of chloroplasts” and “Confocal Microscopy”.

2) The Materials and methods portion is very sketchy and does not provide sufficient detail to allow the experiments to be repeated and verified by other labs. The researchers/manuscript would greatly benefit from clear statements of how things were actually done in this study, rather than referring to earlier publications for methodology.

It is routine to refer to previous work unless there is a modification to the previously described method. Therefore, we either described how the experiments were performed or referred to our previous publications in the Materials and methods section. In many cases, we cited our previous publications that describe the method, and also, we reiterate the details of the experiment. We did cite some methods without reiterating details, and in these cases, we repeated the details from previous publications in this manuscript.

3) It is important to know the time at which leaves were infiltrated (how many hours after exposure to light/daylight)?

We have stated this in the Materials and methods section. All experiments were performed in similar continuous light conditions (subsection “Transgenic marker lines and transient expression by Agroinfiltration”).

4) Results section"Beaks extend into longer stromules. Longer stromules were seen associated as just the tips of stromules bound to MTs or the full-length of the stromule extended along MTs. More complex stromule structures were observed associated with MTs." What does the “or” in the statement mean? While the stromule-tip-association suggests a good link to stromule extension, what happens when a stromule is totally aligned with a MT. Is this a coincidence or is there more to it? What are the more complex stromule structures?

We have adjusted this statement in the text to state “the full-length and the tip of the stromule”, since we have never seen the stromule extending along the MT without the tip also associated. We did not elaborate on these two observations, since we did not have any data suggesting they were entirely unique or a coincidence of the angle of the tip association to the body of the chloroplast. Nevertheless, there is a possibility that this is significant; we therefore feel it is still important to report here our observations. In response to the “more complex stromules”, these were described as kinked or branched stromules in the next sentence. We rewrote the text accordingly to be clear in the revised version in the first paragraph of the subsection “Stromules interact and extend along microtubules”.

5) 11% of stromules were not associated with MTs. The authors explain this by resorting to complete speculation and "MT-independent mechanism of stromule formation ". What are these other mechanisms and where is the evidence for them and their link to the observations presented here? If all stromule formation and extension is not based on MTs then I do not see the big discovery here. The observations presented then just become a biased subset that is chosen to make a point while anything that does not fit in is neglected.

Our intention was certainly not to provide a biased subset, but rather, to report the observations of stromules interacting with MT. Most induced stromules under observation (89%) are associated with MTs; but we do see that about 11% of the stromules are not. Our results are not completely speculative, since if there is no association with MT, it must be independent and that statement is merely factual. Furthermore, we have not ignored or neglected the other potential mechanisms for stromule extension and we hypothesized how our results fit in with observations reported by others in the discussion. In brief, we discussed the possibility that actin anchoring, associations with the endoplasmic reticulum or stromules originating from an internal force could control MT-independent mechanism. All of these may play various roles in stromule formation, but in Figure 1 we have focused on the MT interactions and describe other interactions with ER in Figure 2 and actin later in the manuscript. We would like to disagree that our findings are only significant if only 100% of stromules are associated with MTs. Often complex biological processes are not absolute, and we believe our study will significantly advance our understanding of the role of the cytoskeleton during stromule dynamics, formation and stabilization.

6) The authors wished to 'conclusively demonstrate that stromules extend along MTs' and therefore used two inhibitors to disrupt the MTs. Their evidence for MT depolymerization was that stromules disappeared after 15 minutes. How can a specific assay be used and the observations then used as evidence that the assay is working? I find the reasoning faulty and wonder if this not a case of circular reasoning?

We agree that using stromule disappearance, as evidence of MT depolymerization would be circular reasoning. However, we did not make that circular argument. Instead, we show that both APM and Oryzalin result in depolymerization of MTs after 15 min of treatment in Figure 3 and Video 5 and after 1 hour of 1µM of Oryzalin treatment in Figure 7—figure supplement 1. It must be noted that the prior studies (Kwok and Hanson, 2003; Natesan et al., 2009), used these compounds without examining the effects on MTs. Here, we have monitored the depolymerization using MT markers and time lapse microscopy. We restated our results to overcome this misunderstanding.

The stromules disappeared 15 minutes after APM and Oryzalin treatment: is this 15 minutes after pressure infiltration? Is it 15 minutes after the 4mm^2^ leaf excision was taken for viewing?

We clarified that it was 15 minutes after pressure infiltration of the treatments. The following text was added to the Materials and methods section, “All time points started immediately following the infiltration of the treatment.”

What does disappearance mean? Did they retract? Were they not formed? It must be made clear what disappearance means.

How stromules disappear during the treatment is an interesting question. We described our observations, but left the interpretation up to the reader. It appears that from our reported observations that it is a combination of both mechanisms and we stated this in the revised text. We do see a correlation between the depolymerization of a MT and the retraction of a stromule associated with it (Figure 3, bottom row; Video 5). Also, we see beaking that does not appear to continue into an elongated stromule (best observed in Video 5). To address this more directly, we counted the number of extension and retraction events after Oryzalin treatment and show that extension decreases and retraction increases (Figure 8), further supporting that both extension and retraction contribute to a disappearance of stromules.

7) The concentrations of inhibitors used are, in general, extremely high and would completely disrupt cellular functions, especially after pressure infiltrations. The authors provide minimal information on this by stating "The concentrations that resulted in the microtubule depolymerization/stabilization without any lethal effect on the leaves were used further for experiments." It must be made clear in the Materials and methods section what was done to ascertain that the cells were actually surviving and functioning normally after the harsh treatments.

We provided more details in the Materials and methods subsection “Inhibitor treatment”. Without any lethal effect means – leaf is not dying or showing any sign of microscopic abnormality. When a cell dies, we see increased autofluorescence in the tonoplast and nucleolus, and chloroplast swelling that eventually leads to rupture. We did not see these effects, but to make it more clear, we repeated Oryzalin inhibitor experiments using a lower concentration over a longer period of time. We also examined the actin microfilament network, which remained intact. Although MTs were not completely depolymerized under these conditions (Figure 7—figure supplement 1), these data were provided to address these concerns.

8) The VIGS assay creates a weak point in this study. The time frame for the experiments involving VIGS is well beyond the induction period for stromules. The researchers actually observed plants four days after VIGS and found numerous anomalies including significantly altered / bundled cortical MT organization in leaves. They also found twice the number of stromules compared to VIGS vector control. The conclusion of high stromules number after 4 days in NbGCP4-silenced plants is at best just a correlation; it is not a direct evidence suggesting a clear cause and effect relationship. Perhaps the greatly altered MTs changed a lot of other things that could then then also have affected stromule frequency. the plant was clearly no longer operating in a normal manner. The same reasoning applies to the further increase in stromule length and number upon expressing p50 subsequently. In this case the observations were taken 24 hours later. Could something more have happened to create a physiological change in the 24 hour period? What happened at 4, 8, 12 hrs? That is definitely sufficient time for stromule formation to have occurred (Brunkard et al., 2015).

We strongly feel that these experiments are a strong point of the study that provides genetic support for the role of MTs in stromule formation and extension; however, we are happy to rewrite this section to make our approach and results clear.

We altered MTs by silencing *GCP4*. This altered state of MTs induced stromules without any inducer, such as p50 treatment. We made the observations 4 days after silencing because VIGS takes about 3-10 days to knockdown the expression of a target gene. Since GCP4 is an essential gene, significant knockdown of this gene has effect on the plant phenotype. Hence, we choose to do our analyses after 4 days of silencing (~50% reduction in *GCP4* expression) because plants look similar to the VIGS control plants. At this time frame compared to the control, *GCP4* silencing alone is sufficient to induce stromules because the MTs are bundled in *GCP4* silenced plants. As we state in the manuscript, the stabilization of MTs is sufficient to induce stromules. This also correlates with our Taxol treatment data.

Our method of inducing a defense response is different than the Brunkard et al., 2015 paper, and their time points do not make sense in the context of the experiments that we have described here. We cannot make observations at 4, 8, 12 hours because we need to give enough time for p50 to express after infiltration. Therefore, we did our observation at earliest time point possible after infiltration of p50; i. e. 24 hours. Brunkard et al., 2015 did not use p50 as the inducer. Most of their studies are based on the isolated chloroplasts and not using intact leaves under a biological treatment.

9) There are some really obtuse statements:“Stromules appeared to interact statically, and not dynamically, with AFs, suggesting there are actin anchor points along stromules.” What is meant by 'interact statically and not dynamically'? Please explain the interactions. Is the word 'interaction' itself not denoting a dynamic phenomenon?

We apologize if our use of static and dynamic resulted in obtuse statements. Below we have defined our word usage, but if the reviewer believes different wording is better, we would be open and appreciative of constructive suggestions. However, we believe the description below will clarify our word choice:

The definition of static (adjective): “characterized by a lack of movement, animation, or progression”

The definition of dynamic (adjective): “marked by usually continuous and productive activity or change.”

The definition of interaction (noun): “a kind of action that occurs as two or more objects have an effect upon one another.”

Interaction does not denote a dynamic phenomenon on its own. We used interaction as a noun and static and dynamic as adjectives that qualify or modify the meaning of interaction. Similarly, we used interact as a verb, and statically and dynamically as adverbs. In this context, we use static to describe the interaction with AFs, since the interaction was stable along individual points along AFs and did not move. We use dynamic to describe the interaction with MTs, since the interaction point changed over time, resulting in movement or change in position.

Further, the stromules became thin at anchor points – “We observed a clear thinning or constriction at the site of stromule-to-actin interaction points along the length and across the body of the chloroplast". Did the actin anchor create this thinning? As I understand the authors are talking about an anchor point and not a band of F-actin. If not, then how is the thinning of a stromule explained?

These are interesting questions. We do not know if the actin anchor caused the thinning. The stromule thinning and the actin anchoring are correlated; stating that it is causal would be an over interpretation. Nevertheless, reporting this observation opens interesting questions into the mechanism, and we hope it incites future research. Our high-resolution imaging did not reveal a band or ring of F-actin and we hypothesize that instead a tension point from the interaction with AF is generated, causing the thinning. We currently added in that the mechanism for the thinning is unknown at the end of the second paragraph of the subsection “Actin filaments serve as anchor points but not as tracks for stromule extension”.

What is meant by “across the body of the chloroplast”? Are we still talking about thinning?

Here we see a thinning or constriction from one side to the opposite side of the chloroplast body, that forms a groove along the body of the chloroplast that correlates with actin microfilaments. We thank the reviewer for pointing out that this is unclear, and we provided a 3D model of grooves along the body of the chloroplast in Figure 6—figure supplement 2.

10) In a similar manner the authors state "At 8 min, the stromule began to re-extend along the MT, between 8 min and 9 min, the body of the chloroplast was released and this resulted in stromule driven chloroplast movement." What are the authors trying to convey? The body of the chloroplast was released! From what (?) and how did this result in a conclusion of 'stromule driven chloroplast movement'? The authors did not actually establish 'stromule driven movement'. Did the stromule pull the body or push it? Did the stromule elongate further or did it retract in order to drive chloroplast movement?

We have made this clear in the text and we have also added more data on stromule-driven movement (Figure 8; Figure 8—figure supplement 1).

In the described data we were following an individual chloroplast and associated stromule as an example. It is clearly defined – at 1 minute during the observation, stromule is extending; at 3 min it is retracting; but then at 8 min, it begins to re-extend. We hypothesize, that the chloroplast body was released from actin, but since we do not have direct evidence, it was not explicitly stated. We did not want to over interpret the data, however, we included this in the sixth paragraph of the Discussion. Also, we changed our wording to “chloroplast movement is correlated with stromule movement”.

Related, we now have analyzed the correlation between the angle of stromules and the angle of chloroplast movement. These comprise new, exciting data in this revision. We analyzed the correlation based on the stromule angle in the first frame and the subsequent angle of movement of chloroplasts. Using a circular correlation coefficient, r(FL), we were able to determine if it was more likely a pulling or pushing force. A positive r(FL) value would suggest a pulling force and a negative r(FL) value would be consistent with a pushing force. The value was 0.76, suggesting that the mechanism is more likely a pulling force. However, in the text we did not directly state that it is a pulling force, but rather, proposed that stromules could potentially provide the force for movement, or alternatively, they may just guide chloroplast movement. Since our data does not clearly support either mechanism, we called it stromule-directed movement, rather than stromule driven or stromule guided movement.

11) It is well observed that stromules arise in many different areas of a chloroplast randomly. Does this always coincide with chloroplast movement? The authors address this issue: "To examine how disruption of stromule and chloroplast AF anchoring effects chloroplast movement, we followed two chloroplasts in an epidermal cell treated with CytD." As I understand from the manuscript only one chloroplast was actually observed to reach the conclusion since the 1st one had restricted movement. Interestingly this singular observation is supposed to have added to the evidence that stromules may direct chloroplast movement in the absence of intact AFs.

In the previous version of the manuscript we presented three different examples of chloroplast movement correlated with the movement or orientation of a stromule in Figure 5, Figure 6 and 7. In the new manuscript, we analyze and quantified numerous stromule-directed chloroplast movement events and added in additional examples (Figure 6–Figure 9; Figure 8—figure supplement 1, Figure 8—figure supplement 2). Over half of chloroplast movement is stromule-directed and this is partially disrupted by partial disruption of MTs.

12) The weakest part of this manuscript is the proposed link to 'innate immunity'. This part of the study takes the manuscript into a highly speculative mode where the nice demonstrations of microtubule alignments and the presence of actin are undone by a lack of rigorous experiments. The authors point to a feature of innate immunity as 'eventual peri-nuclear clustering of chloroplasts'. It is notable that chloroplast clustering around nuclei happens routinely in leaves, even without bringing innate immunity into play. In the excised 4mm^2^ sections what is the baseline number of chloroplasts clumping together and how is it accounted for by the idea of innate immunity? Perhaps the wounding of tissue is responsible for innate immunity? This still does not explain the chloroplasts around nuclei in normal, unwounded (or in any untreated tissue). Clearly stromules have been observed by the authors under conditions that trigger the innate immunity but I fail to see an exclusive response where a cause and effect relationship is indicated with regard to stromules.

The proposed link to innate immunity builds upon our prior work (Caplan et al., 2015; Caplan et al., 2008) and we do not believe that it is highly speculative. We improved our description of how this connects to our prior work and progresses our understanding of how and why chloroplasts cluster around nuclei during plant innate immunity. We agree that chloroplasts are seen around nuclei in untreated mock tissue and present this data in the new Figure 10. Mock tissue has nuclei with chloroplasts clustered, but this increases during plant innate immunity. Similarly, chloroplasts often have stromules, and they increase during plant innate immunity as we have shown previously (Caplan et al., 2015). We have presented a working model in Caplan et al., (2015) that stromules and perinuclear clustering acts to amplify innate immunity signaling. Neither can induce a defense response on its own, but constitutive stromule induction can accelerate HR-PCD. In the revised version, we provided full microscopic images that we used to count the perinculear clustering; they clearly show that many chloroplasts cluster near nuclei in the presence of p50 (Figure 10). We used appropriate controls to show that this is specifically induced during p50 response. We are not using the wounded tissue here; all our observations are with tissue away from the primary infiltration point. Furthermore, we have clearly shown in our published paper (Caplan et al., 2015), that stromules are specifically induced during p50 response in *NN* plants but not in *nn* plants.

Some references are missing and should be added. Reference for the presence of chloroplasts in tobacco pavement cells is needed.

Only one example has been provided, making it difficult to fully respond to the concern. To our knowledge, we have referred to all the relevant references except for the one provided by the reviewer, which we believe may not be required. A recent paper published by a group of leading plant cell biologists (Barton et al., 2016) clearly states that it generally well accepted that *Nicotiana* pavement cells have chloroplasts. Interestingly, that same manuscript reports that there are multiple lines of evidence that Arabidopsis also contains chloroplasts in epidermal pavement cells, even though a recent stromule paper (Brunkard et al., 2015) refers to these plastids as leucoplasts.

The Introduction as well as discussion of earlier work on actin and microtubules must be done more thoroughly rather than be given a cursory mention. The authors must clearly state how their results advance the field and are not a repetition of earlier inhibitor experiments with a rather speculative link to chloroplast movement and plant immunity.The authors have only looked at chloroplasts in pavement cells. For this study to have a real impact the observations must take into account other kinds of plastids, including plastids in mesophyll tissue and leucoplasts, all of which are known to display stromules.

We have described and referred to the previous papers in the Discussion. We explain how our data compares to prior inhibitor studies (there are only two primary papers and others are reviews) and how our work advances our understanding of stromule formation and dynamics. We have already shown in our recent paper (Caplan et al., 2015) that stromules are induced in mesophyll plastids. We have focused on epidermal cells for consistency and because stromules and their dynamics can be more accurately measured in epidermal cells compared to mesophyll cells, which have more tightly packed chloroplasts. Examining stromules also in tissues containing leucoplasts, such as roots, would require an entire replicate of our experimentation, which we believe is beyond the scope of this manuscript. Overall, we anticipate that our initial study will open exciting new avenues of research.

Reviewer #2:[…] This work provides new and a significant information about the formation/regulation and function of the understudied plastid stromules. Although one can argue that the work it is generally based on correlative results, I consider that this data will be a keystone for future studies in this topic. I think that the manuscript fits with eLife's scope. However, I have some concerns about how the results are presented, described and its interpretation. Also, I consider that authors should support with more data the proposed model. Particularly, authors should address two things:- The authors use the terms associated, bound, touching, interacting, etc. as "synonyms" but without specifying what they exactly mean. They should define these terms in a "precise" stromules-MTs distance(s) manner. Indeed, I do not think authors demonstrated interaction in any case. Authors should do 3D co-localization analyzes. In addition, they should consider doing FRET analysis if this is possible.

We rewrote and restated the text to remove the terms associated and touching. For consistency we used the more general term interacting throughout the manuscript. We only used bound to describe static interactions with actin. Stromule tips do co-localize with MTs and AFs in 3D and the majority of our analysis is generated from 3D confocal microscopy z-stacks over time. We apologize if that was not clear in text. However, confocal microscopy cannot provide a “precise” stromule-MT distance, since it lacks the required resolution. We have included much higher resolution transmission electron micrographs of MT in close proximity (nanometers) to stromules (Figure 1—figure supplement 1).

FRET analysis is currently not possible. FRET requires a direct protein-protein interaction, and we do not know what proteins on the stromule surface interact with MT directly or MT associated proteins (MAPs or motors) indirectly. We thank the reviewer for the suggestion, but without the identity of these molecular components, we cannot conduct this experiment at this time, but we will consider them in the future.

- Authors should analyze the dynamics changes of MTs and AFs networks after p50 defense induction. Are MTs stabilized after p50 expression? Are perinuclear AFs increased after p50? These results could be of great support for the proposed model. I believe the authors could do both analysis using the data that they already have.

These are excellent questions and we included these data on dynamic changes of MTs and AF networks during a p50-induced defense in the revised manuscript (Figure 5). We examined perinuclear clustering, but it was difficult to quantify. Instead, we disrupted AFs using CTD and observed a decrease in perinuclear clustering of chloroplasts following p50 defense induction (Figure 10).

The authors should also address the following comments:Abstract:1) Misspelling: effects should be affect.

Changed in the Abstract.

2) Authors should revise and change the statement suggesting that AFs have no role in stromule extension. I think that providing "…anchor points for stromules to prevent their retraction while extending on MTs…" sounds like an important role.

Modified in the Abstract: "Although actin filaments (AFs) are not required for stromule extension, they provide anchor points for stromules to prevent their retraction while extending on MTs."

Results:1) Section "Microtubules are required for stromule extension". Authors should consider that the drugs used inhibit the increase of stromules per plastid associated to mock treatment instead of reducing them.

Changed accordingly.

2) Section "Microtubule stabilization increases stromule number, length and stability", first paragraph: What are authors considering as MTs stabilization in Figure 2? The image showing cells treated with paclitaxel should be label indicating this MTs stabilization. Are stromules longer in cells treated with paclitaxel?

Paclitaxel has been used extensively for MT stabilization and cancer research (>30,000 publications on Pubmed), and therefore, we did not characterize it here. After Paclitaxel treatment, we observed longer stromules and multiple stromules emanating from individual chloroplasts. We clarified that description on (subsection “Changes in microtubule organization increases stromule number, length and stability”, first paragraph, see Figure 3). However, we chose to focus our extensive characterization of MTs and stromules in *NbGCP4*-silenced plants that have altered MTs. In those experiments, we examined both the effects on MTs and stromules, including stromule length (Figure 4–Figure 5). Since, it is difficult to directly ascribe the changes in stromules to MT stabilization, the section header was changed to: “Changes in microtubule organization increases stromule number, length and stability.”

3) Section "Microtubule stabilization increases stromule number, length and stability". Figure 3: Authors point out that the stromules number increase observed in NbGCP4-silenced and in p50-induced plants are "remarkably similar". They should analyze the MTs stability during p50 induction to support this statement.

We have provided new data in Figure 5 (see subsection “Changes in microtubule organization increases stromule number, length and stability”, last paragraph). In summary, p50-induced immunity results in similar changes as *NbGCP4*-silenced plants.

4) Section "Microtubule stabilization increases stromule number, length and stability". Figure 3: Authors should describe and discuss the extension/retraction results obtained after treatment with p50.

See new data in Figure 5 (see subsection “Changes in microtubule organization increases stromule number, length and stability”, last paragraph).

5) Section "Stromules direct chloroplast movement in the absence of actin filaments", second paragraph: Although authors mention that exist "opposing forces" and a "rapid pulling" provided by stromules, they do not have any measure of these traction forces. Authors should be cautious with results interpretation and re-phrase this paragraph.

We modified the text accordingly. We also added this statement: “Our data show that stromules may direct chloroplast movement in epidermal pavement cells; however, it remains unknown if stromules provide a driving force or only guide chloroplast movement.”

6) Section "Actin microfilaments mediate perinuclear chloroplast clustering during plant immune response", second paragraph: authors should replace the sentence "…connections, but were unable to find…" by "…connections, but we were unable to find…"

We have changed this in the first paragraph of the subsection “Actin microfilaments mediate perinuclear chloroplast clustering during plant immune response”.

Discussion:1) Third paragraph: "…and propose the difference…" should be replace by "…and propose that the difference…"

We have changed this in the third paragraph of the Discussion.

2) Third paragraph: What do the authors mean with the statement "directly interacted"? I do not think they showed any interaction.

We removed the word “directly”.

Figures:1) Figure 1: How authors define tip connection or extension along MTs? How close are stromules and MTs? Are authors using z-stacks projections to analyze this distance? Authors should clarify this point in every image/quantification where an interaction/connection/touching, etc. suggestion is made.

The following sentences were modified: “These observations were made in maximum intensity projections of z-stacks generated by confocal microscopy….”, and: “A stromule-to-MT interaction was designated if these two structures were overlapping or not resolvable by confocal microscopy. However, since the resolution of confocal microscopy is relatively low, we verified the close interaction between stromules and MTs using transmission electron microscopy (TEM).”

2) Figure 1: Are these analyses using only z-stack projections or they consider 3D/4D dimensions? Figure 1: Is the algorithm calculating the 3D distance between stromules and MTs?

The algorithm does not detect 3D distances. In the last paragraph of the subsection “Stromules interact and extend along microtubules”, we added that the algorithm was used on maximum intensity projections.

3) Figure 2: Authors should show if exist statistical differences between times/treatments.

Figure 2 in old version is now in Figure 3. We have tested multiple comparisons across the time and treatments. There is no other statistically significant difference.

4) Figure 2: What do the authors consider as stabilization of MTs in this image? They should do an analysis similar to the Figure 3.

Please see point 2 (Results) above.

5) Figure 2: Is there a positive correlation between the increase on stromules number and the stromules "associated" to MTs?

We did not measure if there was a positive correlation, mainly due to the caveat that it is difficult to quantify if they are “associated” by confocal microscopy because of the resolution limit of the technique.

6) Figure 5: Authors should add the separated MTs and AFs channels to this figure.

Due to space limitations, we chose to provide a diagrammatical representation rather than showing all channels separately. However, if the reviewer still believes that separated channels would provide additional information, we can provide it as a supplemental figure.

7) Figure 3—figure supplement 2: Authors should add stats to this quantification. They should also consider using this data as a main figure.

We added in the total number stromules measured and the statistical significance if it existed. We agree with the reviewer that there is enough important data presented to be a main figure. However, since this was from the same experimental data set and just additional quantifications of the same data, we chose to keep it as a supplement to this figure.

Reviewer #3:[…] The Introduction and Discussion of the paper are well written, touching on the key literature. The microscopy is beautiful and sophisticated analyses are performed on some of the image data sets using tools that other plant biologists will be interested in knowing about and perhaps applying. A strength of the study are the time series images showing evidence for guidance of stromule dynamics by cortical microtubules, this result was nicely controlled for by using multiple markers for microtubules, making the possibility of an aberrant interaction between stromules and an overexpressed microtubule marker unlikely. However, there are issues with data interpretation that need to be addressed and I felt that a number of major and secondary conclusions were not supported well by the data presented.A) The authors test the role of microtubules in stromule maintenance by applying depolymerizing and stabilizing drugs and by manipulating microtubule organization by genetic knock down of a nucleation complex protein. Previous studies had indicated that microtubules play a relatively minor role in stromule abundance, but here microtubule depolymerization resulted in a significant reduction in stromules per plastid compared to the mock control. This was the strongest section of the paper but I have a few comments.1) The following point is not a major criticism, but an odd thing with the drug experiments was that stromules significantly increased in abundance in the mock control over 15 minutes. The reduction in stromule numbers with drug treatment observed at the same time were only slightly lower than the numbers observed at time 0-5 minutes after treatment. If just stromule numbers at time zero and 15 minutes were compared, one would not infer a strong effect of microtubules. The reason for the increase in stromule number in the mock is neither explored nor explained. A couple of possibilities are that stromule formation may be stimulated by stress caused by placing the tissue in the microscope slide mount, or perhaps that light used for the imaging stimulates stromule formation. Since it is an effect on the increased stromule numbers that is observed with drug treatment, it would be nice to have bit more insight into why these number are elevated over the course of the experiment. The two possibilities mentioned could be tested by not imaging at time point 1, and by determining if media exchange during imaging can mitigate the induced increase in stromules over time in the slide mount.

We apologize for not explaining the induction with the mock treatment. The only difference between the media control and the mock treatment is the 0.2% DMSO from the inhibitor stocks. The mock treatment (often referred to as the vehicle control in other fields) with DMSO is the proper control, but since we saw a difference with our additional media control, we felt scientifically obligated to report these results. Stromules can be induced by a wide variety of stimuli. DMSO induced stromules, but nonetheless, these stromules were disrupted by the drug treatments.

2) It was a good idea to manipulate microtubules by a secondary means, other than drug treatment. However, here the authors conclude that altered microtubule dynamics were responsible for observed reduction in stromule extension and retraction rates. Since microtubule dynamics were not measured, nor were observations shown of possible interaction between stromule tips and microtubule ends, it is not clear why this conclusion was reached. What is evident is that microtubule organization is altered in the GCP4 knock down, with an increase in parallel organization of microtubules in leaf epidermal cells, a result that has been shown previously for knockdown of GCP4 and also for GCP-WD/NEDD1. While the results are striking, it is not clear why a change in polymer organization would drive a change in stromule dynamics. My suggestion would be to keep the interpretation open and not ascribe it to microtubule dynamics. It is possible that the effect is indirect. For example, by altering levels of free MAPs due to reduced total lattice in these cells.

We agree with the reviewer and modified the text to keep the interpretation open and did not ascribe it directly to microtubule dynamics. We appreciate these thoughtful comments and changed the text accordingly.

3) The authors conclude that microtubules play a direct role regulating stromules. While agree that the data shown are consistent with that idea, it is possible that the relationship is less direct, as the authors touch on in their Discussion when considering the ER and membrane contact sites.

We adjusted our text to state that our data is consistent with microtubules playing a role regulating stromules. Thank you for the suggestion. We have also observed stromules pushing through the ER network while extending along MTs. We added Figure 2 and two videos (Video 3–Video 4) showing these observations and described these data in the subsection “Stromule extensions through the ER is directed by microtubules”.

4) The tip tracking method is very nice. It is kind of a shame though that it is not used to more rigorously quantify stromule tracking of microtubules. Only one example of such tracking is shown, with no summary measurements made over many cells, something that could be done with such a method. A further point here is that it should be defined more clearly how "on tube" vs. "not on tube" is determined, since the underlying molecular relationships are sub-resolution.

Our algorithm worked well on MTs marked with EB1; however, our peers recommended that we do not use EB1 and instead conduct our studies using the GFP-TUA6 line. However, our algorithm does not work with GFP-TUA6 because of free tubulin masking clear detection of MTs. Related, quantitative data using EB1 might be misleading, since the overexpression could change the dynamics. The quantitative data on stromules and interactions of stromules with MTs was conducted manually. Despite these caveats, we believe that these data provide a nice illustration of chloroplast movement along MTs and more clearly defined that the “on tube” were points in which the tip colocalized with the skeletonized MTs.

5) The SOACs analysis is used to assess features of polymer orientation, curvature and length. While this analysis produced a nice color-coded image of the orientation of segmented features, these data were not used to quantitatively assess patterns of polymer orientation, which again, seems a missed opportunity to show robustness of observations over many cells and samples. A second point, and an important one, is that by inspection of the raw data and the processed data, the method is not an appropriate means to assess polymer length, and those data should be dropped. It is not possible with such segmentation to tell if extended structures are composed a few long polymers or many shorter polymers, nor can it tease apart what happens were polymers meet and overlap at angles. Determining the length distribution in these arrays is an important goal in the field, but it is not trivial.

Although these quantitative comparative measurements of orientation were not made in the original work describing, SOAX, and only a visual output was provided. However, we developed a method based on the mean resultant length of MTs to quantify these images. The mean resultant length was calculated by converting the MT angles into individual vectors, adding the vectors together, and calculating the mean. MRL values are between 0 and 1, with 0 indicating that MT angles are random and 1 indicating all MT angles are the same and completely aligned.

We agree with the reviewer that the SOAX analysis cannot accurately assess polymer length due to unknowns that are not resolvable by confocal microscopy. This is not a characteristic that is easily measured, although we feel that our quantitative measurement, albeit indirect, demonstrates a method for measuring changes in microtubules. If the reviewer still insists, we will remove these data, however we hope the reviewer agrees there is some value to these data if more carefully described as potentially being a few long polymers or many short polymers. We added in the following statement: “The snake length is not a direct measurement of MT length, since this approach cannot accurately distinguish between two MTs that are bundled together. Nonetheless, this measurement further suggests that silencing NbGCP4 alters MTs.”

6) It is stated that GCP4 knock down results in greater bundling. This is not determined in the present manuscript.

The reviewer is correct that we did not determine that GCP4 knock down results in greater bundling. Those conclusions were drawn from Kong et al. (2010) in which they state that amiR GCP4 results in bundling in Arabidopsis. We apologize if our text suggested that we showed this directly here, and we more carefully cited the prior research that knockdown of GCP4 results in bundling. We also removed the term “bundling” from the description of our work in the Results and left that speculation only in the Discussion.

B) The authors conclude that actin filaments do not extend along actin filaments but that actin filaments do serve as static anchor points for stromules, and also prevent stromule retraction.1) The data for stromule extension along actin cables are not quantified, but presented as example images and a video. These images show little evidence for stromule extension along actin bundles, but to be robust, a method should be used to census the relationship of stromules and actin structures so that data from multiple cells and tissue samples can be assessed. Even so, it should also be at least discussed that actin features may be present that did not label with the probes used.

We agree with the reviewer that our data do not show stromules extending along actin bundles, and the best example when examined closely, did not appear to be “on actin” but simply close by (Figure 6—figure supplement 1). We made that clearer in the text. In general, we almost never saw stromules extending along actin microfilaments and hence did not see a need to quantify that negative data. We also agree that mTalin and LifeAct may not show all the actin and added that to the second paragraph of the Discussion. We chose to use two different markers, since they each have their caveats, but the reviewer is correct that neither perfectly labels all actin.

2) In the CytD experiments, why didn't stromule number increase from 0 to 15 minutes in the mock as was observed for the microtubule drug experiments? If this effect does not occur in every experimental setup, the microtubule drug experiments should be repeated under conditions where stromule number does not go up in the mock.

As described above, this is most likely due to the DMSO concentration. We repeated these experiments with much lower concentrations of DMSO and drug treatments. We also added in additional quantifications of length, extension and retraction events and velocity, stromule-directed movement and movement type (constant and sudden) following Oryzalin or Cytochalasin D treatments (Figure 7, Figure 8).

3) The evidence of static anchor points at actin filaments is weakly presented. A single kymograph is shown for analysis of stalling during stromule retraction, and in this single analysis it is not clear that the stall actually occurs at the position of actin signal. The signal corresponding to the stromule tip is offset from the actin signal. A much more rigorous analysis of retraction and actin overlap sites is required.

We regret that the image we chose does not show the retraction clearly. Since this is a new assay we developed, it may not have been completely understood. However, it is clear from the depth of this critique, that this reviewer is indeed an expert on cytoskeleton dynamics; so if our point was missed by the reviewer there is a good chance it will be missed by others. Previously, we showed just one image of a stromule partially retracted (hence, the tip was offset from the signal). In the revision, we show three time points in which it was fully extended, kinked/partially retracted, and retracted back to an actin microfilament (Figure 6). We also show retraction back to an actin microfilament in Figure 8. We rectified that misunderstanding by showing montage of these retraction events. To further show these data, we added in three additional representative examples in Figure 6—figure supplement 1.

4) Likewise, the image sequence in Figure 5 and corresponding video are not convincing of retraction back to an actin-defined anchor point – the stromule tip appears to overshoot the "kink" position at the actin branch site. This is also a single image sequence example.

We provided more examples of the retraction events to actin anchor points (Figure 6—figure supplement 1). Our retraction assay was our way to avoid the over interpretation that a simple intersection between a stromule and actin is an anchor point. Unlike, the interaction with MTs, which has a motion trajectory that matches the path of a MT, the interaction with actin is static and is more difficult to quantify. However, we did try to quantify this by looking at retraction events indirectly.

5) The analysis in Figure 5 looks more quantitative, but it is a less direct measure of interaction. Rather, it is measuring a predicted consequence if the proposed interactions exist. It is also poorly explained and it took some time to determine what the graph represents. I think the second bar was meant to be labeled "not fully retracted", rather than "actin anchored', which is a conclusion of function rather than an observation.

We changed the bar labels to “full retraction and “partial retraction” and we thank the reviewer for this suggestion. The reviewer is correct that this is an indirect interaction, but we currently do not have the tools to quantify it in a more direct way. It is indeed a consequence of what would occur if an actin anchor exists and is somewhat predictive. We described this assay more carefully. As I’m sure the reviewer is aware, it is difficult to quantify these interactions without knowing the proteins involved in the interactions.

C) The authors conclude that loss of actin anchoring allows stromules to direct plastid movements.1) Once again, these data consist of example images, not more global measurements across cells and tissue samples. In fact, just one image series example. It is a beautiful video, but it cannot be determined from this one example how generalized the phenomena shown are, nor how strong is the correlation of plastid movement with the direction stromule extension. A suggested means of analysis would be to assess each plastid displacement of x distance or greater in a set of time series acquired from multiple cells and leaves. For each displacement, stromule position and orientation would be determined and an orientational resultant would be estimated. The relationship between the predicted resultant angle of "pull" and the actual direction of movement could be plotted and analyzed.

The data was a representative time lapse from multiple observations, and we thank the reviewer for clear guidance on how to quantify the relationship of the movement and we have now quantified these data. Excitingly, we found that stromule-directed movement is a more general phenomenon and does not require actin disruption. We hinted at this in our first submission in Figure 5 that showed dynamics with MTs and AFs, and here we quantified the movement. As suggested, we calculated the orientation angles for the initial stromule angle that would “pull” and the actual direction of chloroplast movement. We used a circular correlation coefficient to show that the angles are indeed correlated and not randomly similar. We plotted the relationship directly as an XY scatterplot as well as looking at the difference between the two angles. We also show that Oryzalin treatment partially disrupts stromule directed movement. We were hesitant to call it “pulling” since it is unclear from our data if the stromule provides a pulling force or just guides the movement.

2) A partial loss of actin structure by the CytB treatment is somewhat unsatisfying. The authors chose not to use LatB, which is more effective, because MT organization is also affected. However, I think these data should still be shown to ask if the observed movements of plastids are still seen when actin is more completely disrupted. It will be evident if stromule interaction with MTs (MT-dependent interaction sites) is also affected.

We repeated our CytD treatment experiment and quantify stromule movement as suggested. Using a lower concentration of Cytochalasin D over a longer period of time fully disrupted AFs, but did not disrupt MTs (Figure 7—figure supplement 1). Complete disruption of AFs resulted in a complete loss of chloroplast movement, including stromule-directed movement (Figure 8). However, stromules still persisted and extension was more constant and less sudden with slower extension and retraction rates (Figure 7).

D) The authors conclude that actin anchoring mediate plastid accumulation at cell nuclei during the innate immune response and suggest a model by which plastid accumulation is facilitated by plastid movement driven microtubule- and stromule-based plastid motility, followed by actin mediated capture at the nucleus.Another case of data by limited example. While the example shown in Figure 7 is compelling, more than this is needed. I was surprised that the hypothesis was not tested by actin disruption, for example. Further, it should be asked if plastid motility, and nuclear accumulation are significantly reduced if microtubules are depolymerized.

We thank the reviewer for this suggestion and we disrupted MTs with Oryzalin and AFs with Cytochalasin D. Oryzalin treatment only partially disrupts stromules, and this level of disruption had no effect on perinuclear clustering. Interestingly, Cytochalasin D significantly reduced perinuclear clustering (Figure 10). Although this supports our hypothesis, the data should be cautiously interpreted since our data suggests that actin plays multiple roles in this process, including interactions with the body of the chloroplast controlling positioning, along the stromule and with nuclei.

E) In general, the authors should be careful about distinguishing conclusion and interpretation from observation. This issue occurs throughout the manuscript.

We thank the reviewer for this suggestion and we adjusted the text to more accurately distinguish between conclusions and interpretation. As we hope the reviewer can gauge from our responses, we do have a clear understanding of the difference, but that was not conveyed properly in the text. We believe the changes to the text and the new data have corrected this in the revision.

[Editors’ note: the author responses to the re-review follow.]

As you can see from the detailed reviews below, the reviewers very much appreciate your efforts to improve the initial manuscript. Yet they feel some additional adjustments are required before the paper can be published. These are mostly of organizational/semantic nature, and do not require additional experimentation. Apart from answering point by point to the reviews below, please pay particular attention to the following:1) The reviewers are of the opinion that the analysis of retraction pausing with sites of actin filaments needs to be strengthened quantitatively. Currently, the data consist of example image sequences and an indirect measure involving pausing frequency, which is not explained very clearly. E.g., it is not obvious that this experiment correlates pausing with sites of possible actin action.

The previous analysis was a quantification of retracting stromules specifically pausing at AFs. We apologize for not describing this clearly. We have rewritten the description of the assay and results in the text and believe that the new description is much clearer. We also added in additional quantitative data to show that the majority of stromules that paused during retraction had a retracting tip colocalized with an AF, suggesting that these are possible sites of actin anchoring. Only a small fraction of retracting stromules (5.7% of the total) paused and did not have a stromule tip colocalizing with AFs. We have presented the data in Figure 6 and description in the second paragraph of the subsection “Actin filaments serve as anchor points but not as tracks for stromule extension”.

2) The reviewers believe that it would be a good idea to quantify tracking of microtubules by stromule tips rather than to show one example of such tracking.

In response to this, we manually tracked and quantified the velocity of stromules tips extending along EB1-Citrine- and GFP-TUA6-marked microtubules. We are still hesitant to show more EB1 data, since this is not an ideal marker for microtubules and do not want to mislead others into quantifying stromule dynamics using EB1. Instead, we show here that the rate of stromule extension is significantly lower with EB1, compared to GFP-TUA6 or stromules extending with no MT marker. These quantitative data are presented in Figure 1 and corresponding text in the last paragraph of the subsection “Stromules interact and extend along microtubules”.

3) Clarification is needed about how the circular correlation coefficient data relate to the results with the 30 degree cutoff angle. These data are new to the revised manuscript and need to be explained more thoroughly.

The main difference between the circular correlation coefficient data and the analysis using a 30-degree cutoff angle is that the circular correlation coefficient requires paired stromule angles and chloroplast movement angles. Therefore, only chloroplasts with stromules can be analyzed. If we restrict the circular correlation analysis to only paired movement events with a 30-degree angle difference, the r(FL) value is 0.95. A r(FL) value of 0.95 suggests angles are highly correlated and this is why a 30-degree difference was chosen. Using this 30-degree cutoff, we were able to examine all movement events, including chloroplasts that have no stromules. We have included this in the second paragraph of the subsection “Stromules direct chloroplast movement”.

Reviewer #2:[…] Nevertheless, one caveat is that authors are not showing quantitative data on the stromules tip tracking. The authors argue that the algorithm only works for the overexpressing EB1 construct, which could be changing the general MTs dynamics. I think this could also be true for the other OE constructs. Thus, authors should consider adding tip tracking experiment quantifications, and show the reproducibility for this method, at least for EB1.

We have now quantified the velocity of stromules tips extending along MTs. We did not use the algorithm for the tracking, but rather manually tracked the stromules in imageJ, and removed the single instance of tracking with the algorithm. We are still hesitant to add in more EB1-Citrine data in this manuscript, since it significantly decreased the speed of stromule extension (new data in Figure 1). To make the limitations of the algorithm clear and to emphasize all of the data in the manuscript was quantified by manual tracking, we added in the following statement: “The algorithm only accurately detected the slower moving motion when EB1-Citrine was used as a MT marker, and therefore, was not used in other experiments. The length, velocities, extension and retraction frequencies, and types of motion were quantified manually in all other experiments.”

Furthermore, we agree with the reviewer that any overexpression construct for marking MTs may change MT dynamics. Consequently, we quantified changes in stromule dynamics in the MT inhibitor and *NbGCP4*-silencing experiments without MT overexpression markers.

Reviewer #3:[…] Point-by-point responses to authors' replies to initial review.Point A1: In my first read, I missed the fact that DMSO treatment was associated with an increase in stromule number over time. This is perhaps because the media control was not shown in the figure for comparison. The revised text makes it more clear that DMSO causes an increase in stromule number over the media control, and that drug treatment reduces this increase. However, I am now wondering why the media controls are not shown in the figure. It would be useful to know if drug treatment also causes a reduction of stromules in the media control, even if there is no increase in time otherwise.

We added in the infiltration media alone control to the graph. Most reports just show the vehicle control (DMSO in this case), but we agree with the reviewer that showing the media control would be useful for others in the field; we added this into Figure 3. It is important to report that DMSO can induce stromules, and when conducting any drug treatment assays, the concentration of DMSO must be the same for all treatments. It is not the focus of this work, but we do believe it is important to show this effect. However, the DMSO treatment is just media with the same amount of DMSO required to do the drug treatment. The requested drug treatment in media control cannot be included, since DMSO is required to solubilize the drugs.

If not, does this mean that induced stromules are somehow different from existing stromules in regard to microtubule depolymerization?

This is an interesting question, and more generally, are there different types of stromules? Are stromules induced during plant innate immunity the same as stromules induced during other stimuli, such as DMSO? We thank the reviewer for the question and added in the following statement in the Discussion: “In general, the formation of stromules may vary based on differences in cell, plastid, or stimulus type.”

Point A2: The revised text addresses the concern raised about ascribing the changes in stromule dynamics to changes specifically in microtubule dynamics. However, I still think it would be appropriate to make clear that, while the manipulation here is pretty specific, the observed changes in stromule dynamics may be either a direct or indirect consequence of changes in microtubule organization. For example, the statement – "Collectively, these results indicate that change in MT organization in NbGCP4-silenced plants or during p50 induced immunity control stromule dynamics." – is stronger than I think the data warrant and should be restated. "Collectively, these results indicate that changes in MT organization caused by NbGCP4-silencing plants or during p50 induced immunity are correlated with changes in stromule dynamics, indicating a possible role, direct or indirect, for MT organization in modulating stromule dynamics."

In the previous revision, we made numerous changes to the text to emphasize that these events are correlated. We thank the reviewer for finding this statement that we missed. We changed it to “Collectively, these results indicate that changes in MT organization caused by NbGCP4-silencing plants or during p50 induced immunity are correlated with changes in stromule dynamics, indicating a possible role, direct or indirect, for MT organization in modulating stromule dynamics."

Point A4: Thanks for the clarification. I agree that the figure is useful for portraying a sense of dynamics in a static format. The original text did not make it clear that the figure was produced manually. Unfortunately, this is also not clear in the revised text. The manual method is described in the Materials and methods, but the text and figure legend do not indicate how this figure panel was made. I think a simple edit to the figure legend would fix this.

We edited the text and figure to make it clear that only the depiction of the motion in Figure 1 was conducted with the algorithm and all the quantitative data was collected using manual tracking. Please see our response to reviewer #2 for a detailed description.

Point B1: I think it is important to give a sense of how much space was searched through to say "almost never". For example, you could state that "at least x stromules were examined in at least y cells, and only n stromules showed any evidence of extending along actin filaments. Even in these cases…"

We thank the reviewer for this suggestion, and we added: “Out of 73 stromule tip extension events from 34 cells, the vast majority (93%) of stromule tip extensions were not observed along AFs.”

Point B3: See comment for 5 below.Point B4: See comment for 5 below.Point B5: While more examples of stromules retracting to locations proximate to actin filaments, Figure 6 provides the only quantitation of this behavior. As indicated in the first review, it took me a while to figure out what I thought this panel represented. In reviewing this experiment again, I confess that I still find it unclear what exactly was measured here. The bar graph shows the percentage of fully retracted vs. partially retracted stromules. The figure legend states: "The percent of stromules containing actin anchors was quantified by monitoring retraction events in the samples described in panel A. Retracting stromules without actin anchors resulted in full retraction." However, in the text it states "70.3% + 0.02% (SEM) of retracting stromules partially retracted to one or more AF." So, was just the percent of partial retraction measured, or were retraction pause events also correlated with actin signal? If the latter, I might expect to see an analysis of how often retraction pausing was observed at sites of actin signal, but these data are not shown. If all partial retractions are indeed caused by actin anchors, then the percent of partial retraction alone would serve as an indirect estimate of the prevalence of stromules with actin anchors. However, is this known? Can stromules pause for other reasons? If they can, this experiment does not really probe prevalence of actin anchors well, nor does it test for the role of actin in pausing.

We apologize that our description of this experiment was not clear. We repeated the analysis of these data and rewrote the description. The repeat of the analysis was spurred by the reviewer’s question, “can stromules pause for other reasons?” We originally, did not quantify this because it was a rare event, but it is an excellent question. Therefore, we quantified how often stromules tips paused during retraction without the tip colocalizing with an AF and found that 5.7% of stromule retractions show this behavior. It is unclear if these pause events are at AFs that are not labeled (common problem with AF markers) or if it is pausing at some other interaction point. Nonetheless, these data further emphasize that the majority of partial retraction events show a paused stromule tip colocalized with AF signal. We no longer describe it as a “partial retraction”, since this ambiguous descriptor may be the point of confusion. Instead, we describe it more accurately as “retracting stromule tips paused for multiple, consecutive frames and showed colocalization with an AF.” See the second paragraph of the subsection “Actin filaments serve as anchor points but not as tracks for stromule extension” and Figure 6.

Point C1: The new data measuring the correlation of stromule angle with movement angle are an important addition to the manuscript. The angular correlation plot in the supplementary data is actually stunning. I would put it in the main figure. However, I think it should be stated more clearly how plastids were selected for this analysis. Were just plastids with one stromule selected? If so, how were they chosen?

We chose to show the histogram data in the main figure because this was used to define the 30 degree difference for stromule directed movement classification, and therefore, leads into the next graph of the that figure. Therefore, we believe it should be left in the main figure. However, we also thought the angular correlation plot was quite striking and visually showed how correlated the stromule angle and the chloroplast movement angle were in these data. To clarify how plastids were chosen, we added in the following statement: “Only chloroplasts containing one or more stromules were used for this analysis because it depends on comparing paired measurements of the angle of the stromule from the chloroplast body attachment point to the tip and the angle of chloroplast movement.”

Also, a significant question is raised by the oryzalin experiments. In oryzalin treated cells, the circular correlation coefficient of stromule angle with the direction of plastid movement actually appears to go up (.85 vs.76 for no oryzalin). Yet when a 30 degree difference in stromule and movement angles is established as a cutoff for positive stromule directed movement, it is concluded that oryzalin treatment reduces stromule directed movement. It is hard for me to see how these two results are reconciled. If the correlation coefficient alone is considered, one would not conclude that oryzalin treatment reduces stromule directed movement in fact, it may even increase it.

We examined the difference in r(FL) values for DMSO and ORY are found they are not significant. To examine this, we calculated the standard errors of the r(FL) values using a jackknife method. A two-sample t-test shows that the r(FL) values for DMSO and ORY were not significantly different from each other (P=0.52), and consequently, this data is not in conflict with the data using the 30 degree cutoff. We chose the 30 degree cutoff because when using that cutoff, the angles were highly correlated with an r(FL) value of 0.95. The circular correlation analysis requires that we only analyze chloroplasts with stromules. However, we can use the 30 degree cutoff to examine chloroplast movement events with and without stromules. When this analysis is conducted, chloroplasts still move when microtubules are disrupted with ORY, but there is less stromule directed movement. See the second paragraph of the subsection “Stromules direct chloroplast movement”.

Point D1: It seems that the new drug tests during perinuclear clustering were useful. I agree that it may be challenging to interpret the results with cytochalasin D as it now turns out that actin is required for plastid movement and not just anchoring in these cells. Interestingly, this duality of actin being required for both movement and anchoring was also observed by Wada and colleagues for light driven plastid movement (it would be good to point this out in the Discussion).

Thank you for this suggestion and we added in the following sentence: “Interestingly, the N-terminal coiled-coiled domain of CHUP1 is also required to anchor chloroplasts to the plasma membrane, revealing a complex, dual role of actin during chloroplast movement and anchoring (Oikawa et al., 2008)”.

I am, however, a bit confused by continuing to propose a model where plastids are directed or guided during the early stages of the clustering response by MT's, since no effect on clustering by MT disruption was observed. In other words, the new data do not support all the particulars of the model.

We speculate in the Discussion that since MTs are required for stromule extension and involved in chloroplast movement, they are likely also involved in stromule extension to nuclei. However, that is speculative and we changed the wording and added in that further studies need to be conducted: “The role of MTs during the formation of stromule-to-nuclei connections requires further studies. However, our data suggests that once those connections are formed, stromules may guide or pull chloroplasts towards the nucleus, which then results in perinuclear clustering of chloroplasts.”

There seemed to be general agreement in the first review that this last experimental section was not a strength of the manuscript. I believe it may be a good idea to leave these studies out for now and have them be part of a future manuscript addressing specifically the mechanism of the clustering response.

We agree that the other experimental sections of the manuscript are stronger, mainly because there are a lot of future experiments that could be done to elucidate the mechanisms involved in perinuclear clustering of chloroplasts. The need for future experiments can be viewed as a weakness, but also as a strength, since it will potentially spur discussions in our field and additional studies. Therefore, we thank the reviewer for suggesting making this a separate, future manuscript, but after much discussion, we believe presenting the data we have now will be more beneficial to other in this field of research.

Other comments and questions on the revised text:Data questions:1) Section “Microtubules are required for stromule formation and extension”: In relating the results that support a role for MT's in stromule extension, it is stated that stromules in the drug treated cells completely retract after 15 min. of treatment. If this is the case, can the authors comment on the data shown in Figure 3, which indicate only a modest drop in the number of observed stromules in the drug treated cells from time 0' to time 15'? Also, it is not clear that these differences are significant. The standard errors for APM look like they mutually overlap the means at 2 sigma (0' to 15'), It is not clear what is going on with the stats for the oryzalin data.

We apologize for the confusion, but the statement in the subsection “Microtubules are required for stromule formation and extension”, was a description of a single data set shown in Figure 3 and Video 5; the quantification was shown in Figure 3. It actually highlights how stromules will stay extended even on a small fragment of MT. We made it clear that the descriptions of stromule disruption are single examples shown in Figure 3 and Video 5 in the aforementioned subsection.

We are only showing significance between the DMSO vehicle control and APM or the DMSO vehicle control and oryzalin. Since there was an increase in stromules in the DMSO control, the significance was only examined at time 15’. There was no significance between time 0’ to time 15’, and consequently, we did not report significance. We now stated this explicitly in the Figure 1 legend as: “Compared to the DMSO vehicle control, stromules significantly decrease after treatment with APM (20 µM) and Oryzalin (300 µM) at 15 min; no other comparisons were significant.”

We would also like to mention that new extensive data on stromule dynamics after oryzalin treatment were add in Figure 7, Figure 8, and 10 and complement these data in the last revision.

2) Section “Actin filaments serve as anchor points but not as tracks for stromule extension”, last paragraph: Can the authors comment on the observation that 70% of stromules were observed to only partially retract, a state that is attributed to actin interaction, whereas the number of observed stromules in cells treated with 200 µM CTD was the same as in mock treatments? If actin interaction is the primary means of preventing complete retraction of stromules, it seems that the number of observed stromules would be expected to be lower in cells where actin is disrupted.

The reviewer is correct that we expected to see a decrease with CTD, but did not observe one at 30 min after CTD treatment. In time-lapse data of this treatment, we saw that stromules initially retracted, but then re-extended on MTs (Figure 8—figure supplement 2, Video 11), and we speculate it’s this re-extension that results in no significant change after 30 min. It is also possible that actin anchoring just alters the dynamics of stromules and stromule driven movement as shown in Figure 7, Figure 9 and Figure 10, and has little role in maintaining the number of stromules. We hope that we or others will discover the proteins required for these interactions, so that their function can be studied more extensively in the future.

[Editors' note: further revisions were requested prior to acceptance, as described below.]

The manuscript has been improved but there are some remaining issues that need to be addressed before acceptance. These are outlined below under reviewer #3. Additionally, since the work was in review, another paper appeared in the Plant Journal with similar results to yours. You should refer to the other study in your revised manuscript and point out any similarities/differences in the work (see doi: 10.1111/tpj.13813).

We also noticed the online version of Plant Journal paper while our third revised version of the manuscript was under review. As suggested, we have added similarities and differences in our Discussion section. The main similarity is that stromules extend along MT. Our study shows this more extensively and quantitatively, by using three different markers and reporting different stromule morphologies, the percent interaction with MTs, stromule length, the rate of extension and retraction under multiple conditions, including during an innate immune response and with various cytoskeleton inhibitors. Both studies with Oryzalin support the conclusion that the role of MTs during stromule formation was under appreciated because stromules can hold on to small fragments of MTs and complete disruption of MTs with Oryzalin is difficult.

There are multiple differences between our paper and the Plant Journal paper. One of the largest differences is their model proposes slow moving anchoring points on MT, while we propose anchoring points along AFs. Their study did not mark and monitor AF and stromules simultaneously, and consequently, they would not have observed AF anchoring. Our study marked MTs, AFs, and stromules simultaneously to observe their complex interactions. They also suggest that fast moving stromules move along AFs, but they did not mark AF to show the extension. We also saw fast moving stromules that did not associate with MTs. However, we did mark AF, but interestingly, we did not observe them along AFs. We propose that they extend by an unknown force, potentially an internal chloroplast force. Our paper also shows that perinuclear association of stromules with nuclei is mediated by AFs. Lastly, our work proposes a novel role for stromules during chloroplast movement. Overall, we are excited about this similar publication and anticipate that the similarities and differences of these papers published nearly back-to-back will spark a lot of discussion, enhancing the impact of both papers.

Reviewer #3:The authors have addressed my previous concerns save one. At the end of the Results, the authors conclude:"In conclusion, we propose a model in which perinuclear clustering of chloroplasts involves stromule extension along the MTs, stabilization of these extensions by anchor points to AFs surrounding nuclei and stromules guide chloroplasts towards nuclei during an immune response."However, the authors reported that no connections with microtubules by stromules were observed during P50 induction of perinuclear clustering, and treatment with oryzalin had no effect on clustering. Thus, the experimental results do not support a role for guidance of stromules by microtubules in perinuclear clustering, indeed, they appear to contradict this possibility. The statement about microtubule guidance needs to be omitted from the proposed model for perinuclear clustering.

We have modified the statement: "In conclusion, we propose a model in which perinuclear clustering of chloroplasts involves stromule anchoring to AFs surrounding nuclei and stromules guide chloroplasts towards nuclei during an immune response.”

The description of the experiments addressing pausing of stromule retraction at AFs is now much more clear. However, the statistics in Figure 6 compare the rates of stromule pausing with the rate of full retraction. This comparison is relevant for asking if pausing is significantly more frequent than full retraction, but the main question here is whether pausing occurs at AFs more frequently than might be explained by chance alone. The rate of pausing at AFs by chance alone is a function of AF density; the higher the AF density, the more often pauses would be observed in association with them, even if there is no functional connection. Thus, AF density needs to be taken into account in a statistical test for the observed rate of AF-associated pausing (this can be a bit tricky due to the dynamics of the AFs). However, it seems pretty clear from the figures and the movies that the density of labeled AF's is lower than could easily explain a 77% rate of pausing at labeled AFs by chance alone (this would mean that, on average, about 3/4 of the stromule length would overlap AF signal). Even though the association is challenging to test formally, a statement to this effect might make more clear why the observed rate of 77% is a reasonable suggestion of an association.

The assay that we conducted to look at stromule pausing during retraction was new here, and we thank the reviewer for further assisting us in describing the method and results. We agree that pausing at AFs occurs more frequently than might be explained by chance alone and we have added this to the description. We added in the following statement to that effect: “The pausing of retracting stromules at AF cannot be explained by chance alone because the density of AFs and the colocalization of stromules with AFs observed appeared to be much less than 77.1% (Figure 6; Figure 6—figure supplement 1; Video 7).”